# Countering the potential re-emergence of a deadly infectious disease—Information warfare, identifying strategic threats, launching countermeasures

**Rex N. Ali**[1]\*, **Harvey Rubin**[2], **Saswati Sarkar**[1]

**1** Department of Electrical and Systems Engineering, University of Pennsylvania, Philadelphia, PA, United States of America, **2** Perelman School of Medicine, University of Pennsylvania, Philadelphia, PA, United States of America

\* alirex@seas.upenn.edu

## Abstract

### Objectives

Eradicated infectious diseases like smallpox can re-emerge through accident or the designs of bioterrorists, and cause heavy casualties. Presently, the populace is largely susceptible as only a small percentage is vaccinated, and their immunity is likely to have waned. And when the disease re-emerges, the susceptible individuals may be manipulated by disinformation on Social Media to refuse vaccines. Thus, a combination of countermeasures consisting of antiviral drugs and vaccines and a range of policies for their application need to be investigated. Opinions regarding whether to receive vaccines evolve over time through social exchanges via networks that overlap with but are not identical to the disease propagation networks. These couple the spread of the biological and information contagion and necessitate a joint investigation of the two.

### Methods

We develop a computationally tractable metapopulation epidemiological model that captures the joint spatio-temporal evolution of an infectious disease (e.g., smallpox, COVID-19) and opinion dynamics.

### Results

Considering smallpox, the computations based on the model show that opinion dynamics have a substantial impact on the fatality count. Towards understanding how perpetrators are likely to seed the infection, we identify a) the initial distribution of infected individuals that maximize the overall fatality count; and b) which habitation structures are more vulnerable to outbreaks. We assess the relative efficacy of different countermeasures and conclude that a combination of vaccines and drugs minimize the fatalities, and by itself, drugs reduce fatalities more than the vaccine. Accordingly, we assess the impact of increase in the supply of drugs and identify the most effective among a collection of policies for administering of

**Data Availability Statement:** All relevant data are within the manuscript and its Supporting information files.

**Funding:** SIGA Technologies, Inc. has provided gift to support the Sarkar lab. The funders had no role in study design, data collection and analysis, decision to publish, or preparation of the manuscript.

**Competing interests:** SIGA Technologies, Inc. has provided our laboratory a monetary gift. This does not alter our adherence to PLOS ONE policies on sharing data and materials. SIGA Technologies, Inc. had no role in the study design, data collection and analysis, decision to publish, or preparation of the manuscript.

drugs for various parameter combinations. Many of the observed patterns are stable to variations of a diverse set of parameters.

## Conclusions

Our findings provide a quantitative foundation for various important elements of public health discourse that have largely been conducted qualitatively.

## 1 Introduction

The devastating potential of a sudden outbreak of an infectious disease is self-evident in this time of a pandemic. The havoc caused by the COVID-19 outbreak may not only be replicated but may also be amplified should an eradicated infectious disease re-emerge. While naturally occurring smallpox was eradicated in the 1970s through decades of a global vaccination campaign, it could re-emerge under various scenarios [1–4]. Smallpox is considered a "potential weapon in a bioterrorism attack" [5]. Variola virus (smallpox virus) is a category A bioterrorism agent [6] and stocks of the virus are known to officially exist in two high-security biosafety level 4 laboratories in the United States (Centers for Disease Control and Prevention) and Russia (VECTOR Institute) and potentially elsewhere too [1–4]. For instance, in 2014, scientists at the National Institute of Health (NIH) discovered a half-dozen forgotten vials of smallpox in a storage room on its campus in Bethesda, Maryland [7, 8]. In addition, a virus similar to Variola, horsepox, has recently been synthesized from genetic pieces ordered in the mail [9], and smallpox may be recreated using similar techniques.

If smallpox does re-emerge, vaccine hesitancy—that is, refusing immunization on non-medical grounds (e.g., religious and philosophical beliefs, safety concern, disinformation)—can thwart attempts to proactively prevent it. Anti-vaccination movements also known as "Antivaxxers" have routinely propagated disinformation regarding immunization for various diseases, e.g., that measles, mumps, and rubella (MMR) vaccine causes autism [10, 11]. A recent survey has shown that only about 50% of women in the US would opt for COVID-19 vaccine [12]. US media estimates that about 50% of US service members would decline COVID-19 vaccine. US marine services has put this number at 40% among US Marines, and 57% at one of its prominent bases, Camp Lejeune in North Carolina [13]. Vaccine hesitancy caused a spike in measles outbreak in 2019 [14]. Consequently, the World Health Organization (WHO) added vaccine hesitancy to their list of top 10 threats to global health in 2019 [15]. Vaccine hesitancy will particularly damage the containment of future outbreaks of smallpox because only a small percentage of the current populace is vaccinated, and their protection is likely to have waned. In 1972 routine administration of the smallpox vaccine to infants was discontinued in the US. In 1976 and 1982 respectively, administration of the vaccine to healthcare workers and international travelers were discontinued. The vaccine is no longer available for the public, and only recommended for some military personnel and lab workers who work with a related virus [5].

Social media has fueled vaccine hesitancy by escalating disinformation on immunization [16], and in the event of an outbreak may enable malefactors who seed the disease to simultaneously amplify anti-vaccine campaigns and manipulate the target populace to refuse preventives. Also, in the age of social media, opinions regarding receptivity to vaccines rapidly evolve, through social networks that overlap with but are not identical to biological networks. Specifically, during physical interactions, both diseases and opinions may spread, whereas only

opinions may spread through remote (e.g., electronic) interactions. In some cases, only the disease might spread because individuals share the same physical space (e.g., public spaces like beaches, parks, public transports) without exchanging ideas. Thus, spread of smallpox and opinion regarding receptivity of vaccines must be jointly investigated.

Vaccine-hesitant individuals are likely to be more receptive towards receiving drugs. Additionally, underlying medical conditions like immunocompromise contraindicate a section of the populace from receiving fast-acting live vaccines. Both vaccine-hesitant and immunodeficient (also known as immunocompromised) individuals can benefit from antiviral drugs (e.g., tecovirimat (TPOXX) and cidofovir [17]) which may be used to both treat infected persons as well as prevent disease in those exposed. These are currently stockpiled in the Strategic National Stockpile for use if there is an outbreak of smallpox in the United States [17]. Drugs have an advantage in a public health setting because they can be administered without the intervention of a healthcare professional, e.g., TPOXX is taken orally twice daily [18]. The downside, however, is that they need to be taken for many days throughout the outbreak. Thus, considering the comparative advantages and disadvantages of drugs vis-a-vis vaccines, only a combination of countermeasures may be able to successfully counter smallpox should it reemerge. One can envision different policies for administering antiviral drugs: (a) Policy 1—administer to those who have fever or rash; (b) Policy 2 –administer only to those with rash; (c) Policy 3—once the number of cases in a neighborhood exceeds a certain threshold, administer to everyone in the neighborhood. We, therefore, need a framework to assess the application of a combination of countermeasures and multiple countermeasure application policies. Note that the supply of the countermeasures would be finite in practice, a constraint that ought to influence the design of informed countermeasures.

Finally, geography, spatial distribution and mobility patterns of individuals is crucial in understanding the spread of an infectious disease and the efficacy of countermeasures. Locations inhabited by the target population are comprised of different neighborhoods, some of which are adjacent, some geographically disparate. Connections between individuals who are in geographically distant neighborhoods is expected to be more frequent than those between the disparate ones leading to heterogeneous interaction rates. Beyond common wisdom, this observation has recently been substantiated through data analysis and theoretical modeling [19, 20]. Specifically, analyzing friendship links on Facebook between individuals in every county-pair, Bailey *et. al.* have shown that the intensity of friendship links between counties in US declines strongly with increase in geographic distance between them [19]. It is worth noting that as of September 2014, more than 58% of the US adult population used Facebook and in US, Facebook friends are mostly mainly real-world friends and acquaintances. Thus, Facebook friendship network is representative of a real-world friendship network [19]. Using a theoretical model, Patacchini *et. al.* have independently observed that the intensity of social relationships between two individuals decreases with the geographical distance between them [20].

Next, interactions often happen through mobility which must observe geographical constraints. That is, when individuals move from one neighborhood to another, they pass through those in-between; thus, moving between two geographically disparate neighborhoods involves encountering individuals in the intervening neighborhoods. The effectiveness of the countermeasure may well be substantially enhanced by exploiting groupings of populations in distinct locales. Additionally, the perpetrators of the attack may exploit the geography to strategically implant the initially infected individuals and persuade individuals in specific regions to refuse vaccines, following a spatial distribution that maximizes the spread of the disease.

Mathematical modeling is an indispensable analytical tool that can help us prepare for and respond to a smallpox incident. Several models have been constructed to understand the

spread of smallpox and the efficacy of vaccination and quarantine as a response [1, 3, 21–25]. Note that immunocompetent individuals (that is, those with healthy immune systems) can be immunized by the live vaccine (ACAM2000) which provides immunity in a short time after administering if taken while they are susceptible (that is before they are infected) or in the early part of incubation (that is shortly after infection) [1, 26]. On the other hand, the immunodeficient individuals can only be given the Modified Vaccinia virus Ankara (MVA) vaccine (Imvamune) [1, 26]. MVA takes two-shots 30 days apart to provide immunity [26]. Although approximately 20% of people in the United States are immunodeficient [24, 27, 28], the existing work largely does not consider the impact of the immunodeficient population. However, [3, 24] considered such persons in their model, assuming though that they are contraindicated from all vaccinations and excluded them from participating in vaccination altogether. However, [1, 26] have shown that immunodeficient persons can receive the MVA vaccine but it will take a longer time to provide immunity. Only one study (i.e., [1]) has considered administering drugs. The evolution of opinions, mobility, impact of geography, finite supply of drugs, have not been considered even in isolation in the context of evolution of smallpox. None of the existing research naturally *simultaneously* modeled these variables, particularly in conjunction with the combination of multiple countermeasures and different application strategies.

Epidemiological investigation of other infectious diseases have considered spatial heterogeneity in the spread of infection e.g., [29, 30]. But these have not investigated questions particularly relevant for bioterrorist attacks, namely 1) which spatial distributions of the initially infected individuals maximize the spread of the disease; 2) which topological connectivities of habitations enhance vulnerability to infectious epidemics. Most importantly, the joint spatio-temporal spread of disease and opinion dynamics and the impact of one on the other remain an unchartered territory in the modeling of infectious diseases. Thus, public discourse on the correlation between the two dependent evolving processes has remained qualitative. These opinion dynamics, encompassing receptivity to vaccine once developed and wearing protective gears like masks, is expected to strongly influence the evolution of many infectious diseases, including COVID-19, in the present moment. The role of the social and biological networks and their overlap in this joint spread need to be understood. For example, do spatial distributions of the initially infected individuals, distributions of those initially professing a specific opinion, the choice of countermeasures (since receptivity to different countermeasures differ), and mobility rates have significant impact on the evolution of opinions and thereby on fatality counts? Given the myriad of the parameters that influence the joint spread, does the nature of the variation of fatality counts with respect to one parameter change drastically when another parameter is varied? These questions arise because of an apparent intrinsic overlap between information warfare, strategic threats of bioterrorism (e.g., the spatial distribution of the initially infected may be strategically selected if the outbreak is the result of bioterrorism) and the choice of countermeasures. In this paper we undertake the first step towards answering these questions.

Our contributions are as follows. We develop a computationally tractable mathematical model that jointly captures the evolution of a smallpox incident and vaccine hesitancy (opinion dynamics) over time and space and the impact of various combinations of spatial topologies, mobility rates, opinion exchange rates, disease spread rates, distributions of the initially infected and vaccine-hesitants, countermeasures, and strategies for their applications. The model captures the essence of stochastic evolution, while retaining computational tractability, and therefore easily scales to typical target population sizes for infectious diseases encompassing millions of individuals. We utilize the model to quantify the impact of the opinion dynamics, different countermeasures and application strategies, topologies, distributions of the initially infected and vaccine-hesitants and mobility patterns on metrics that capture the

overall health of the system such as total number of fatalities, and visits to health-care facilities. This provides a quantitative foundation to public health discourse pertaining to the relation between disease and opinion spreads that have largely been conducted in the qualitative sphere thus far. The quantifications confirm several common-place intuitions, and go beyond by unearthing the exact nature of dependence of the above public health metrics on several key parameters and helps us anticipate the strategic choices that future potential bioterrorist attacks are likely to adopt. We also discover several stable patterns of variations of public health metrics with respect to important parameters. These patterns are stable in that the nature of the variations with respect to a parameter does not change if values of the other parameters change. Since many of these parameters assume widely differing values in different environments that arise in practice, the recurrence of stable patterns is an important finding. This is likely to simplify public health policy choices pertaining to incentivizing the spread of opinions favorable towards reception of vaccines over social and other media and the choice of new urban designs resilient to pandemics. Finally, we have chosen smallpox as a specific example of an infectious disease, since (1) it is highly infectious; (2) it has a high death rate; and (3) its disease progression parameters are known with reasonable certainty owing to years of research. But, our framework ports to any other infectious disease (e.g., COVID-19) that spreads between individuals in proximity, through the consideration of a different set of disease states and parameters. We illustrate this generalization considering COVID-19 as an example.

## 2 Methods

### 2.1 Developing the model—State transition formulations

Infectious diseases evolve in different stages. Each stage exhibits different symptoms and the initial stages need not show any symptoms. In smallpox, individuals move from the stage of susceptibility to early incubation to late incubation to prodrome to early rash and then to late rash. From the late rash, patients either recover or die. Different stages have different durations. The incubation phase typically lasts 8 to 17 days and does not have any symptoms. The prodromal phase begins at the onset of the first symptoms (fever, chills, headache) and lasts for 3 days on average. The early and late rash periods last for 3 and 7 days respectively [1]. We use *susceptibles*, *incubators*, *prodromals* to denote the individuals in susceptible, incubation and prodrome stages respectively. Patients can infect susceptible individuals in the prodrome, early and late rash stages.

Efficacy of countermeasures depends upon the stage in which they are administered. Vaccines prevent the onset of smallpox with certain probabilities if administered prior to the late incubation stage. Live vaccines (ACAM2000) provide immediate immunity during this period [26]. Immunodeficient individuals are contraindicated for receiving the live vaccine, instead they are administered the MVA vaccine (Imvamune) [1, 26]. MVA takes two-shots 30 days apart to provide immunity [26]. Antiviral drugs like tecovirimat (TPOXX) and cidofovir [17] are generally administered to infected persons for treatment and can be used for prevention during the period in which they are administered.

The application of countermeasures introduces additional states, namely preempted, which we discuss in greater detail in the respective sections (Sections 2.1.2—2.1.4).

We now describe the state transitions, progressively considering the following scenarios: (1) No countermeasure—neither drugs nor vaccines administered; (2) Drug only; (3) Vaccine only; (4) Both drug and vaccine.

**2.1.1 No countermeasure.** We first consider the case that all individuals are in the same neighborhood, that is, they interact with each other at the same rate (homogeneous mixing). Each individual is either immunocompetent or immunodeficient. Individuals in either

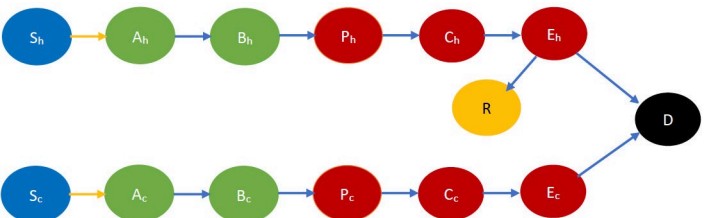

**Fig 1. Smallpox disease progression overview when no countermeasure is implemented.** The symbol S denotes susceptible, A and B denote early and late incubation respectively, P denotes prodrome, C and E denote early and late rash respectively, R denote recovered and D denotes dead. The suffix *h* denotes immunocompetency, and *c* denotes immunodeficiency e.g., $S_h$, $S_c$ respectively denote immunocompetent and immunodeficient susceptible, etc. Table 4 in Appendix A.1 of S1 Appendix contains all the relevant abbreviations we used to represent each state. The states in blue color are the susceptibles (not yet infected but they are prone to infection) while those in light green color are still in incubation period, hence they are not infectious. The states in dark red are infectious while those in gold have recovered and black denotes dead. In addition, the yellow arrows show interactional changes, namely susceptibles transitioning to the incubation state after contracting the virus from infectious individuals. The blue arrows indicate a non-interactional change, namely the natural progression of the disease.

category may be in one of the following states: susceptible, early incubation, late incubation, prodrome, early rash, late rash, recovered, dead. Fig 1 depicts the state transitions pictorially. There are two kinds of transitions: (1) *interactional* and (2) *non-interactional*. The first kind of transition occurs as a result of physical interactions in which the biological contagion spreads, that is, when two individuals are in close proximity and one is susceptible while the other is infectious (that is in one of the following states prodrome, early rash, late rash), then the susceptible is infected with a certain probability and transitions to early incubation stage. The second kind of transition occurs as a result of the natural evolution of the disease in infected individuals. For instance, the disease transition from early incubation to late incubation and from late incubation to prodrome after about 7 and 5 days respectively. From the late rash stage, the immunocompetent individuals either transition to the recovered or to the dead state, while immunodeficient individuals invariably die [31].

**2.1.2 Drug only.** We consider three policies for administering drugs: policies 1, 2, 3. Under the first two policies only the people with symptoms are administered drugs (refer to Introduction for the descriptions of these policies). Animal trials have shown that treatment with antiviral drugs cure (with a certain probability) animals who have already received the virus even after they show symptoms. The cured animals do not develop the disease even if they receive the virus in future [32]. We assume the same for human (like [1]). Thus, under these policies, upon receiving the drug, individuals enter a state in which they do not have the disease nor do they develop the disease in future. We, therefore, introduce a state called *preempted*, which we denote by Q, into which individuals receiving the drug transition to with the specified probability. It is an *absorbing* state, i.e., individuals can only enter this state, not leave it.

Under the third policy, drugs are administered to everyone, including those not exposed, in neighborhoods of heavy outbreak. Antiviral drugs prevent the occurrence of smallpox (with a certain probability) during the period they are administered, e.g., TPOXX prevents smallpox during the period it is taken orally twice daily [1, 18]. We assume that once individuals without symptoms are administered this drug, to prevent the onset of smallpox even after possible exposure, they continue to receive the drug until the disease is completely contained. Thus, once an individual starts to receive the drug he can no longer have smallpox (with the specified probability) until containment of the outbreak (that is, until the end of the duration we consider). In this sense, we consider that antiviral drugs preempt smallpox. Thus, for the purpose

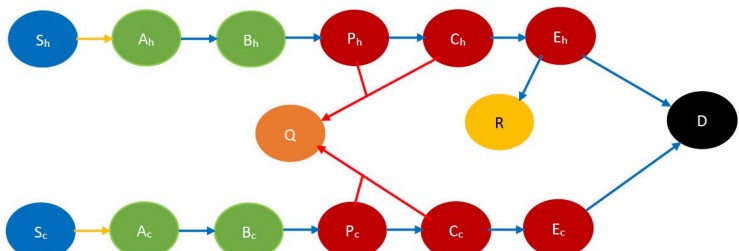

**Fig 2. Smallpox disease progression overview for "drug only" scenario.** The state in orange color denotes the preempted state. The red arrows denote preemption via drugs. Every other state and transition has the same meaning as described in the caption for Fig 1.

of our model, without loss of generality, individuals (with or without symptoms) enter the *preempted* state (denote by $Q$) with the specified probability, if they are administered drugs. The probability is 0.99 before the onset of rash, 0.8 during early rash and 0 during late rash [1].

The states and state transitions are otherwise similar to that of the no countermeasure scenario. The state transitions have been depicted pictorially in Fig 2.

**2.1.3 Vaccine only.** The individuals who receive vaccine transition to the preempted state, after developing permanent immunity to the infectious disease. There is a delay incurred in developing this immunity after the vaccination process is completed and even then immunity is developed with a certain probability. Thus the recipients transition to the preempted state after this delay and with the associated probability. The probability is 1 (0.8, respectively) for susceptibles (early incubators, respectively) [1]. Vaccines are not effective when administered in late incubation stage or beyond; thus, there is no transition from these states to the preempted state. Some individuals may not be willing to receive vaccines. We refer to those willing to receive vaccine as *cooperative*, and the rest as *non-cooperative*. Opinion regarding cooperation evolves with interactions with other individuals. Thus, this scenario needs to model both opinion and disease dynamics and the two are coupled. Thus, the state space needs also to be enriched to consider opinions and their evolutions. Finally, one also needs to distinguish between immunocompetent and immunodeficient individuals because the latter can receive vaccines that act only very slowly regardless of their willingness to do so, and may not recover from a serious infectious disease [31].

Interactions can be of the following kinds: (a) physical interactions with an exchange of opinion and biological contagion (e.g., friends and acquaintances visiting homes of each other); (b) physical interactions without any exchange of ideas (e.g., people commuting on a bus, train, etc.); (c) virtual interactions with an exchange of ideas (e.g., a health worker counseling a susceptible individual over the phone or internet). (a) can cause infection and change in opinion, (b) can cause only infection, (c) can cause only change of opinion. All the above represent *interactional transitions* in this case in which both disease and opinions spread through interactions. Fig 3 depicts the state transitions of this scenario pictorially. Here we only consider the case that the cooperatives persuade the non-cooperatives to become cooperatives during opinion exchange, but in practice the opinion exchange may change opinions in the reverse direction too. In Section 2.1.4, we discuss how opinion changes in the reverse direction may be accommodated through minor modifications.

Referring to the above state transitions, we need to consider four different outcomes for interaction between individuals:

(a). Neither of the individuals gets infected or changes their opinion after the interaction. For instance, let a cooperative susceptible, $S_{ah}$, interact with a cooperative early incubator

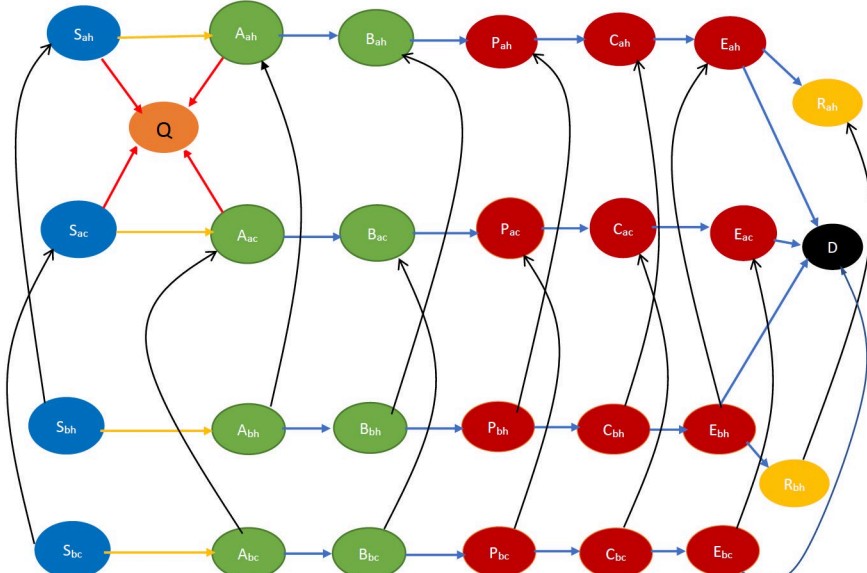

**Fig 3. Smallpox disease progression overview for vaccine only scenario.** Building on the definitions of states in Fig 1, the suffixes *a* and *b* respectively denote willingness to receive vaccine and otherwise e.g., $S_{ah}$, $S_{ac}$ respectively denote immunocompetent and immunodeficient susceptibles that are willing to vaccinate. Similarly, $S_{bh}$, $S_{bc}$ respectively denote immunocompetent and immunodeficient susceptibles that are not willing to vaccinate. The states in the top two rows and the transitions between them are identical to those in the bottom two rows except for the index "a" being replaced by "b". The difference in this index represents a partition of these states based on the willingness to vaccinate. Sans this partition, the states and the transitions are similar to those in Fig 2—specifically, the states in the top two rows and the transitions between them, and those in the bottom two rows, by themselves, are similar to the states and the transitions in Fig 2 (except that the transition to the preempted state happens from a larger set of states in Fig 2). Table 7 in Appendix A.3 of S1 Appendix contains all the relevant abbreviations in this scenario. The states in blue color are the susceptibles (not yet infected but they are prone to infection) while those in light green color are still in incubation period, hence they are not infectious. The states in dark red are infectious while those in gold have recovered, those in yellow are immunized, and black denotes dead. In addition, the yellow arrows show susceptibles transitioning to the incubation state after contracting the virus. The blue arrows indicate the natural progression of the disease. The red arrows denote preemption via vaccination while the black arrows indicate opinion evolution.

say $A_{ah}$. This interaction does not lead to infection neither does it lead to a change in opinion about immunization.

(b). Neither of the individuals gets infected, but one (and only one) changes his opinion, e.g., a cooperative susceptible, $S_{ah}$, interacts with a non-cooperative early incubator, $A_{bh}$. The susceptible does not contract the infection but the early incubator might change his opinion to become $A_{ah}$.

(c). One of the individuals gets infected, but neither changes his opinion. This happens for example in a physical interaction between a susceptible (e.g., $S_{bc}$) and an infectious individual (e.g., $P_{bc}$) both of whom share the same opinion.

(d). One of the individuals gets infected, and the other changes his opinion. This happens for example in a physical interaction between a susceptible (e.g., $S_{bc}$) and an infectious individual (e.g., $P_{ah}$) who have different opinions ($S_{bc}$ may become $S_{ac}$ after the exchange).

**2.1.4 Both drug and vaccine.** An individual may be preempted by receiving either the drug or the vaccine. The preempted states are $Q_a$ or $Q_b$ respectively representing preempted

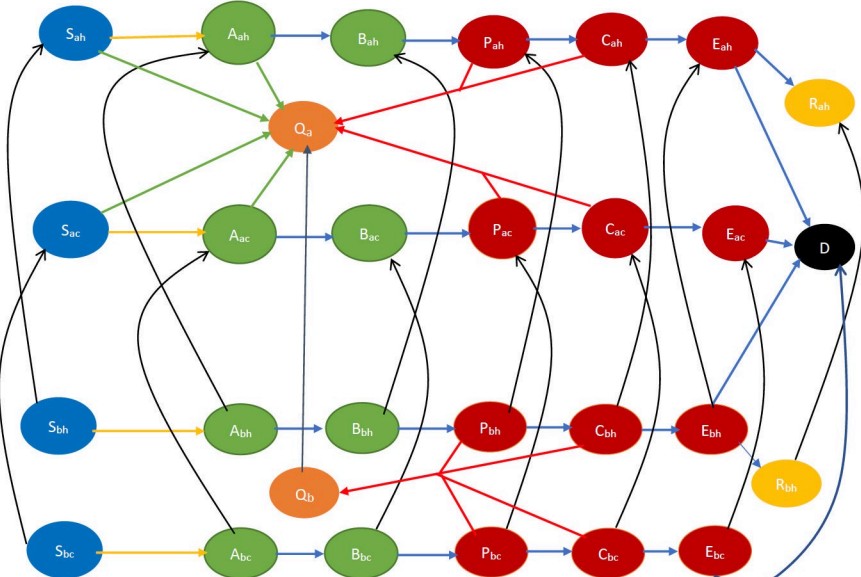

**Fig 4. Smallpox disease progression overview for both drug and vaccine scenario.** The states colored orange are the preempted states. Every other state and transition has the same meaning as described in the caption for Fig 3.

cooperative and non-cooperative individuals. An individual can reach $Q_a$ by receiving either drug or vaccine, while he can reach $Q_b$ only by receiving drug. Although an individual transitions to the preempted state upon receiving either drug or vaccine, he may receive *both* drug and vaccine in the course of the outbreak. For example, if a vaccine is not effective in an individual (that is, his immunity does not increase to the level that future infection is prevented) then he does not transition to the preempted state after receiving the vaccine. Then even after receiving the vaccine he may develop the disease upon receiving the virus from an infectious individual. Once he develops symptoms he may be treated with the drug, which may cure him and then he would transition to the preempted state. Similarly, an individual who has not been infected may be treated with the drug and during the treatment he transitions to the preempted state. Subsequently, instead of continuing his drug treatment until the end of the outbreak, he may be vaccinated and the drug treatment terminated after the vaccine develops the immunity in him which ensures that he remains in the preempted state. Trials on animals have revealed that the combination does not cause adverse physiological impact [32].

The state transition in this "both drug and vaccine" case may be obtained by combining those of the "drug only" and "vaccine only" scenarios—more specifically, by adding to the vaccine-only scenario, the transitions to the *preempted state* induced by delivery of drugs. Refer to Fig 4 for the state transitions. Note that in this figure, we assume that during opinion exchange cooperatives convert non-cooperatives. To accommodate persuasion in the opposite direction, one simply needs to invert the state transition directions corresponding to opinion exchange, namely the directions of the black arrows.

## 2.2 Capturing the impact of spatial heterogeneity

We divide the target geographical area into smaller regions, referred to as *clusters*, such that the constituent regions are reasonably spatially homogeneous, that is, individuals inhabiting that region interact with each other at similar rate. The clusters can for example be neighborhoods of a city. The target geographical region can be a city, a county, a state, or even a

country. Clusters correspond to natural groupings of individuals in communities. Accordingly, people within the same cluster have high contact rates between themselves, while those in different clusters have fewer contacts.

So far, an individual has been characterized by his cooperativity, stage of the disease, preemption, immunocompetency or immunodeficiency. Now, he is also identified by the cluster he inhabits. Thus, the cluster becomes part of his state description, which also changes as he moves between the clusters.

To capture the geography of the target region and the mobility rates of the individuals between the clusters, we represent every cluster as a node on a graph. Now, the mobility rates across the clusters may be represented through a *mobility rate matrix* on the graph. It is a matrix with rows and columns labeled by the cluster numbers, and whose $i, j$ th entry represents the *mobility rate*, $\kappa_{i,j}$, of individuals in cluster $i$ to cluster $j$. Here, $\kappa_{i,j} = 0$ if it is not possible to move directly from $i$ to $j$ (e.g., if these are not geographically adjacent or if traffic rules do not permit vehicular mobility from $i$ to $j$). If it is possible to move from $i$ to $j$, then $\kappa_{i,j}$ represents the probability that an individual goes to cluster $j$ when he decides to move out of cluster $i$ divided by the expected time he spends in cluster $i$.

We have pictorially represented some cluster decompositions in Fig 5. The central node in Fig 5a for example represents the downtown of a city, and the other nodes represent the neighborhoods surrounding it. Fig 5b depicts the cluster decomposition of a commercial and business district located at the edge of a river, sea or ocean.

We use matrices to specify physical and virtual contact rates within and across clusters. Individuals in cluster $i$ get in physical proximity of another in cluster $j$ at a certain rate, the rate is much higher if $i = j$, than when $i \neq j$. Only a fraction of these contacts spread the disease. We consider that *disease spread rate* $\phi_{i,j}$ is the product of the rate at which an individual in cluster $i$ gets in physical contact with another in cluster $j$ and fraction of contacts between susceptibles and infectious individuals that spread the disease to the susceptibles. Rate of an event is the expected duration between successive occurrences of the event. Typically, $\phi_{i,i}$ would be much higher than $\phi_{i,j}$ for all $i \neq j$. Analogously, *opinion spread rate* $\alpha_{i,j}$ is the product of the rate at which an individual in cluster $i$ exchanges opinions with another in cluster $j$ and fraction of such exchanges that change cooperativity.

## 2.3 The Clustered Epidemiological Differential Equation (CEDE) model

A joint investigation of infectious disease and opinion dynamics in presence of spatial heterogeneity inevitably leads to a computationally complex model with 1) a multiplicity of states representing a combination of stages of the disease, immunocompetency, and cooperativity; and 2) a multiplicity of state transitions representing interactional transitions due to spread of

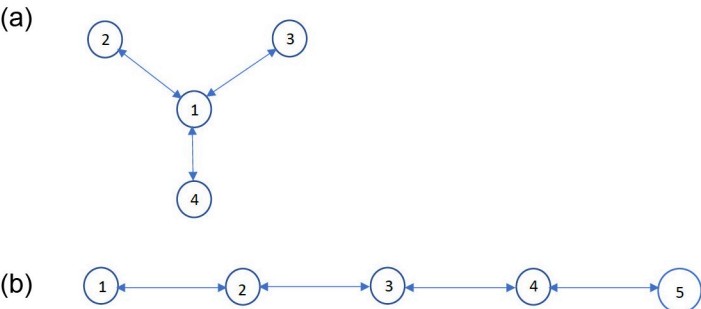

**Fig 5. Two example topologies.** (a) Star topology. (b) Linear topology.

the disease and opinions, non-interactional transitions due to mobility of individuals and the natural progression of the disease in infectious individuals. We model these by adapting the *metapopulation epidemiological model* [29, 30] which relies on a set of differential equations (co-authors of this work have utilized the metapopulation model as well [33, 34]). Metapopulation models have some distinct advantages in modeling the spread of infectious diseases. In general, estimating the spread of infectious diseases is computationally challenging because it involves millions of individuals in habitation sizes one needs to consider. Metapopulation models alleviate this challenge by relying on differential equations, which constitute computationally simple tools and can be solved using readily available numerical techniques. The differential equations capture the evolution of states of different fractions of the total population. Thus, the computational time does not increase with the increase in the size of the populace.

Metapopulation models have however thus far not captured the transitions due to spread of both disease and opinions and application of various countermeasures. We have been able to adapt the metapopulation model to capture these attributes, we describe the adaptations in Appendix A in S1 Appendix and refer to the resulting model as *clustered epidemiological differential equations* or the CEDE, given that our target area is spatially decomposed in subregions referred to as clusters.

We briefly describe the CEDE here. Each variable in the CEDE represents the fraction of the population who are in a particular system state, each state representing the combination of the cluster inhabited, the stage of the disease, immunocompetency, and cooperativity. Each differential equation captures the evolution of a particular variable. Thus, the solution of the system of differential equations provides the fraction of individuals in different states at given times, that is, the spatio-temporal distribution of the disease and opinion spread. The terms in the differential equations are either quadratic or linear. The quadratic ones represent the interactional transitions (refer to the yellow arrows in Figs 1–4 and the black arrows in Figs 3 and 4) and the linear ones represent the non-interactional transitions (refer to the blue arrows in Figs 1–4). Note that interactions always involve two individuals, hence interactional transitions are represented by quadratic terms; in contrast, the non-interactional transitions involve only one individual and are therefore represented by linear terms. This is typical of epidemiological models starting from the classical Kermack–McKendrick formulation [35] and onward to the metapopulation models [29, 30, 33, 34]. Our work differs from the metapopulation epidemiological models in that it captures two different evolving processes spreading simultaneously, the disease and the opinion, through different but possibly overlapping contact processes (physical proximity, opinion exchange); the metapopulation models typically capture only one evolving process, namely the disease, spreading through physical proximity. The spread of the two processes involves two broad categories of interactional transitions: (1) susceptible to early incubator; (2) noncooperative to cooperative and vice versa. The metapopulation models typically capture only the first kind of interactional transitions. Since interactional transitions are represented by quadratic terms, the second kind leads to additional quadratic terms in our model. Our work also has additional linear terms representing preemption due to application of countermeasures.

We now provide more details on the computation time of the CEDE. Let there be $n$ clusters. Then there are 29 x $n$ system states in the most complex CEDE we propose, that is for the scenario in which both vaccines and drugs are administered (Appendix A.4 in S1 Appendix). In this scenario, there are $29n$ differential equations and $29n$ variables. Thus, the computation time increases with the increase in the number of clusters and system states, but the rate of increase is linear in each of these. The linear increase ensures that the CEDE can be computed fast even for a large number of clusters, that is, when the target geographical area is large, e.g., of the size of a country.

The CEDE is however a deterministic model, while many of the state transitions are stochastic. Nevertheless, through an application of a classical result of probability theory, under some commonly made assumptions on the stochastic evolutions, one can show that as the number of individuals increases, the fractions of individuals in different system states in the stochastic system converge to the solutions of the CEDE, and the convergence becomes exact in the limit that the number of individuals is infinity. Thus the CEDE approximates the stochastic process better as the number of individuals increase. The assumptions under which the convergence guarantee holds is that the stochastic evolutions are *Markov*, that is, the amount of time an individual spends in each system state is *exponentially* distributed, which is what we assume to estimate the parameters of the system (Appendix B in S1 Appendix). Markovian assumptions are commonplace in modeling the spread of infectious diseases (e.g., as noted in Chapter 2, p. 28, [36]). In the Supporting Information, we have stated the classical result and have shown that it guarantees convergence in our specific case. The arguments therein are standard (similar arguments have been utilized in a recent work by one of the co-authors involving the application of the CEDE in a different domain [34]), albeit long.

Our CEDE model is inherently flexible in that it can accommodate opinion dynamics, arbitrary topologies, mobility patterns, countermeasure combinations, countermeasure application strategies and constraints (e.g., finite or infinite supply). In Section 3, we use the CEDE to evaluate different public health metrics under different combinations of the above attributes. Despite this modeling flexibility, the CEDE remains computationally tractable and provides analytical convergence guarantee.

Finally, considering COVID-19 as an example, we show in Section 5 that our framework ports to any other infectious disease that spreads between individuals in proximity, through the consideration of a different set of disease states and parameters.

## 3 Results

We utilized our model to obtain insights on the impact of (1) opinion dynamics as to receptivity to vaccine; (2) the spatial distribution of the initially infected; (3) various countermeasures; (4) geography; and (5) policies for administering the countermeasures on public health metrics. We mostly consider the total number of fatalities as the public health metric, but in some cases we also assess the number of visits to health care facilities.

We now describe how we choose the parameters for the numerical computation and their default values. In the numerical computation, unless otherwise stated, we use the default value for each parameter. But we also vary the value of each parameter in a wide range because in practice these parameters assume different values in different environments and there is no one standard value (with one exception which is explicitly stated below).

We consider both single and multiple clusters throughout, with each cluster consisting of 10 million individuals. We consider two specific examples of multiple clusters throughout: the star and linear topologies shown in Fig 5a and 5b. We choose these examples because they represent actual distribution of habitations (refer to the paragraph before Fig 5a and 5b). Besides these topologies are representative of two fundamentally different characteristics: 1) the star is centralized in that every cluster (other than the central cluster) is adjacent to the central cluster and the maximum distance between any two clusters is two; 2) the linear is spread out in that the maximum distance between two clusters increases with increase in the size of the topology, the distance between the peripheral clusters at the opposite end is $n - 1$ when there are $n$ clusters overall. Our default choices are $n = 4$ for star and $n = 5$ for linear, but we also consider several other values of $n$ for both topologies (Figs 12, 16—19) and compare the public health metrics across topologies and $n$ (Table 2).

We now describe the choice of the mobility rates. We consider that if clusters $i$, $j$ are not adjacent, the mobility rate $\kappa_{i,j} = 0$ and if they are adjacent $\kappa_{i,j} = \kappa$. For example, in Fig 5a, clusters 1, 2 are adjacent, while 2, 3 are not adjacent. Similarly, in Fig 5b, clusters 1, 2 are adjacent, while 1, 3 are not adjacent. Our default choice is $\kappa = 0.01$, but we also consider different values of $\kappa$ (Fig 8), and vary $\kappa$ in a wide range (Figs 12 and 13).

We now consider the disease spread rate $\phi_{i,j}$. We assume $\phi_{i,j} = 0$ if $i \neq j$, that is, there is no direct physical contact and therefore no direct spread of disease between people inhabiting different clusters. We refer to $\phi_{i,i}$ as $\phi$. $\phi$ can be obtained from *Basic Reproduction Number, $R_0$* following standard mathematical techniques [3, 37]. Now $R_0$ equals 6.9 for smallpox [1, 38], which provides the default $\phi$, $\phi = 0.00173$ (refer to Appendix B.1 in S1 Appendix for details). We also vary $\phi$ in a wide range in Figs 14a and 15a.

We next consider the opinion spread rate, $\alpha_{i,j}$. We assume that $\alpha_{i,j} = 0$ if $i \neq j$, that is individuals in different clusters do not directly exchange opinions. We consider that for all clusters $i$, $\alpha_{i,i} = \alpha$. For simplicity, we will refer to $\alpha$ as the opinion spread rate henceforth. The default choice for $\alpha$ is 0.001, but we also consider different values of $\alpha$ (Fig 10) and vary $\alpha$ in a wide range (Figs 6–9). We will consider two different scenarios in our computation: 1) cooperatives convert non-cooperatives during opinion exchange (default case); and 2) non-cooperatives convert cooperatives during opinion exchange.

We define the fractions of individuals who are willing to vaccinate at the initial time as *initial cooperativity*. Our default choice for initial cooperativity is 0.8 in each cluster, but we also consider other values of initial cooperativity (0.2 in Fig 9b, 9d and 9f, and 0.25 in Fig 13). We also vary cooperativity in a wide range (Figs 14b and 15b) and consider scenarios where initial cooperativity is nonuniform (Fig 13).

Our default choice for initial number of infected individuals is 10000. This choice has been motivated by the fact that given that smallpox has been extinct for sometime, its future outbreak may be a result of a bioterrorism attack. Such attacks often involve deliberate infection of a large number of individuals, as discussed in [1]. Also, since each cluster consists of 10 million individuals, even under the default choice, only a small fraction of the overall populace is infected initially. Our default assumption is that all the initially infected individuals are in an early stage of the disease, namely prodrome. Note that initially, the infected individuals are likely to be in an early stage of the disease. We choose prodrome as this initial stage as this is the first stage in which an individual becomes infectious. We also consider lower numbers of initially infected individuals, namely 1, 1000 (Figs 6, 7 and 9a and 9b), as also scenarios in which the initially infected individuals are distributed across different stages of the disease (Figs 7 and 9b). We consider three distributions of initially infected individuals across the clusters: (1) uniform distribution across clusters; (2) concentration in the central cluster (clusters 1 and 3 respectively in the star and linear topologies); and (3) concentration in a peripheral cluster (clusters 2 and 1 respectively in the star and linear topologies). These are respectively referred to as "Uniform", "Central" and "Peripheral", with "Uniform" being the default.

We choose the disease progression parameters and default values of the preemption rates utilizing data available in the existing literature on smallpox. Refer to Appendix B in S1 Appendix for a description of the methodology for calculating the above and also citations of the relevant literature. Briefly, in calculating the preemption rates we consider 1) the delay incurred in administering the vaccine or drug to an individual once he agrees to receive it (e.g., delay incurred in getting the health worker's appointment); and 2) the subsequent delay for the vaccine or drug to be effective (i.e., in preventing or curing the disease, prevention requires developing the bodily ability to not develop the disease even upon receiving the virus). Even after the latter delay, the vaccine or drug becomes effective only with a certain probability; this probability depends on the stage of the disease an individual is in. Accordingly, we define the

(a)

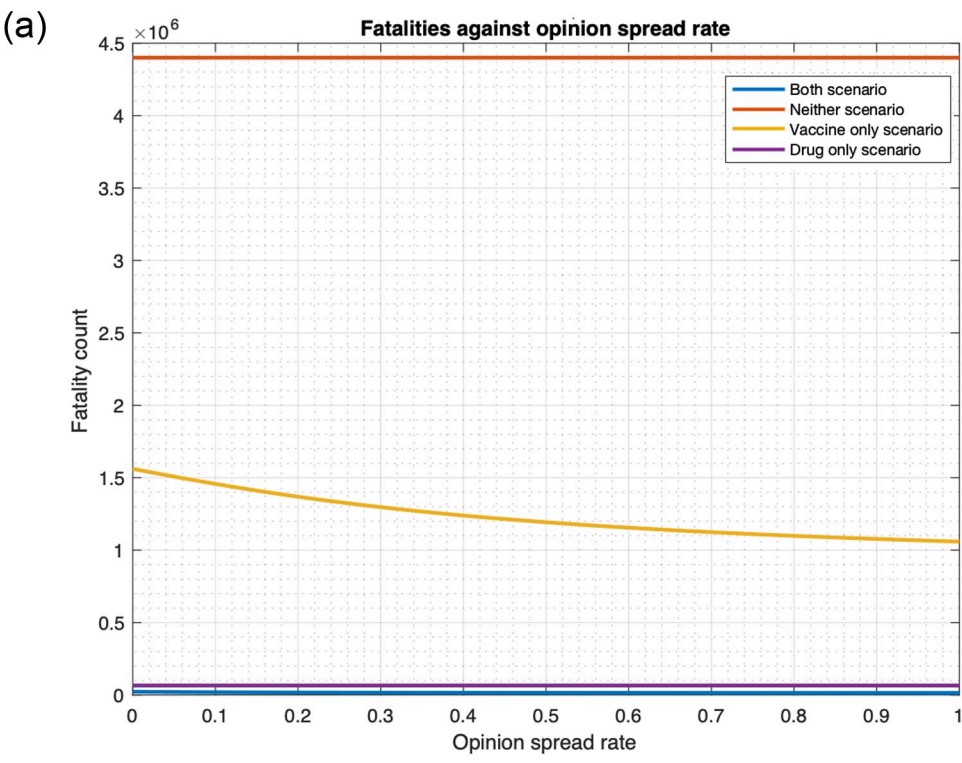

(b)

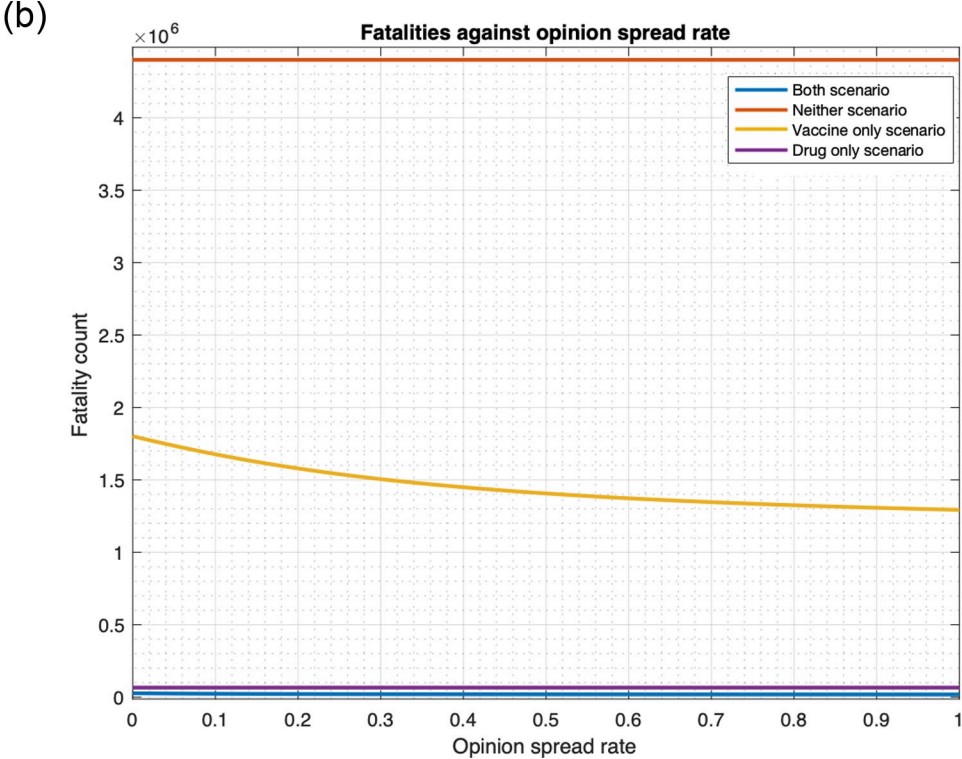

**Fig 6. Relationship between fatality and opinion spread rate (single cluster).** (a) Number of individuals infected at the initial time is 1,000. (b) Number of individuals infected at the initial time is 10,000.

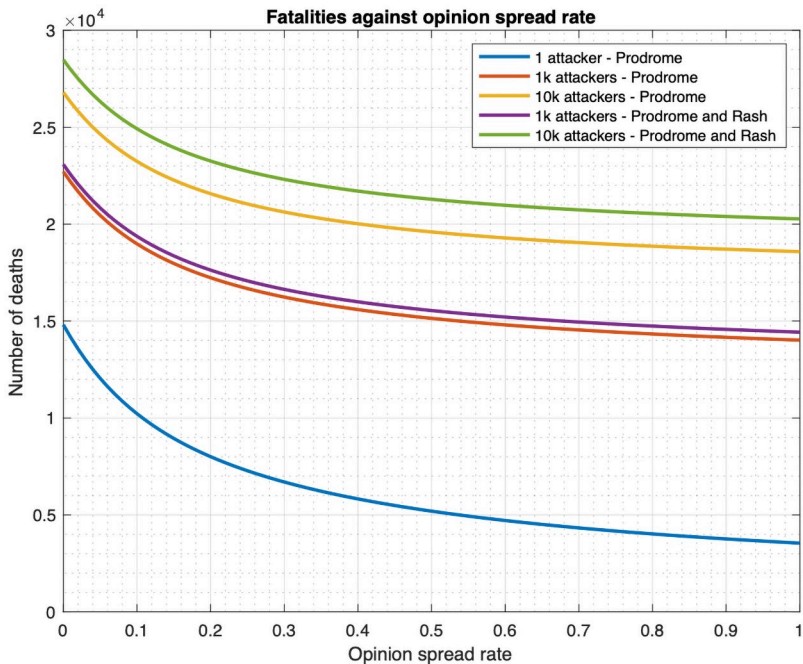

**Fig 7. Fatality versus opinion spread rate for different numbers and stages of initial infections in a single cluster (both drug and vaccine scenario).**

preemption rate as the product of the reciprocal of the expected total delay (in both parts above) and the probability above. Note that throughout the numerical computations we fix the disease progression parameters at their default values; we do not vary these as these constitute fundamental characteristics of the disease and data on various stages of smallpox are well-documented. We do however vary the preemption rates around the default values to assess the system under various operational conditions (Figs 10 and 11). We consider different supplies of drugs, the default setting however is one where the supply is unlimited. We consider three different policies for administering drugs, the default choice is policy 1 in which drugs are administered to anyone with fever or rash. We investigate other policies in Figs 17—19.

### 3.1 Opinion dynamics has a strong impact on fatality rates

Our numerical computations in this section show that opinion dynamics have a strong impact on the fatality for the cases in which vaccines are used as a countermeasure, namely the "vaccine only" and "both drug and vaccine" scenarios. We consider two cases separately: 1) cooperatives converting non-cooperatives (Figs 6 and 8) and 2) non-cooperatives converting cooperatives (Fig 9) during interactions. In the two cases above, the cooperatives (non-cooperatives, respectively) convert the non-cooperatives (cooperatives, respectively) during their interactions at the rate $\alpha$, thus, as $\alpha$ increases from 0 to 1, more (less, respectively) people become willing to receive vaccines, fatality in the "vaccine only" and "both drug and vaccine" scenarios decrease (increase, respectively).

We start with the case that cooperatives convert non-cooperatives during interactions. Fig 6 plots the fatalities in the single cluster case in all four scenarios as a function of $\alpha$ in the range [0, 1]. With 1000 (10000, respectively) individuals initially infected, fatality for "both drug and vaccine" scenario decrease by 38.3% (30.7%, respectively) as $\alpha$ increases from 0 to 1. The fatality for the "vaccine only" scenario respectively decreases by 32.2% and 28.2% in these two

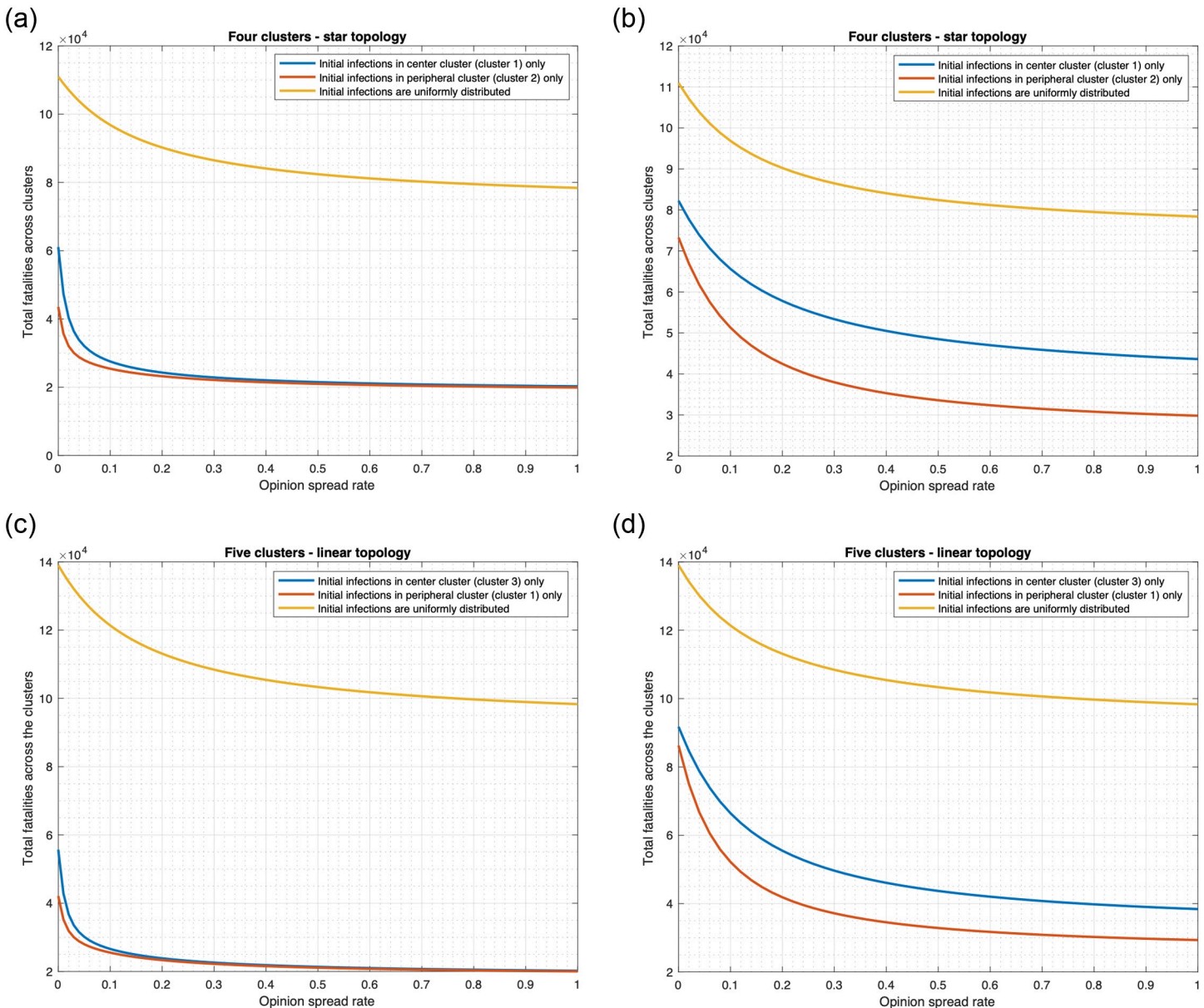

**Fig 8. Relationship between fatality and opinion spread rate (multiple clusters).** (a) Star topology, $\kappa = 0.01$. (b) Star topology, $\kappa = 0.8$. (c) Linear topology, $\kappa = 0.01$. (d) Linear topology, $\kappa = 0.8$.

cases. The fatality for no countermeasure and "drug only" scenarios does not change with any change in $\alpha$.

We now examine if deviation from our default assumption that all initially infected individuals are in prodrome stage alters any trend. In Fig 7 for "both drug and vaccine" scenario, we compare the fatalities when the initially infected individuals are all in prodrome stage with when they are equally distributed across the prodrome and early rash stages. As expected, the fatalities are somewhat higher in the latter case since there is less time to administer drugs before patients expire. Otherwise, the plots are similar in both cases. We also consider the lowest possible number of infected individuals, namely 1. The plot of the fatality as a function of $\alpha$ remains similar, except that it is shifted vertically down, that is, the fatality count is

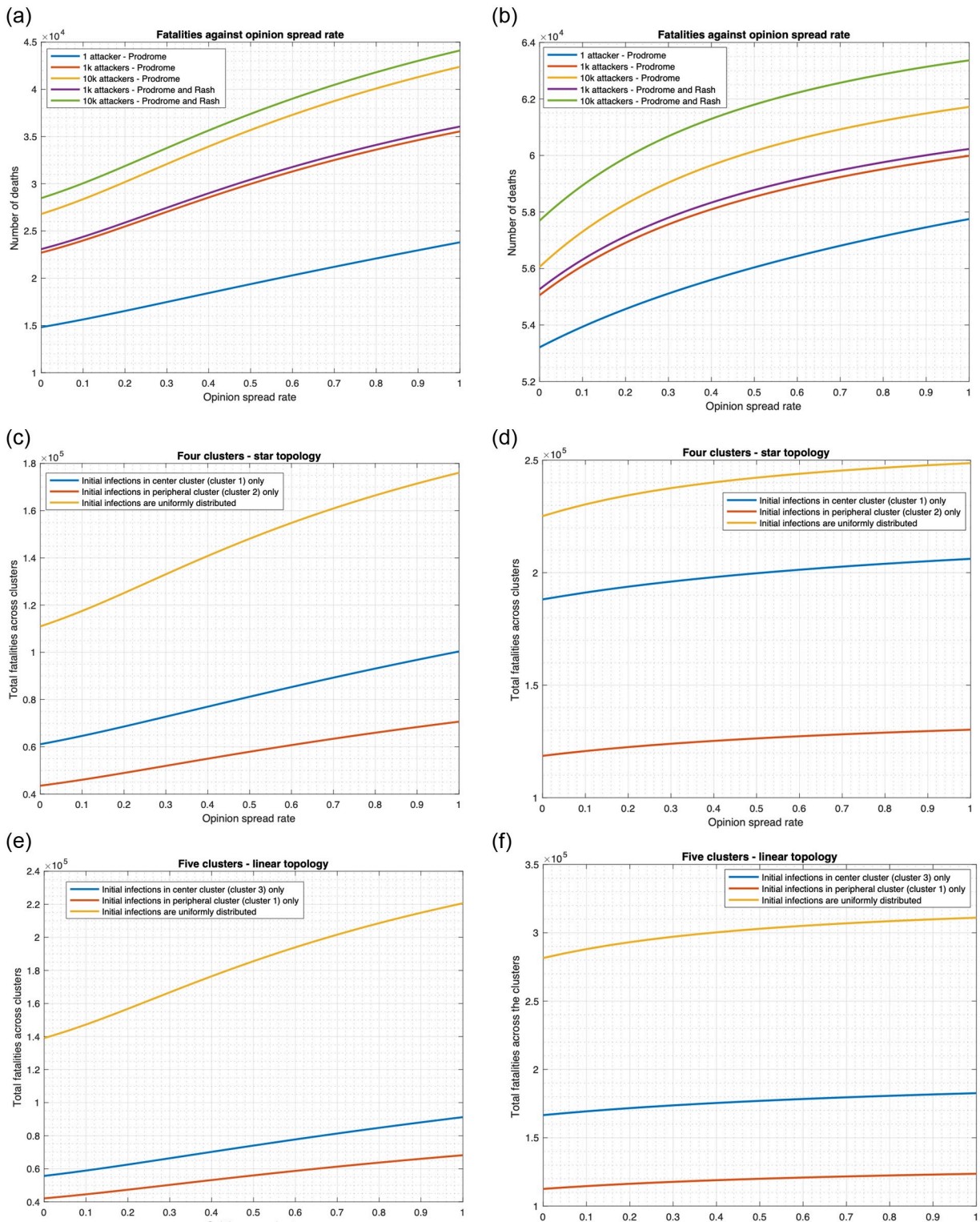

**Fig 9. Relationship between fatality and opinion spread rate when non-cooperatives convert cooperatives.** (a) Single cluster (initial cooperativity = 0.8). (b) Single cluster (initial cooperativity = 0.2). (c) Star topology (initial cooperativity = 0.8). (d) Star topology (initial cooperativity = 0.2). (e) Linear topology (initial cooperativity = 0.8). (f) Linear topology (initial cooperativity = 0.2).

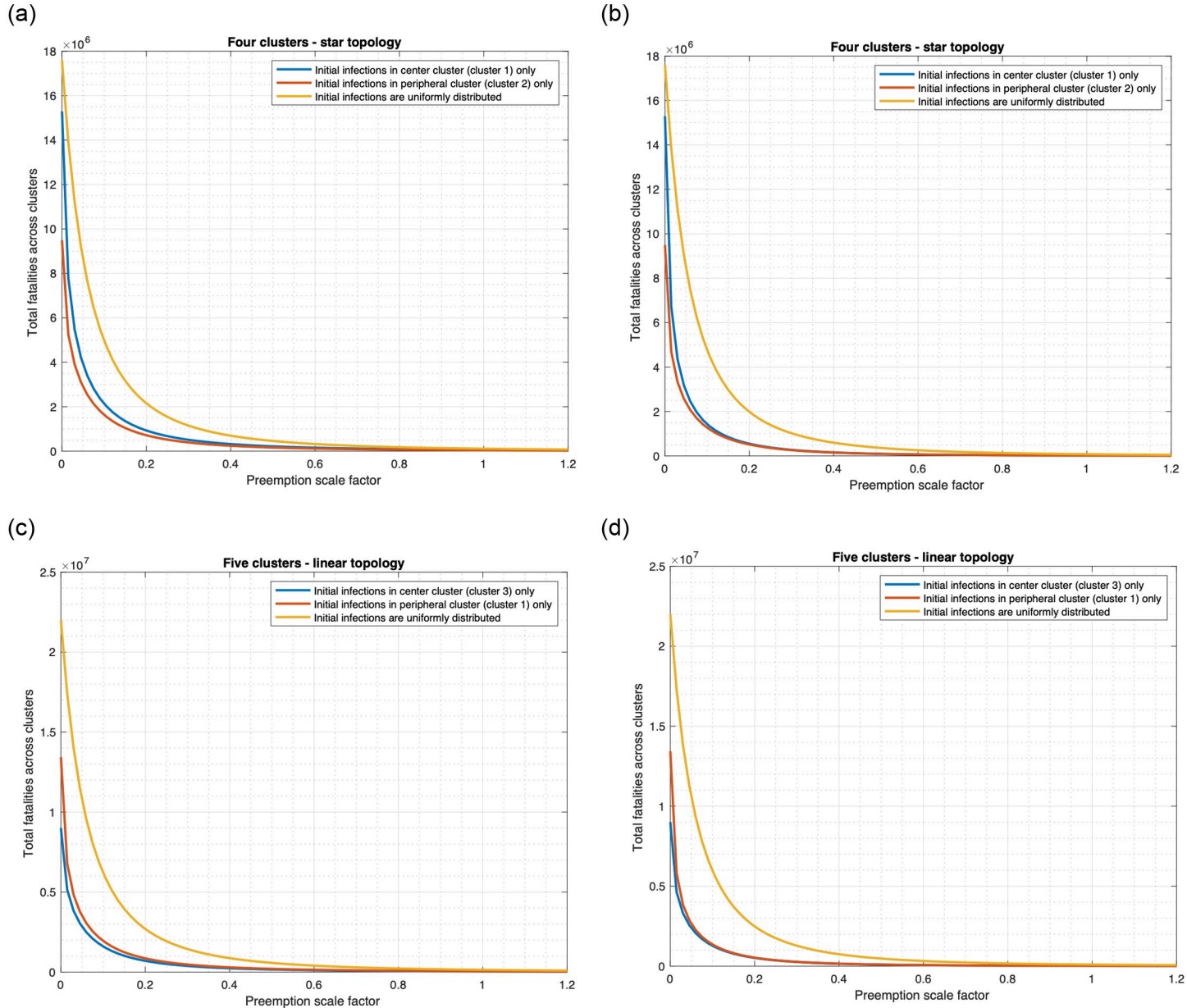

**Fig 10. Relationship between fatality and preemption scale factor.** (a) Star topology, $\alpha = 0.001$. (b) Star topology, $\alpha = 0.8$. (c) Linear topology, $\alpha = 0.001$. (d) Linear topology, $\alpha = 0.8$.

substantially lower. This is expected since the number of initially infected is also substantially lower. Note that the decrease in the fatality count with increase in $\alpha$ is substantially higher; as $\alpha$ increases from 0 to 1, fatality decreases by 76%. Recall that this decrease is 38.3% and 30.7% respectively when the initial number of infected individuals are 1000 and 10000 respectively. Thus, the exchange of opinions becomes less effective as the initial number of infected individuals increases.

Fig 8 plots the total fatality for "both drug and vaccine" scenario as a function of $\alpha$ in the range [0, 1] for multiple clusters. We consider the Uniform, Central and Peripheral distributions of initially infected individuals across the clusters. First consider the default mobility rate

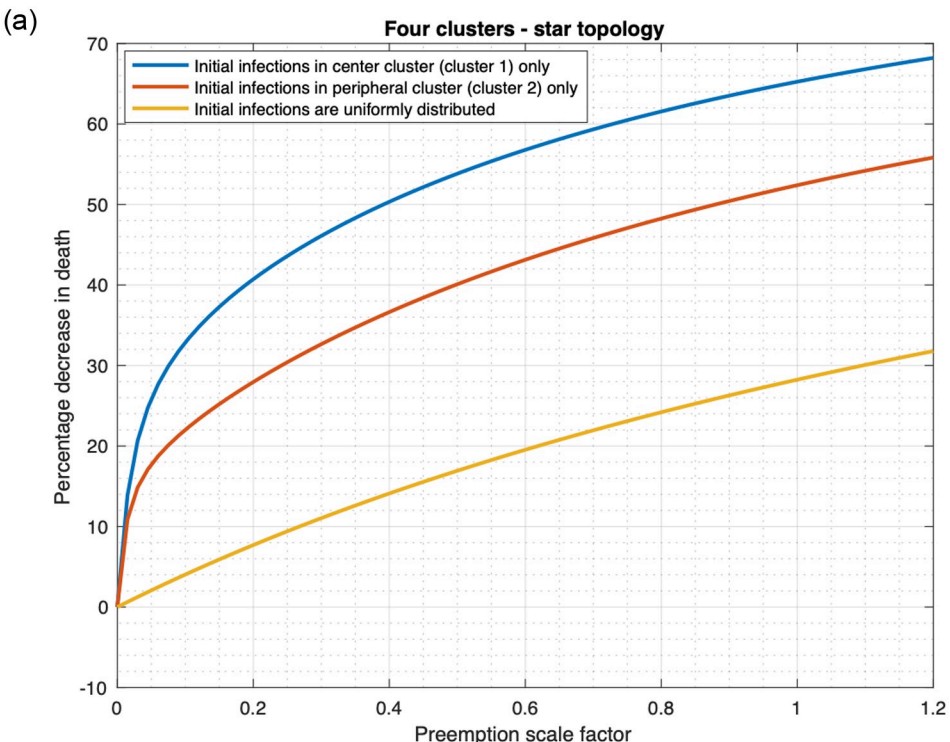

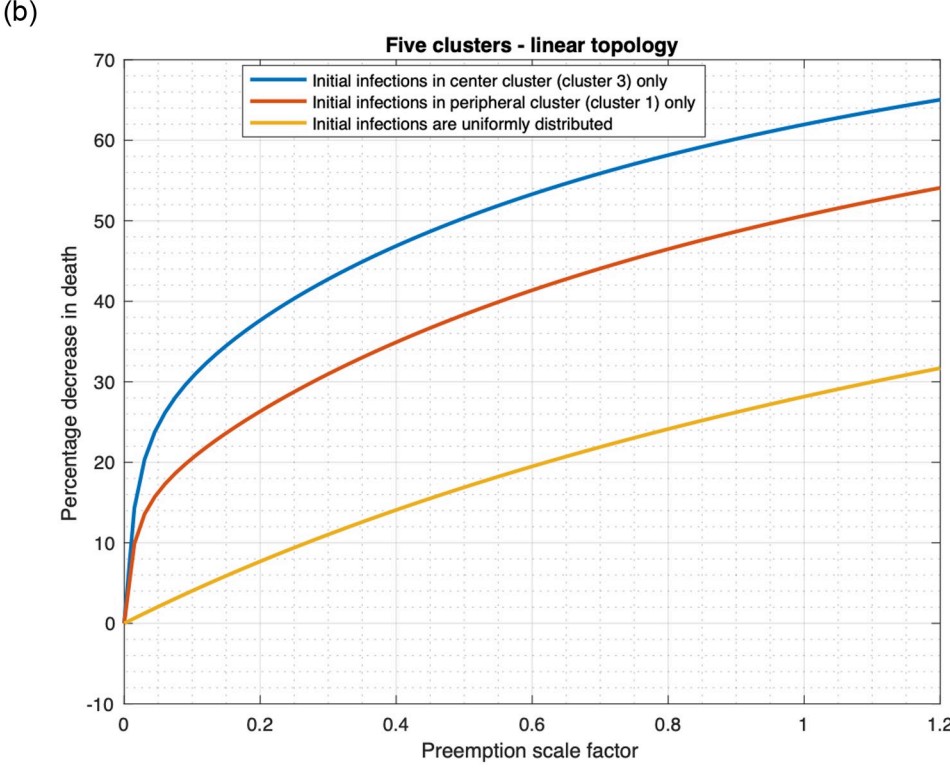

**Fig 11. Percentage decrease in fatality as a function of preemption scale factor.** a) Star topology. (b) Linear topology.

$\kappa$ = 0.01. As $\alpha$ increases from 0 to 1 in the star topology, total fatality decreases by 29.4%, 66.8%, and 54.2% respectively in the 3 scenarios (Fig 8a). For linear topology, the respective decrease percentages are 29.3%, 63.7%, and 52.4% (Fig 8c). When mobility rate increases, the fatality counts increase by similar amounts at each value of $\alpha$, that is, the plots move up vertically but otherwise the nature of the variation with respect to $\alpha$ remains same (see Fig 8b and 8d for $\kappa$ = 0.8). Specifically as $\alpha$ increases from 0 to 1 in the star topology, total fatality decreases by 29.4%, 64.9%, and 68.5% (Fig 8b) respectively for the 3 distributions. Similarly, for the linear topology, the respective decrease percentages are 29.3%, 71.9%, and 71.8% (Fig 8d). Therefore, the decrease in total fatality is considerable in all cases and comparable across the two topologies and for wide variation in mobility rates.

We now consider the case that non-cooperatives convert cooperatives during interactions. We report the results for the scenario in which both vaccines and drugs are administered for $\alpha$ in the range [0, 1] (Fig 9). We consider two different values of initial cooperativity in these: 0.8 (default value) and 0.2 respectively. As expected, with increase in $\alpha$, fatality increases throughout. For example, when the initial cooperativity is 0.8, Fig 9a shows that in a single cluster, as $\alpha$ increases from 0 to 1, fatality increases by 59.85%, 56.55% and 58.17% when there are 1, 1000 and 10000 initially infected individuals respectively and the initially infected individuals are in prodrome stage. When the initially infected individuals are distributed equally between prodrome and early rash stages, fatality count increases by 56.22% and 54.28% for an initial attack size of 1000 and 10000 respectively (when only 1 individual is initially infected, he can only be in 1 stage, we choose that as prodrome). Thus, yet again, the trends are similar for different number of initially infected individuals and distributed across different stages of the disease. We now consider multiple clusters and for brevity, report the results only for the default setting, namely 10000 initially infected individuals in the prodrome stage. As $\alpha$ increases from 0 to 1 in the star topology, fatality increases by 58.62%, 64.34%, and 62.32% respectively for Uniform, Central, Peripheral distributions of the initially infected individuals (Fig 9c). Similarly, as $\alpha$ increases from 0 to 1 in the linear topology, fatality increases by 58.65%, 63.83%, and 61.97% respectively for the 3 distributions (Fig 9e). The pattern of variation of the fatality count with increase in $\alpha$ remains similar at other values of initial cooperativity, only the increase is by lower amounts for lower values of initial cooperativity, as then the non-cooperatives have fewer cooperatives to convert. Considering a single cluster, Fig 9b shows that when initial cooperativity is 0.2 and initial infections originate from prodrome, as $\alpha$ increases from 0 to 1, fatality increases by 8.53%, 8.96% and 10.10% when initial number of infected individuals are 1, 1000 and 10000 respectively. When the initially infected individuals are distributed equally between prodrome and rash stages, fatality count increases by 8.98% and 9.83% for an initial attack size of 1000 and 10000 respectively. As $\alpha$ increases from 0 to 1 in the star topology, fatality increases by 10.45%, 9.57%, and 9.79% respectively for Uniform, Central and Peripheral distributions of the initially infected individuals (Fig 9d). Similarly, as $\alpha$ increases from 0 to 1 in the linear topology, fatality increases by 10.47%, 9.63%, and 9.83% respectively for the 3 distributions (Fig 9f).

In summary, our numerical computations in this section reveal that the overall fatality count sharply decreases (increases, respectively) with increase in rate of spread of opinion, $\alpha$, between cooperatives and noncooperatives when the former (latter, respectively) converts the latter (former, respectively). Our finding holds for a large range of combination of parameters, namely different geographical topology of target region (number of clusters, organization of clusters), combinations of countermeasures, number and distribution of initially infected individuals, stage of disease of the initially infected individuals, mobility rates, initial cooperativities. With change in $\alpha$, the fatality count often changes by more than 50%; the change is smaller if there are fewer individuals available to convert at the initial time. Also, the pattern of

the variation of fatality count with respect to $\alpha$ is similar for different choices of the above listed parameters.

## 3.2 Distribution of initial infections, initial cooperatives, and mobility of individuals have a significant impact

The outbreak of an infectious disease may start from a single cluster or multiple clusters. It is of interest to find out if fatality is higher in one of these scenarios and if the difference is significant. This would help identify how the attacks might have been designed. We also seek to understand how enhanced mobility of individuals affects fatalities. The cooperative individuals help convert non-cooperative individuals. We, therefore, seek to understand how the initial distribution of the cooperatives affects the fatalities.

We consider that both drug and vaccine are used and plot the total fatality count across linear and star topologies as a function of the preemption rate. The basic preemption rates in different states have been calculated in Appendix B in S1 Appendix. We consider a preemption scale factor that multiplies all these rates, that is, scales the rates, and we plot the fatality count as a function of this scale factor (Fig 10).

For each topology we consider the following distribution of the initially infected individuals: 1) Uniform 2) Central 3) Peripheral.

Fig 10 reveals the following. Other things being equal, the fatalities are much higher for Uniform than for Central and Peripheral. Note that all the initial infections originate from one cluster in the latter two. For instance, considering $\alpha = 0.001$, preemption rate as 1, the fatality for Uniform is 1.87 and 2.61 times that for Central and Peripheral in the star topology (see Fig 10a). For linear topology, the corresponding ratios are 2.49 and 3.30 (Fig 10c). Also, the fatalities are higher for Central than for Peripheral in the case that the initial infections occur in one cluster. For instance, at 0 preemption rate in the linear topology, the fatality for Central is 1.49 times that for Peripheral (see Fig 10c and 10d).

Fig 10 also shows that increasing the preemption rate substantially decreases the fatality regardless of the distribution of the initial infection. The preemption rate may be increased by decreasing the delay in administering the vaccine and delivering the drug to the individuals which may be accomplished by increasing the number of health workers and the rate of delivery of the vaccine and drugs to the health facilities.

Next, Fig 10 also shows that the variation of the fatality count vis-à-vis the preemption scale factor remains similar for different values of $\alpha$, though the fatality counts decrease with increase in $\alpha$ for all positive preemption scale factors and distribution of the initially infected. The decline in fatality count with increase in $\alpha$ however increases with increase in the preemption scale factor (Fig 11). Specifically, in Fig 11 at each value of preemption scale factor, we plot the decrease in fatality count when $\alpha$ increases from 0.001 to 0.8 as a percentage of the fatality count at $\alpha = 0.001$. This quantity increases substantially with increase in the preemption scale factor, because greater amount of conversion of noncooperatives due to greater frequency of information exchange (that is higher $\alpha$) is more effective at higher values of preemption rate (equivalently, scale factor). The latter happens because at low preemption rates even when a larger fraction of the population is willing to receive vaccine they can not readily receive vaccines because of low delivery rates. Note that the plots for Uniform, Central and Peripheral are parallel to each other for both topologies, thus the nature of the variation is similar in the three cases; the decline is maximum for Central and least for Uniform. Thus, conversion of noncooperatives is least effective for Uniform distribution of initially infected individuals.

We now plot the fatality count as a function of the mobility rates in the star and linear topologies in Fig 12a and 12b.

We observe the following. If the initially infected individuals are uniformly distributed across the clusters, then fatality count does not change with the mobility rate, assuming that the mobility rates between two clusters are the same in both directions. If the initial infections occur only in one cluster, fatality substantially increases with the mobility rate. For instance, as the mobility rate increases from 0 to 1 in the star topology (Fig 12a), the total fatality increase by 202.6% (from 27, 807 to 84, 136) if the initially infected individuals originate from the central cluster only. Similarly, the total fatality increases by 166.5% (from 27, 807 to 74, 112) when the initially infected individuals originate from a peripheral cluster. These observations may be explained as follows. Mobility helps individuals in different clusters mix with each other. Thus, if the initially infected individuals are concentrated in a few clusters, higher mobility rates help spread the disease faster into other clusters and lead to high fatalities. This is the scientific basis for reducing travel links (land or air) between different regions when the infection is localized. But if the initially infected individuals is spread out uniformly, even without any mobility, the disease spreads in each region from the individuals initially infected. Thus mobility enhances the spread only marginally and thereby increases the fatality only marginally. Thus spatial quarantining strategies do not help much if the initially infected individuals is uniformly spread. Thus, the uniform spread of initially infected individuals constitute the most deadly form of a deliberate attack.

We also note that regardless of the mobility rate, fatality is significantly higher when the initially infected individuals are uniformly spread as compared to when the initially infected individuals are concentrated in a few clusters. For instance, when the mobility rate is 0, the total fatality when initially infected individuals are uniformly distributed across the linear topology would be 11.4 times the fatality that occurs if infections originate from one cluster (Fig 12b). When the mobility rate becomes 1, the total fatality for the uniformly distributed initially infected individuals would be 1.9 times the fatality that occurs if initial infections occur only in cluster 1.

We now investigate the impact of different distributions of initial cooperatives. We consider two cases: 1) uniform distribution across all clusters 2) concentration in only one cluster, the central cluster. Since each cluster has 10 million individuals, in the second case we can have at most 10 million initial cooperatives, that is initial cooperativity is at most 0.25. We consider cooperativity as this maximum value in the second case, and for consistent comparison we consider the same in the first case too. We plot the fatalities for these two cases and three different distributions of the initially infected, namely Uniform, Central, Peripheral in Fig 13. Note that the terminologies Uniform, Central, Peripheral refer to the distribution of the initially infected individuals which may be different from the distribution of the initial cooperatives. Thus we can have initial cooperatives uniformly distributed across clusters and the initially infected individuals confined to the central cluster (i.e., Central). Fig 13 reveals that the fatality count is much higher when the initial cooperatives are concentrated in one cluster, e.g., under Uniform, the fatality in this case is 1.5 times that when the initial cooperatives are uniformly distributed across clusters. As noted before, when the initial cooperatives are uniformly distributed, the fatality under Uniform far exceeds that under Central and Peripheral, and the fatality under Peripheral is the least of the three. This observation largely extends to when the initial cooperatives are concentrated in the central cluster, except when the mobility rate is very low in which case, the fatality under Peripheral exceeds that under Central. In this case, the individuals in different clusters have little or no physical or virtual contact (since we assume that these contacts can not take place across clusters). Thus, in the Central and Peripheral cases, the infected individuals are largely confined to the central and the specific peripheral

(a)

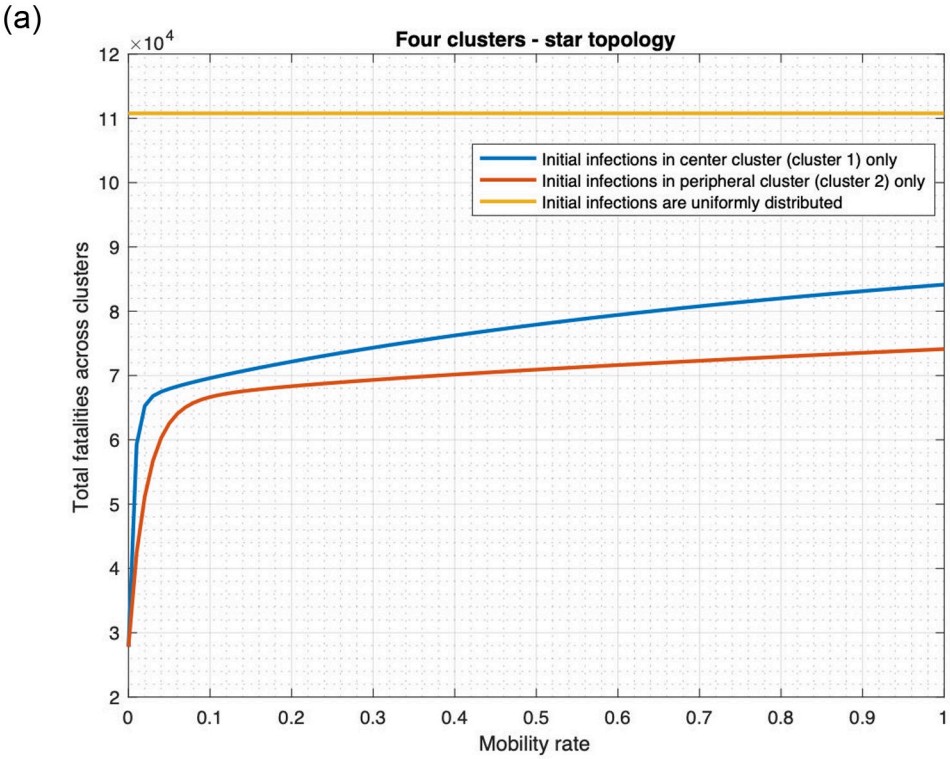

(b)

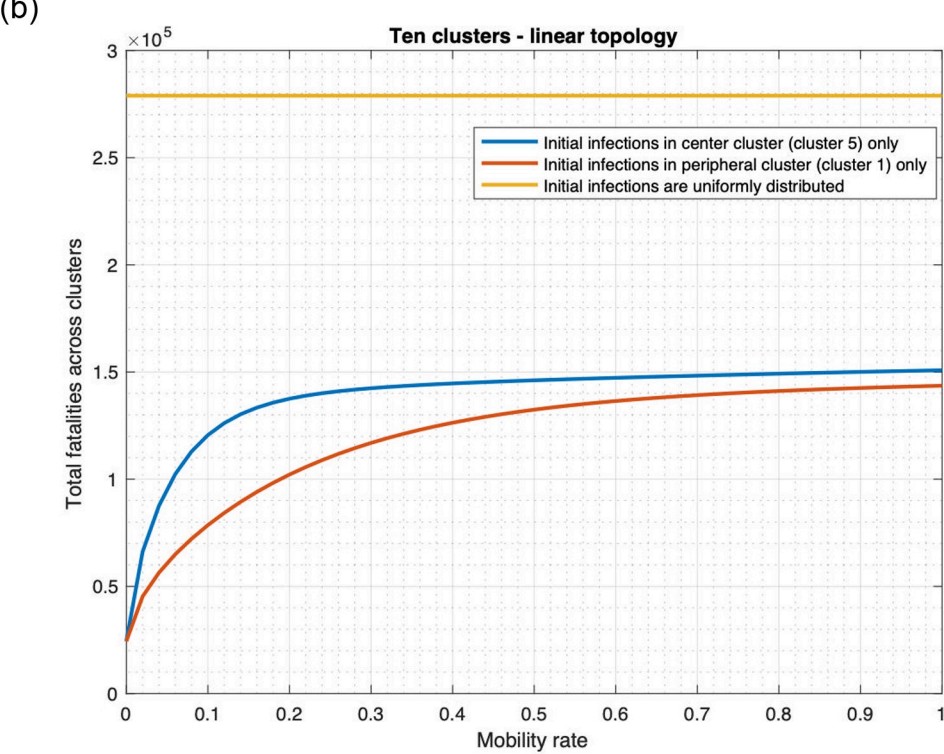

**Fig 12. Relationship between fatality and mobility rate.** (a) Star topology. (b) Linear topology.

(a)

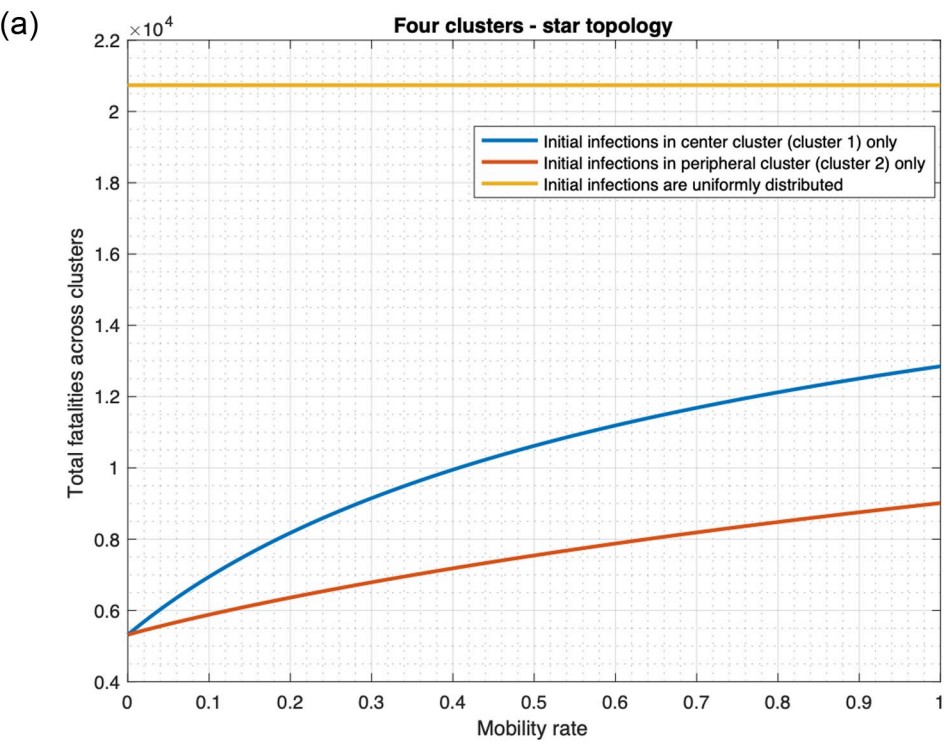

(b)

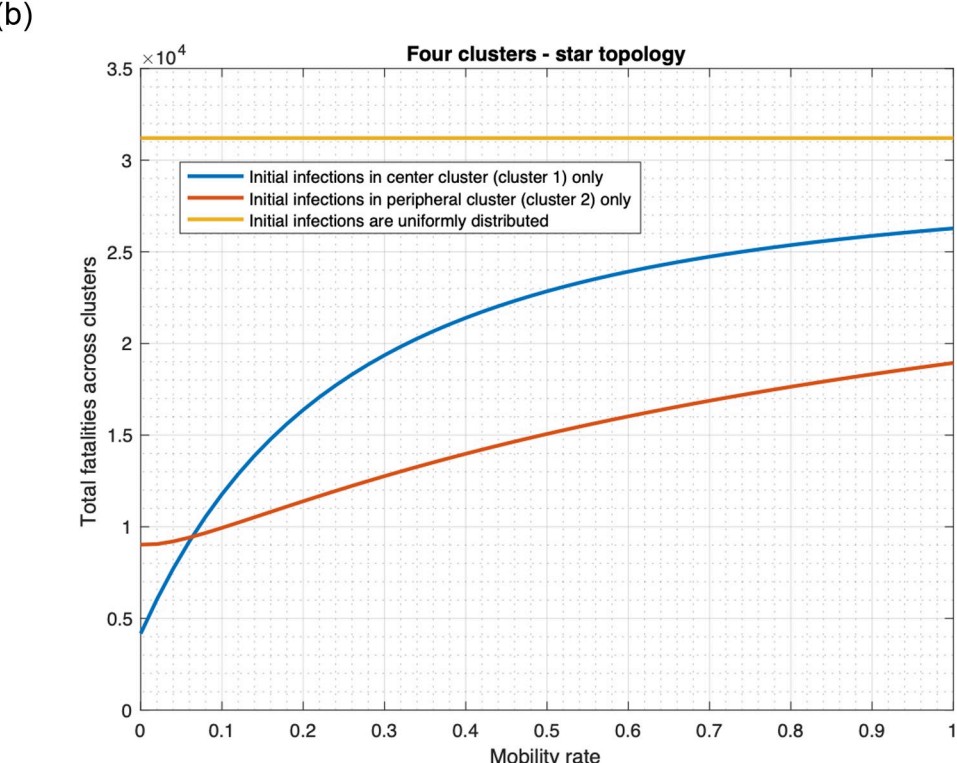

**Fig 13. Relationship between fatality and initial distribution of cooperatives.** (a) Initial cooperatives are uniformly distributed across clusters (initial cooperativity = 0.25). (b) Initial cooperatives are only in the central cluster (initial cooperativity = 0.25).

clusters (one that had the initially infected). The cooperative individuals are however largely confined to the central cluster in both these cases. In the first case, the spread of infection in the central cluster reduces quickly because the cooperatives get vaccinated (all individuals in the central cluster are cooperatives to start with). Thus fatality remains low. In the second case, the infection spreads extensively in the peripheral cluster because very few individuals become cooperatives and therefore very few receive vaccines. This leads to high fatality count in the specific peripheral cluster which increases the overall fatality count. This phenomenon is a result of localization of both the cooperativity and the disease, and subsides as mobility rate increases. As mobility rate increases, the rank order between the fatality counts becomes the same as that when the initial cooperatives are uniformly distributed. Finally, under Uniform, for both distributions of the initial cooperatives, fatality does not change with change in mobility rate. Thus, distribution of the initially infected largely determines the pattern of variation of fatality count with mobility.

In summary, our numerical computation in this section reveal that other things being equal, fatalities are often significantly higher, when 1) the initial infections are uniformly distributed across various clusters than when all the initial infections only occur in one cluster; 2) the outbreak originates from the central cluster(s) than when it originates from the peripheral cluster(s). Even higher rate of exchange of opinions respectively reduces the fatalities 1) most and 2) least when initially infected individuals are 1) all in the central cluster and 2) distributed uniformly. The fatalities increase with increase in mobility rate when the initial infections occur in one cluster, but do not depend on the mobility rate when the initial infections are uniformly distributed. These findings holds for a large number of choices of other parameters, as applicable, namely, number and organization of clusters, mobility rates, preemption rates, opinion exchange rates, distribution of initial cooperatives, etc.

## 3.3 Administering drugs substantially reduces fatality

We show that using drugs as a countermeasure against smallpox substantially reduces fatality for various choices of parameter values, considering both unlimited (Figs 14 and 15) and limited (Fig 16) availability of drugs.

**3.3.1 Infinite drug availability.** First, Fig 14a plots fatality as a function of disease spread rate, $\phi$, for the four scenarios for one cluster. Visually, we note that for the entire range of $\phi$ considered, the scenarios that involve administering of the drug, namely the "drug only" and "both" scenarios, have much lower fatality than those that do not involve administering of drugs, namely the "vaccine only" and "no countermeasure" scenarios. Specifically, at the default value of $\phi$, namely $\phi = 0.00173$, the fatality for the "vaccine only" and "no countermeasure" scenarios are respectively 17.5 and 66.9 times that of the "drug only" scenario. Also, fatality for the "vaccine only" and "no countermeasure" scenarios are respectively 67.6 and 257.8 times that of the "both drug and vaccine" scenario. And fatality of the "drug only" scenario is 3.8 times that for the "both" scenario. Thus, deploying both countermeasures substantially reduce fatality, but if only one countermeasure is to be deployed, the drug is a better option in terms of fatality. This is because the non-cooperatives consent to the reception of the drug but not to the vaccine and the immunodeficient ones can only receive vaccines that act very slowly.

In addition, Fig 14b plots fatality as a function of the initial cooperativity. The conclusions with respect to efficacy of the drug remain the same as in the previous paragraph. For example, when the initial cooperativity is 0.8, then the fatality for the "vaccine only" and "no countermeasure" scenarios are respectively 27.3 and 66.9 times that of the "drug only" scenario. Also, fatality for the "vaccine only" and "no countermeasure" scenarios are respectively 67.3 and

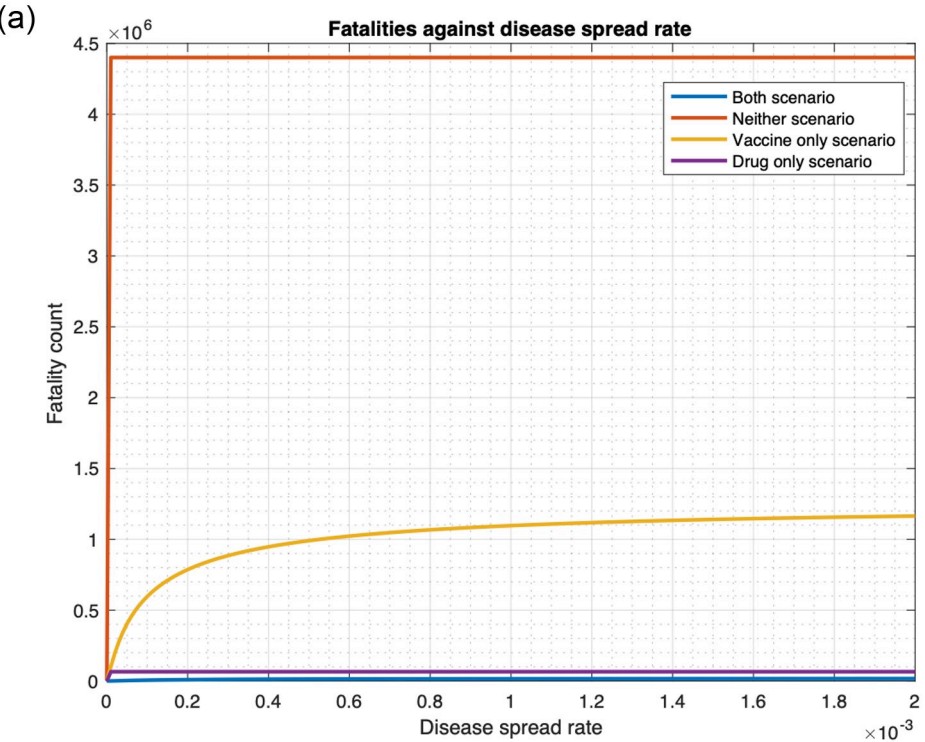

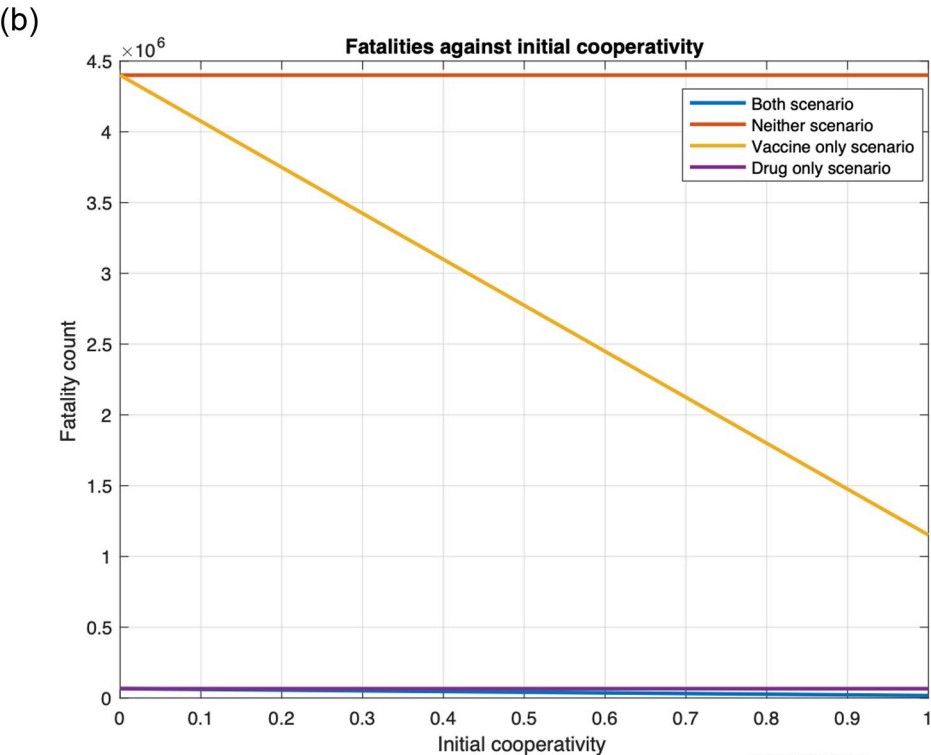

**Fig 14. Fatalities under four countermeasure application scenarios for various values of infection spread rates and initial cooperativities for single cluster.** (a) Relationship between fatality and disease spread rate. (b) Relationship between fatality and initial cooperativity.

(a)

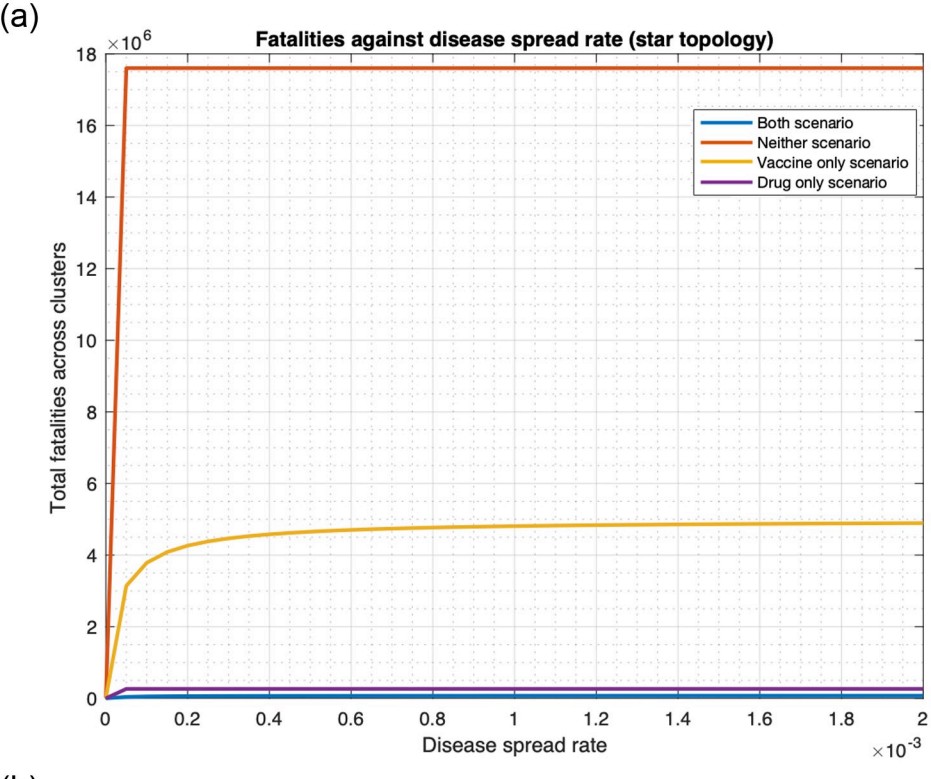

(b)

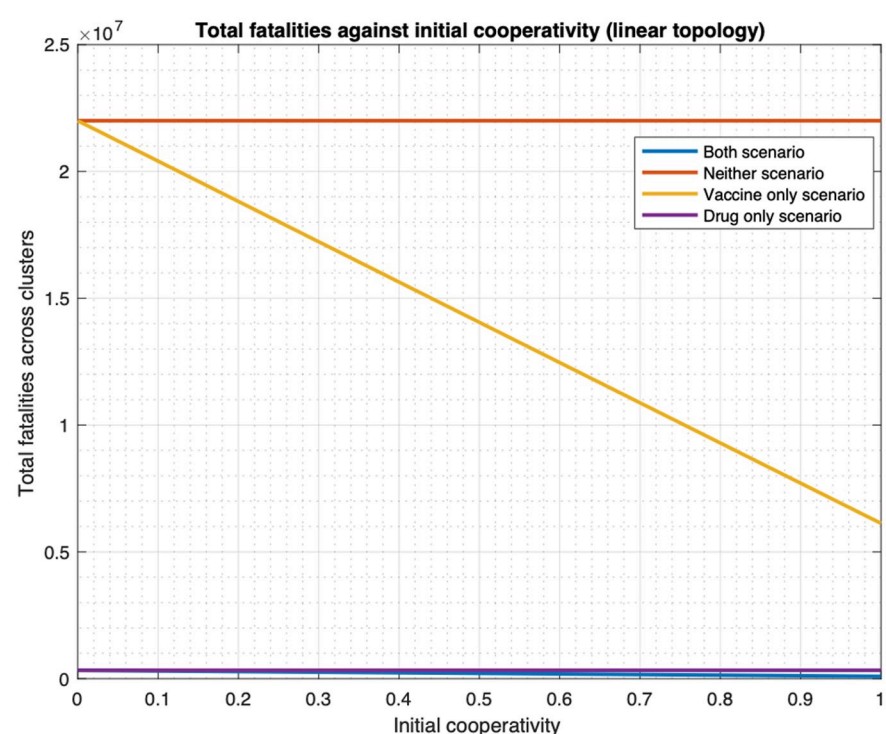

**Fig 15. Fatalities under four countermeasure application scenarios for various values of infection spread rates and initial cooperativities for multiple clusters.** (a) Relationship between total fatality and disease spread rate (star topology). (b) Relationship between total fatality and initial cooperativity (linear topology).

(a)

(b)

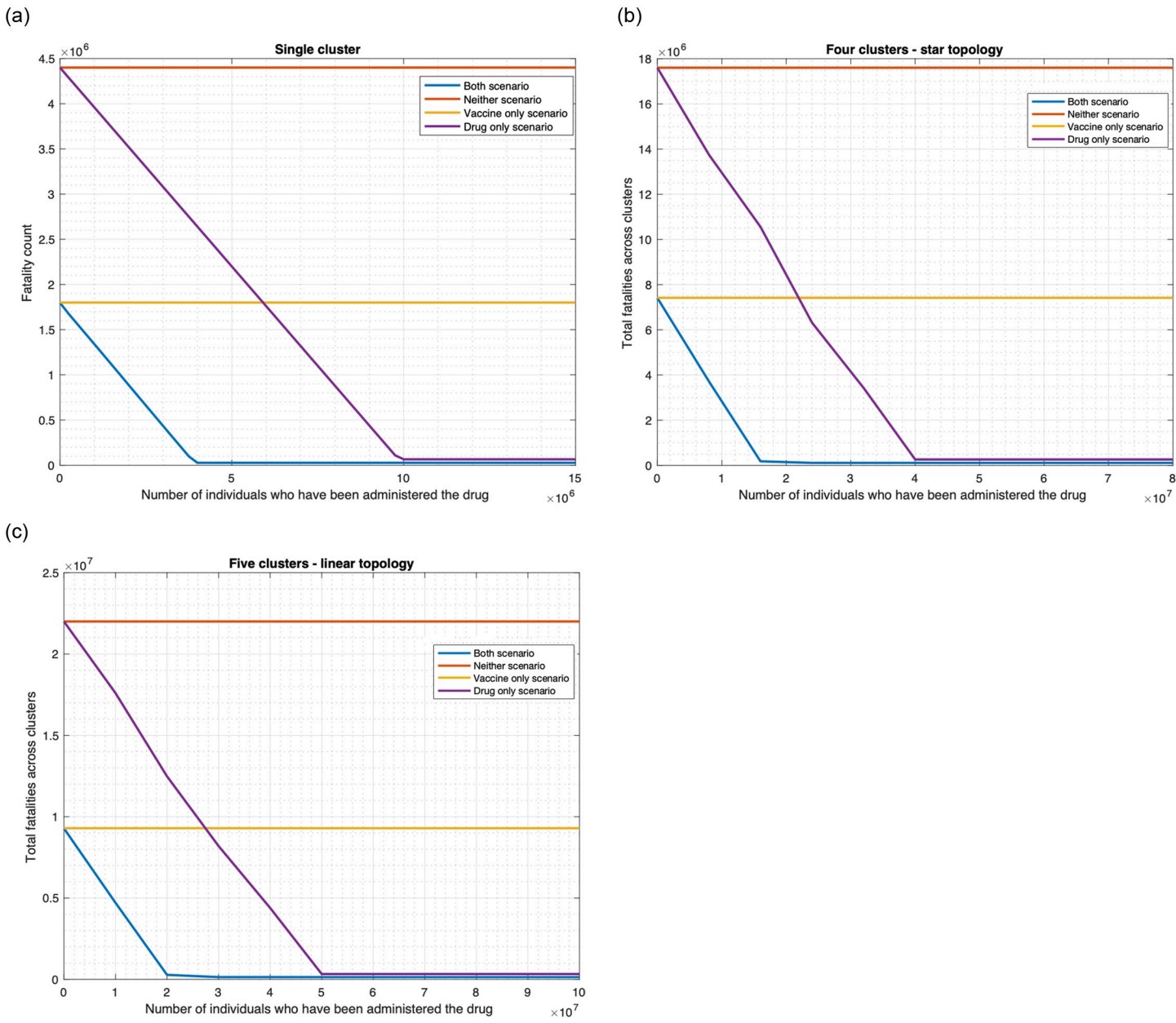

(c)

**Fig 16. Relationship between fatalities and antiviral drug availability.** (a) Single cluster. (b) Star topology. (c) Linear topology.

164.5 times that of the "both drug and vaccine" scenario. And fatality of the "drug only" scenario is 2.5 times that for the "both" scenario.

We now consider the multiple cluster case. Fig 15a plots total fatality as a function of disease spread while Fig 15b plots total fatality as a function of the initial cooperativity. The patterns are similar to what we reported in the previous paragraphs. For example, considering the default value of $\phi$, for the star topology, from Fig 15a, the fatality for the "vaccine only" and "no countermeasure" scenarios are respectively 18.5 and 66.86 times that of the "drug only" scenario. Also, fatality for the "vaccine only" and "no countermeasure" scenarios are respectively 66.91 and 241.28 times that of the "both drug and vaccine" scenario. And fatality of the

"drug only" scenario is 3.61 times that for the "both" scenario. From Fig 15b, for linear topology and for the initial cooperativity of 0.8, the total fatality for the "vaccine only" and "no countermeasure" scenarios are respectively 28.24 and 66.86 times that of the "drug only" scenario. Also, from Fig 15b, total fatality for the "vaccine only" and "no countermeasure" scenarios are respectively 66.95 and 158.51 times that of the "both drug and vaccine" scenario. And fatality of the "drug only" scenario is 2.37 times that for the "both" scenario.

**3.3.2 Finite drug availability.** We now consider that the drug availability is finite and drug is no longer administered when the supply is exhausted. We assume that the first time a patient is administered the drug, he is handed out enough quantity of the drug for his daily consumption throughout the period of the outbreak. Since the period of the outbreak can vary and may depend on re-introduction of the infection from outside, we measure the drug supply in terms of the number of patients who have been administered the drug. Fig 16 plots the fatality as a function of the drug supply as measured above, and reveals that the fatality for the scenarios in which drug is administered, namely the "drug only" and "both drug and vaccine" decrease approximately linearly with increase in drug supply, and remains constant once the supply crosses a threshold. Above the threshold value there is enough drug to administer to the population who need drugs. For the single cluster case, the fatality in the "drug only" case decreases from 4.4 million to 65, 808 as the drug supply increases from 0 to approximately 10 million, 10 million constitutes the threshold value. Note that the population size is 10 million here, thus the fatality does not decrease any further once the drug supply is enough to preempt the entire population. The fatality in the "both drug and vaccine" scenario decreases from 1.8 million to 26, 752 (98.514%, see Fig 16a) as the drug supply increases from 0 to approximately 4 million, which constitutes the threshold value. Since individuals can also be preempted through vaccines, one only needs enough drug supply to preempt 4 million individuals to decrease the fatality count to its minimum possible value. Table 1a shows that the decrease of the fatality with increase in supply is approximately linear for the "both drug and vaccine" scenario. For supply less than or equal to 3.75 million, the magnitude of the deviation of an exact straight line from the fatality plot is less than 0.5%, and is 0.41% on an average. The deviation is higher though in a subsequent interval of relatively small size 3.75–4 million, leading up to the threshold value of 4 million; the maximum deviation in this range is however still less than 13.41% in absolute value.

Similarly, for the four cluster star topology, the fatality in the "drug only scenario" decreases approximately linearly from 17.6 million to 263, 234 (98.504%, see Fig 16b) as the drug supply increases from 0 to approximately 40 million, which constitutes the threshold value. The fatality in the "both drug and vaccine" scenario decreases approximately linearly from 7.58 million to 113, 305 (98.505%, see Fig 16b) as the drug supply increases from 0 to approximately 16.4 million, which constitutes the threshold value. In addition, for the five cluster linear topology, the fatality in "drug only" scenario decreases approximately linearly from 22 million to 329, 042 (98.504%, see Fig 16c) as the drug supply increases from 0 to approximately 50 million, which constitutes the threshold value. The fatality in "both drug and vaccine scenario" decreases approximately linearly from 9.3 million to 138, 821 (98.507%, see Fig 16c) as the drug supply increases from 0 to approximately 20 million, which constitutes the threshold value. Note that for the star topology, the threshold value for "both drug and vaccine" is approximately 4 times that of the corresponding single cluster since there are 4 clusters. This is because the total population is 4 times the population for the corresponding single cluster. Similarly, for the linear topology, the threshold is exactly five times that for single cluster since the population size is 5 times as the number of clusters. Again, the threshold value is less than that in the "drug only" scenario for the same reason as for single cluster. Table 1b and 1c establish the approximate linear nature of the decrease of the fatality with increase in supply for the

**Table 1. The three tables respectively compare the actual fatality count and linear interpolations for the "both drug and vaccine scenario" for 1) single cluster (Fig 16a), 2) star topology (Fig 16b), and 3) linear topology (Fig 16c).** In these tables, the first column (A) represents the number of individuals in millions, who have been administered the drugs, the second column (B) provides the actual fatality counts, the third column (C) provides the fatality counts of a linear interpolation (i.e., straight line with the entries marked in bold as end-points), the fourth column provides the percentage errors, i.e., $100 \times (B - C)/B$.

| 1 | | | |
|---|---|---|---|
| A | B | C | Error (%) |
| **0** | **1797843** | 1797843 | 0.0000 |
| 0.5 | 1564306 | 1571018 | -0.4291 |
| 1 | 1340137 | 1344193 | -0.3027 |
| 1.5 | 1114319 | 1117368 | -0.2736 |
| 2 | 887008 | 890543 | -0.3985 |
| 2.5 | 660891 | 663718 | -0.4278 |
| 3 | 434887 | 436893 | -0.4613 |
| **3.5** | **210066** | 210068 | -0.0010 |
| 3.75 | 98147 | 96653 | 1.5222 |

| 2 | | | |
|---|---|---|---|
| A | B | C | Error (%) |
| **0** | **7409854** | 7409854 | 0.0000 |
| 1.75 | 6628341 | 6621237 | 0.1072 |
| 3.5 | 5843703 | 5832621 | 0.1896 |
| 5.25 | 5055937 | 5044004 | 0.2360 |
| 7 | 4265097 | 4255388 | 0.2276 |
| 8.75 | 3472994 | 3466771 | 0.1792 |
| 10.5 | 2681065 | 2678155 | 0.1085 |
| **12.25** | **1889538** | 1889538 | 0.0000 |
| 16 | 207160 | 199645 | 3.6276 |

| 3 | | | |
|---|---|---|---|
| A | B | C | Error (%) |
| **0** | **9132247** | 9132247 | 0.0000 |
| 2.5 | 8021838 | 8008540 | 0.1658 |
| 5 | 6903523 | 6884833 | 0.2707 |
| 7.5 | 5778566 | 5761125 | 0.3018 |
| 10 | 4649803 | 4637418 | 0.2664 |
| 12.5 | 3518485 | 3513711 | 0.1357 |
| 15 | 2388013 | 2390004 | -0.0834 |
| 17.5 | 1260262 | 1266296 | -0.4788 |
| **20** | **142589** | 142589 | 0.0000 |

(a) Single cluster. Average of the absolute value of the error in the above range is 0.4072%. The maximum of the absolute value of error in 3.75–4 million range is 13.4121%, and is registered at 3.912 million.
(b) Four cluster star topology. Average of the absolute value of the error in the above range is 0.9648%. The maximum of the absolute value of the error in 16–16.4 million range is 9.2907%, and is registered at 16.22 million.
(c) Five cluster linear topology. Average of the absolute value of error in this range is 0.2128%.

"both drug and vaccine" scenario in star and linear cluster topologies respectively. For the linear topology, up to the threshold value of 20 million, the magnitude of the deviation of an exact straight line from the fatality plot is less than 0.5% (Table 1c). The deviation is somewhat higher for the star topology, but is still in acceptable range. For supply less than or equal to 16 million, the magnitude of the deviation of an exact straight line from the fatality plot is 0.96%

on an average and 3.63% at worst (Table 1b). In a subsequent interval of relatively small size, 16–16.4 million, leading up to the threshold value of 16.4 million, the deviation is higher, but is still less than 10% in absolute value.

The fatalities for the scenarios not involving drugs naturally do not change with increase in the availability of the drug. Also, note that when the drug supply is zero, i.e., no drug is available then the fatality in the "drug only" scenario equals that for the no countermeasure scenario, and the fatality in "both drug and vaccine" scenario equals that for the "vaccine only" scenario. For low value of the drug supply, the "vaccine only" scenario has lower fatality than the "drug only" scenario, but as the drug supply increases, the fatality for the "drug only" scenario falls below that of the "vaccine only" scenario. Thus, if the drug supply is low and if one needs to choose between the countermeasures, vaccine is better with respect to fatality count. And, high values of the drug supply substantially reduce the fatality count.

We had only considered the default Uniform distribution of the initially infected individuals in Fig 16 and Table 1. We now consider Central and Peripheral distributions, as also star and linear topologies with different number of clusters, namely 8 and 5 clusters. Figs 17a, 17b, 18a and 18b, and the associated Table 2 reveal that for the default policy, namely policy 1 which was being considered above (under which all people with fever and rash receive drugs), overall, the fatality decreases approximately linearly with increase in supply until a threshold value for a wide range of topologies and distributions of the initially infected individuals. Specifically, for Table 2d corresponding to Fig 18b, the magnitude of the deviation of the fatality plot from a straight line is small right up to the threshold value (20.01 million). For Table 2a and 2b, corresponding respectively to Fig 17a and 17b, the magnitude of the deviation is small except in a small region leading to the respective threshold values (i.e. 10.1, 25.5 million) in which the deviation is considerable. Only for Table 2c which corresponds to Fig 18a, the deviation is also somewhat greater for small values of supply. But, even considering this region, the average deviation remains small in this case. We will discuss the plots for policies 2 and 3 in Section 3.5.

In summary, considering the default policy for administering drugs, namely policy 1, our numerical computations show that the most effective countermeasure consists of a combination of antiviral drugs and vaccines, but if only one countermeasure can be administered it ought to be the former. Again, the fatality counts under different countermeasures are significantly different, in particular, administering drugs substantially reduces the fatality count. We also find that the fatality counts decrease approximately linearly with increase in supply of drugs until the supply is enough to administer to everyone who develops symptoms (fever and rash), and beyond that point these counts do not change with increase in the supply. Our findings hold for different disease spread rates, initial cooperativities, numbers and organization of clusters, distributions of the initially infected individuals and amount of availability of drugs.

## 3.4 Impact of topology

We now compare the fatality counts across topologies and identify patterns that emerge. Towards this end, we consider the fatalities in star and linear topologies reported in Figs 8–12 and 16. We compare the per cluster fatality (total fatality count normalized by the number of clusters) of the star and linear topologies for representative data points in the figures and report in Table 3. In this table, we compare the fatalities for three distributions of initially infected individuals across the clusters: Uniform, Central, Peripheral. To distinguish between the cases that cooperatives convert non-cooperatives and non-cooperatives convert cooperatives, we add suffixes to the terms denoting the three distributions, that is, we denote the first case by UniformC, CentralC, and PeripheralC, and the second by UniformNC, CentralNC,

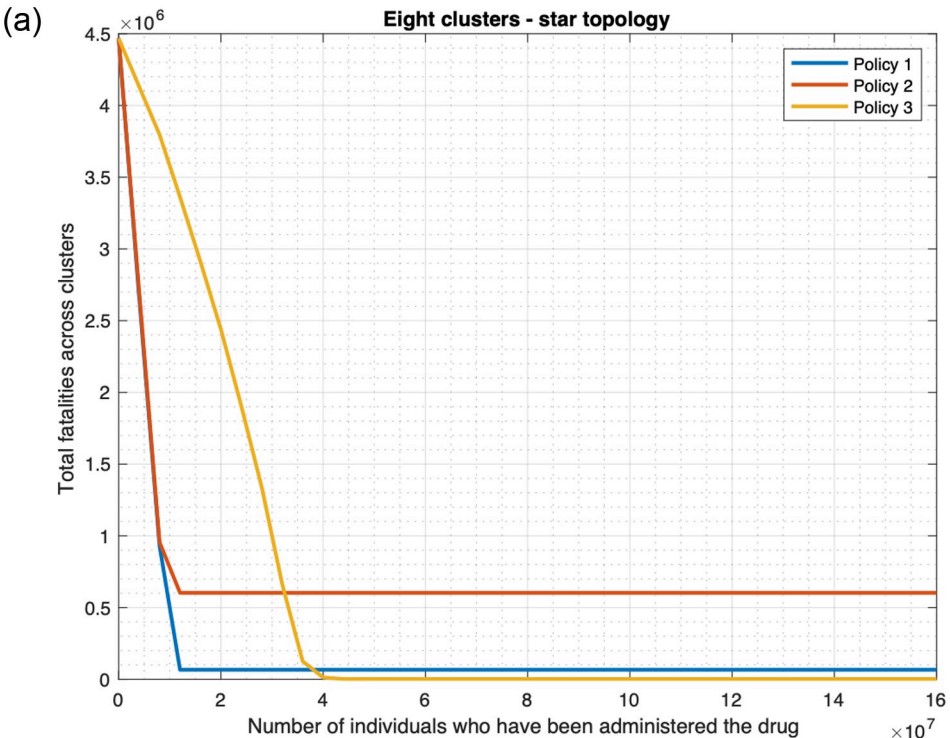

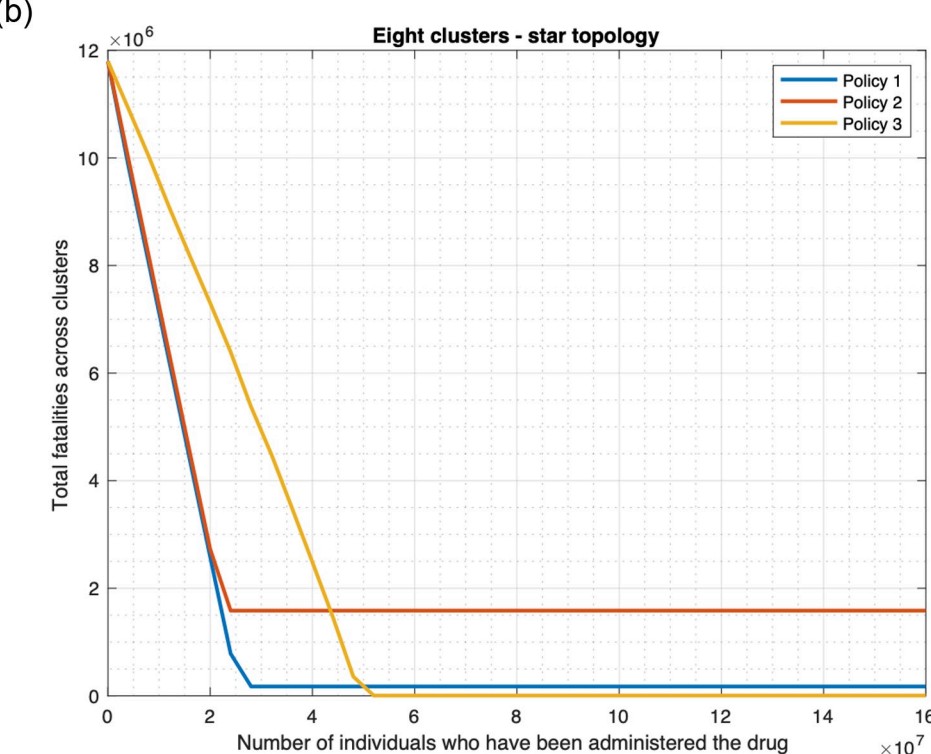

**Fig 17. Policies for drug deployment (star topology).** (a) Initially infected individuals are in the central cluster. (b) Initially infected individuals are distributed uniformly across clusters.

(a)

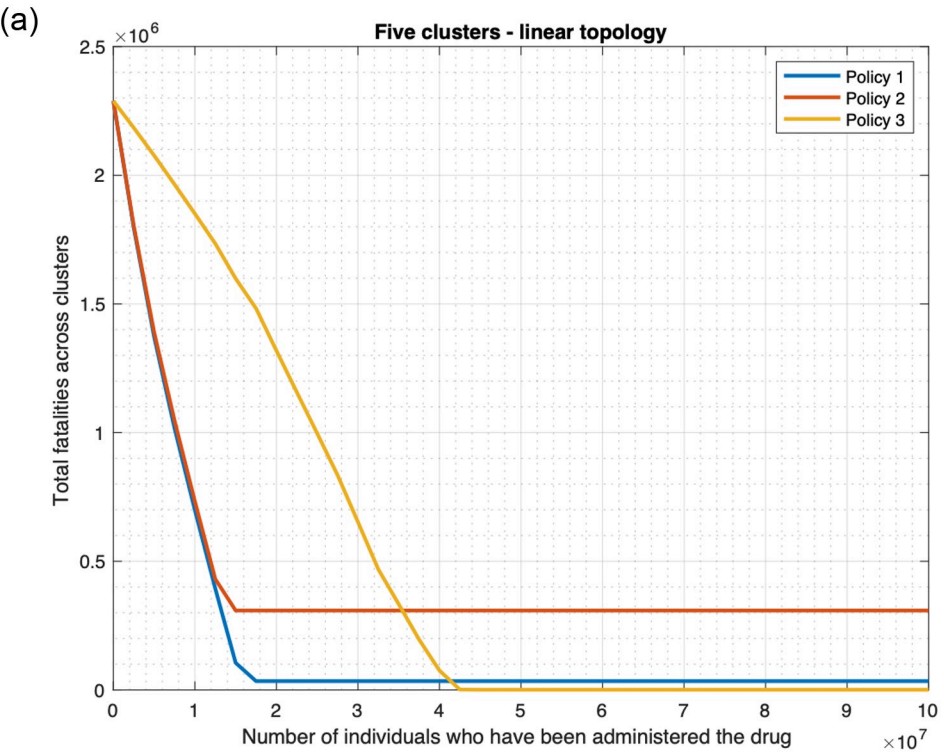

(b)

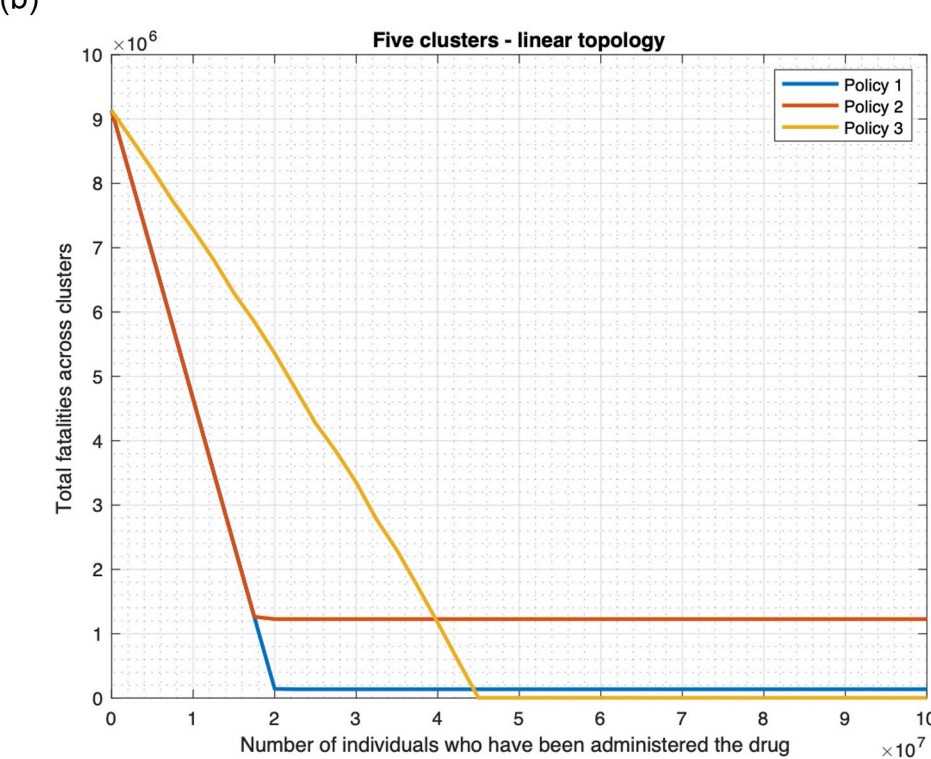

**Fig 18. Policies for drug deployment (linear topology).** (a) Initially infected individuals are in a peripheral cluster. (b) Initially infected individuals are distributed uniformly across clusters.

**Table 2. The four tables respectively compare the actual fatality count and linear interpolations for policy 1 "both drug and vaccine scenario" and 1) eight cluster star topology—Central (Fig 17a) 2) eight cluster star topology—Uniform (Fig 17b) 3) five cluster linear topology—Peripheral (Fig 18a) 4) five cluster linear topology—Uniform (Fig 18b). The columns A, B, C have the same significance as in Table 1.**

| 1 | | | |
|---|---|---|---|
| A | B | C | Error (%) |
| **0** | **4469468** | 4469468 | 0.0000 |
| 4 | 2666804 | 2717588 | -1.9043 |
| 6 | 1792914 | 1841648 | -2.7181 |
| 7 | 1360330 | 1403678 | -3.1866 |
| 8 | 928840 | 965708 | -3.9693 |
| 8.5 | 714402 | 746723 | -4.5242 |
| 9 | 500986 | 527738 | -5.3399 |
| 9.5 | 289797 | 308753 | -6.5411 |
| **10** | **89768** | 89768 | 0.0000 |

| 2 | | | |
|---|---|---|---|
| A | B | C | Error (%) |
| **0** | **11795114** | 11795114 | 0.0000 |
| 4 | 9849097 | 9962080 | -1.1471 |
| 8 | 8031636 | 8129045 | -1.2128 |
| 10 | 7121221 | 7212528 | -1.2822 |
| 12 | 6210846 | 6296011 | -1.3712 |
| 16 | 4392023 | 4462977 | -1.6155 |
| 20 | 2580787 | 2629943 | -1.9047 |
| 24 | 783938 | 796908 | -1.6545 |
| **25** | **338650** | 338650 | 0.0000 |

| 3 | | | |
|---|---|---|---|
| A | B | C | Error (%) |
| 0 | 2288928 | 2005632 | 12.3768 |
| 2.5 | 1796369 | 1688907 | 5.9822 |
| **5** | **1372182** | 1372182 | 0.0000 |
| 6 | 1223365 | 1245492 | -1.8087 |
| 7.5 | 1017290 | 1055457 | -3.7518 |
| 8.5 | 887282 | 928767 | -4.6755 |
| 10 | 698390 | 738733 | -5.7766 |
| 12.5 | 394234 | 422008 | -7.0451 |
| **15** | **105283** | 105283 | 0.0000 |

| 4 | | | |
|---|---|---|---|
| A | B | C | Error (%) |
| **0** | **9132247** | 9132247 | 0.0000 |
| 2.5 | 8021838 | 8008540 | 0.1658 |
| 5 | 6903523 | 6884833 | 0.2707 |
| 7.5 | 5778566 | 5761125 | 0.3018 |
| 10 | 4649803 | 4637418 | 0.2664 |
| 12.5 | 3518485 | 3513711 | 0.1357 |
| 15 | 2388013 | 2390004 | -0.0834 |
| 17.5 | 1260262 | 1266296 | -0.4788 |

(*Continued*)

**Table 2.** (Continued)

| | | | |
|---|---|---|---|
| **20** | **142589** | 142589 | 0.0000 |

**(a)** Average of the absolute value of the error in the above range is 2.8412%. The maximum of the absolute value of error in 10–10.1 million range is 28.02%, and is registered at 10.05 million.

**(b)** Average of the absolute value of the error in the above range is 1.4178%. The maximum of the absolute value of error in 25–25.5 million range is 6.83%, and is registered at 25.358 million.

**(c)** Average of the absolute value of the error in the above range is 5.6191%. The maximum of the absolute value of error in 15–15.6 million range is 29.92%, and is registered at 15.56 million.

**(d)** Average of the absolute value of the error in the above range is 0.2128%. The maximum of the absolute value of error in 20–20.05 million range is 0.28%, and is registered at 20.01 million.

and PeripheralNC. When we consider only the first case, which is the default, we omit the suffixes altogether. The columns titled Normalized Linear (denoted by P) and Normalized Star (denoted by Q) respectively represent the per cluster fatalities for the Linear and Star topologies respectively. We obtain the difference between the per cluster fatalities of the star and the linear topologies (Column titled R) and present the difference as a percentage of the former in the tables (Column titled Change (%)). In Table 3a–3d, we consider the scenario in which both drug and vaccine are administered ("Both"). In Table 3e, we additionally consider "drug only", "vaccine only", and "neither" (no countermeasure) scenarios.

We consistently observe the following from the tables: 1) the per cluster fatalities of the two topologies are about the same for the Uniform distribution; 2) the per cluster fatality is considerably higher in the star topology for the Central and Peripheral distributions. The first happens because the initial number of infected individuals is the same in each cluster and the mobility rates to and from each cluster are the same throughout. We conjecture that the per cluster topologies will not significantly vary depending on the placement of the clusters for Uniform distribution and symmetric mobility patterns. Now consider the Central and Peripheral distributions. The average distance between a central or a peripheral cluster to the other clusters is lower for the star. Thus, if the initially infected individuals are concentrated in only the central or the peripheral cluster, over time the infection spreads faster in the star as compared to the linear leading to a greater number of per cluster fatalities in the former.

### 3.5 Drug administering policy has a significant impact

We compare various drug administering policies with respect to (1) fatalities (2) health care load (number of doctor visits), considering the scenario in which both drug and vaccine are administered.

First we consider only one cluster and then compare policy 1 (i.e., all people with fever and rash receive drugs) and policy 2 (i.e., only people with rash receive drugs). Policy 1 preempts 3, 911, 245 people with antiviral drugs while 26, 752 people died. Policy 2 preempts 3, 455, 823 people with drugs while 241, 433 people died. Thus, the fatality under policy 2 is approximately 9 times the fatality under policy 1. We get the first indication that different policies for administering drugs substantially alter the fatality count. But, clearly, policy 1 will need greater number of drugs as it preempts a greater number of individuals. We therefore next compare different policies for a finite supply of drugs.

We consider three different policies for multiple clusters and limited availability of drugs. Policies 1 and 2 are as before. Under policy 3, drugs are administered to everyone in a cluster once the number of patients with symptoms in the cluster exceeds a certain level. We consider

**Table 3. Per cluster fatality.** The five tables respectively compare the per cluster fatality counts for the "both drug and vaccine" scenario for different opinion spread rates ($\alpha$) (Table 3a and 3b), preemption scale factor ($\lambda$) (Table 3c), mobility rates ($\kappa$) (Table 3d), and drug supply ($m$) (Table 3e). In these tables, the first column (P) represents Normalized linear, the second column (Q) represents Normalized star, and the third column (R) provides $Q - P$, the fourth column, Change = $100 \times R/P$.

| a | | | | |
|---|---|---|---|---|
| Distribution | P | Q | R | Change (%) |
| Uniform/$\alpha = 0$ | 27810 | 27747 | -63 | -0.227 |
| UniformC/$\alpha = 1$ | 19666 | 19598 | -68 | -0.346 |
| UniformNC/$\alpha = 1$ | 44122 | 44012 | -110 | -0.249 |
| Central/$\alpha = 0$ | 11138 | 15270 | 4132 | 37.10 |
| CentralC/$\alpha = 1$ | 4, 041 | 5, 067 | 1, 026 | 25.39 |
| CentralNC/$\alpha = 1$ | 18247 | 25093 | 6846 | 37.52 |
| Peripheral/$\alpha = 0$ | 8421 | 10879 | 2458 | 29.19 |
| PeripheralC/$\alpha = 1$ | 4006 | 4978 | 972 | 24.26 |
| PeripheralNC/$\alpha = 1$ | 13639 | 17660 | 4021 | 29.48 |

| b | | | | |
|---|---|---|---|---|
| Distribution | P | Q | R | Change (%) |
| Uniform/$\alpha = 0$ | 56, 309 | 56, 293 | -16 | -0.028 |
| UniformNC/$\alpha = 1$ | 62, 208 | 62, 176 | -32 | -0.051 |
| Central/$\alpha = 0$ | 33, 316 | 47, 038 | 13, 722 | 41.19 |
| CentralNC/$\alpha = 1$ | 36, 526 | 51, 542 | 15, 016 | 41.11 |
| Peripheral/$\alpha = 0$ | 22, 516 | 29, 657 | 7, 141 | 31.72 |
| PeripheralNC/$\alpha = 1$ | 24, 729 | 32, 561 | 7, 832 | 31.67 |

| c | | | | |
|---|---|---|---|---|
| Distribution | P | Q | R | Change (%) |
| Uniform/$\lambda = 0$ | 4, 400, 000 | 4, 400, 000 | 0 | 0 |
| Uniform/$\lambda = 1$ | 27, 758 | 27, 694 | -64 | -0.2306 |
| Central/$\lambda = 0$ | 2, 687, 531 | 3, 823, 157 | 1, 135, 626 | 42.26 |
| Central/$\lambda = 1$ | 10, 786 | 14, 815 | 4, 029 | 37.35 |
| Peripheral/$\lambda = 0$ | 1, 800, 793 | 2, 373, 055 | 572, 262 | 31.78 |
| Peripheral/$\lambda = 1$ | 8, 233 | 10, 611 | 2, 378 | 28.88 |

| d | | | | |
|---|---|---|---|---|
| Distribution | P | Q | R | Change (%) |
| Uniform/$\kappa = 0$ | 27, 891 | 27, 694 | -197 | -0.7063 |
| Uniform/$\kappa = 1$ | 27, 891 | 27, 694 | -197 | -0.7063 |
| Central/$\kappa = 0$ | 2, 443 | 6, 952 | 4, 509 | 184.57 |
| Central/$\kappa = 1$ | 15, 083 | 21, 034 | 5, 951 | 39.46 |
| Peripheral/$\kappa = 0$ | 2, 443 | 6, 952 | 4, 509 | 184.57 |
| Peripheral/$\kappa = 1$ | 14, 361 | 18, 528 | 4, 167 | 29.02 |

| e | | | | |
|---|---|---|---|---|
| Distribution | Normalized Linear | Normalized star | Difference | Change (%) |
| Both/uniform, $m = 0$ | 1, 858, 493 | 1, 895, 512 | 37, 019 | 1.99 |
| Both/uniform, $m = 50$ million | 27, 764 | 28, 326 | 562 | 2.02 |
| Drug/uniform, $m = 0$ | 4, 400, 000 | 4, 400, 000 | 0 | 0 |
| Drug/uniform, $m = 50$ million | 65, 808 | 65, 808 | 0 | 0 |
| Vaccine/uniform, $m = 0$ | 1, 858, 493 | 1, 895, 512 | 37, 019 | 1.99 |
| Vaccine/uniform, $m = 50$ million | 1, 858, 493 | 1, 895, 512 | 37, 019 | 1.99 |
| Neither/uniform, $m = 0$ | 4, 400, 000 | 4, 400, 000 | 0 | 0 |
| Neither/uniform, $m = 50$ million | 4, 400, 000 | 4, 400, 000 | 0 | 0 |
| Both/central, $m = 0$ | 737, 058 | 1, 013, 650 | 276, 592 | 37.53 |
| Both/central, $m = 50$ million | 10, 803 | 14, 842 | 4, 039 | 37.39 |

(*Continued*)

**Table 3.** (Continued)

| Drug/central, $m = 0$ | 2, 690, 869 | 3, 823, 159 | 1, 132, 290 | 42.08 |
|---|---|---|---|---|
| Drug/central, $m = 50$ million | 41, 522 | 57, 627 | 16, 105 | 38.79 |
| Vaccine/central, $m = 0$ | 737, 058 | 1, 013, 650 | 276, 592 | 37.53 |
| Vaccine/central, $m = 50$ million | 737, 058 | 1, 013, 650 | 276, 592 | 37.53 |
| Neither/central, $m = 0$ | 2, 690, 869 | 3, 823, 159 | 1, 132, 290 | 42.08 |
| Neither/central, $m = 50$ million | 2, 690, 869 | 3, 823, 159 | 1, 132, 290 | 42.08 |
| Both/peripheral, $m = 0$ | 558, 877 | 721, 110 | 162, 233 | 29.03 |
| Both/peripheral, $m = 50$ million | 8, 242 | 10, 627 | 2, 385 | 28.94 |
| Drug/peripheral, $m = 0$ | 1, 804, 136 | 2, 373, 052 | 568, 916 | 31.53 |
| Drug/peripheral, $m = 50$ million | 28, 028 | 35, 916 | 7, 888 | 28.14 |
| Vaccine/peripheral, $m = 0$ | 558, 877 | 721, 110 | 162, 233 | 29.03 |
| Vaccine/peripheral, $m = 50$ million | 558, 877 | 721, 110 | 162, 233 | 29.03 |
| Neither/peripheral, $m = 0$ | 1, 804, 136 | 2, 373, 052 | 568, 916 | 31.53 |
| Neither/peripheral, $m = 50$ million | 1, 804, 136 | 2, 373, 052 | 568, 916 | 31.53 |

(a) Per cluster fatality in Figs 8a, 8c, 9c and 9e. In Figs 8a, 8c, 9c and 9e, we consider a star topology with 4 clusters and a linear topology with 5 clusters respectively. C in UniformC, CentralC, PeripheralC denotes cooperatives converting non-cooperatives. Similarly, NC in uniformNC, CentralNC, PeripheralNC denotes non-cooperatives converting cooperatives. When $\alpha = 0$, opinion does not spread, so the value for "C" and "NC" are the same. Thus, the prefixes are omitted.

(b) Per cluster fatality in Fig 9d and 9f. In these figures, non-cooperatives convert cooperatives and initial cooperativity is 0.2. In Fig 9d and 9f, we consider a star topology with 4 clusters and a linear topology with 5 clusters respectively. NC in uniformNC, CentralNC, PeripheralNC denotes non-cooperatives converting cooperatives.

(c) Per cluster fatality in Fig 10a and 10c. In Fig 10a and 10c, we consider a star topology with 4 clusters and a linear topology with 5 clusters respectively.

(d) Per cluster fatality in Fig 12. In Fig 12a and 12b, we consider a star topology with 4 clusters and a linear topology with 10 clusters respectively.

(e) Per cluster fatality in Fig 16. In Fig 16b and 16c, we consider a star topology with 4 clusters and a linear topology with 5 clusters respectively.

this level as 2000, that is, 0.02% of the population in each cluster. Drugs can be administered as long as the supply lasts.

As shown in Figs 17 and 18, for a large range of the distributions of the initially infected individuals and different topologies and number of clusters, policy 1 always has lower fatality than policy 2. For low values of drug supply, policies 1 and 2 attain lower fatality as compared to policy 3. Thus, policy 1 is the best in this case. Once the drug supply exceeds a certain level, policy 3 attains the lowest fatality among all 3 policies. Intuitively, when the availability of drugs is large, the policy that preemptively administers drugs to most individuals who are in proximity of infected individuals would attain the lowest fatalities. This is what policy 3 does. In contrast, when the drug availability is limited, policies that administer drugs to individuals without symptoms as well will quickly exhaust their supplies and from then onwards not be able to protect those who need it most, e.g., those with symptoms. Thus, policy 3 will not do well under these circumstances, which is what we observe. But, when the drug availability is low, it is unclear if it is better to reserve drugs for those in rash stage (policy 2) or administer drugs to those with fever (policy 1). The model informs us that the latter is a better option in this case, as by additionally preempting those with fever policy 1 reduces the number of infectious individuals and thereby reduces the spread of the infection. Also, note that the size of the supply at which fatality is lower under policy 3 substantially exceeds the threshold for policy 1 (recall that the threshold for a policy is the supply size at which its fatality plot flattens). Thus, until its threshold value, policy 1 is the best choice.

The fatalities significantly differ depending on the policy we deploy. For instance, when the initial infections are uniformly distributed across the star topology and the quantity of drugs available is enough for the population, the ratio between the fatalities of the worst and best

policies at any given point can become as high as 323 and that between the fatalities of the second best and best policies at any given point can become as high as 35.6 as the figures reveal.

During the outbreak of an infectious disease, the number of visits of individuals to doctors' office is an important measure of the health of a system. We consider that an individual will visit the doctor's office at the onset of fever (that is, prodrome state). Thus, the number of doctor visits by individuals equals the number of individuals who enter the prodrome state. We refer to this number as *health care load* and plot it as a function of drug supply for the three policies under consideration. As Fig 19 shows, when the drug supply is 0, the three policies will have equal health care load. This is anticipated as the policies differ only in the conditions under which drug is administered. The health care load decreases only slightly with an increase in drug supply for policy 2. This is because under policy 2 an individual is administered drug only after he develops a rash, which happens after he develops a fever, regardless of the drug supply. The slight decrease happens as greater drug supply enables the preemption of a greater number of individuals who develop rashes and thereby reduces the spread of the disease. For the other two policies, the health care load decreases considerably (and approximately linearly) with an increase in the supply of drugs until the supply reaches a level beyond which the health care load does not change with an increase in the availability of antiviral drugs. The pattern is similar to how the fatality count changes for these policies as a function of the drug supply. And, as the drug supply increases, the health care load under policy 2 becomes substantially higher than those for the other two policies. As shown in Fig 19a, once the drug supply exceeds a certain level, the health care load for policy 2 is respectively 8.94 and 350.9 times that for policy 1 and policy 3. Similarly, from Fig 19b, the health care load for policy 2 is respectively 9.04 and 276.2 times that for policy 1 and policy 3. For low values of drug supply, policy 1 attains a lower health care load as compared to policy 3. Thus, policy 1 maximally reduces the health care load among the 3 policies in this case. Once the drug supply exceeds a certain level, policy 3 attains the lowest health care load among all 3 policies. From Fig 19a, the health care load under policy 1 is 39.2 times that for policy 3 in this region. From Fig 19b, the health care load under policy 1 is 30.5 times that for policy 3 in this region. The explanation for this relative performance is similar to that for fatality counts given in the previous paragraph.

In summary, considering 3 drug administering policies and a large range of the distributions of the initially infected individuals and different topologies and number of clusters, we find that the fatalities under different administering policies are substantially different. Policy 1 (our default option) always has lower fatality than policy 2. Between policies 1 and 3, policy 1 has a lower fatality count unless the drug supply is very large and policy 3 has a lower fatality count only when the supply is very large. Similar observations apply when one considers health care load as the public health metric. Thus, policy 1 attains the best value of both public health metrics in a wide range of operating conditions.

## 4 Discussions

We now summarize the important findings and articulate their significance on preventing the spread of deadly infectious diseases.

### From results of investigation of opinion dynamics

The joint spatio-temporal spread of an infectious disease and opinions that affect behavioral dynamics pertinent to the spread of the disease, and the impact of one on the other, has not been modeled to the best of our knowledge. Thus, public debates on the overlap between information warfare and countering the spread of infectious diseases have largely been conducted in the qualitative sphere. Considering the specific example combination of smallpox (as

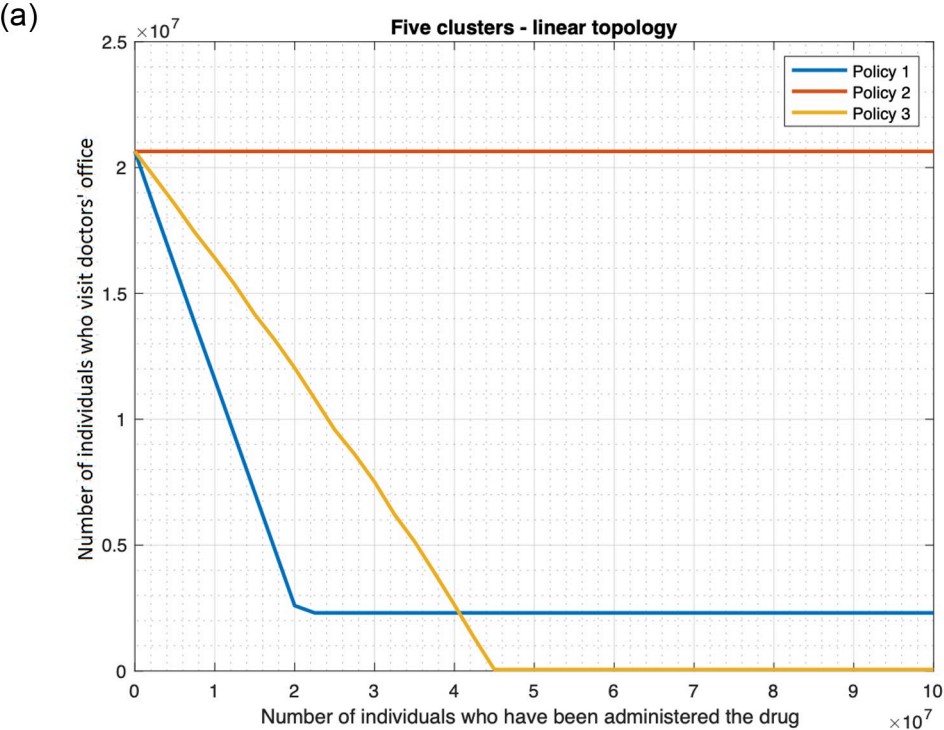

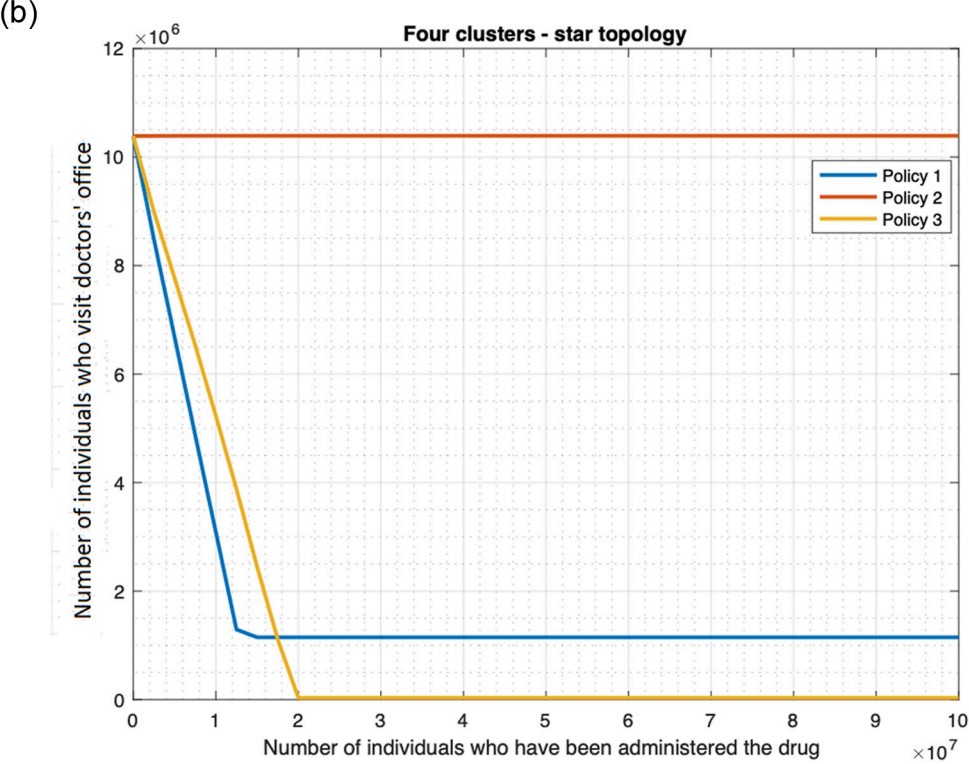

**Fig 19. Health care load against drug availability.** (a) Linear topology. (b) Star topology.

infectious disease) and vaccine hesitancy (as opinion that affects the spread of the disease), we remedy this crucial void in the study of infectious diseases. We formulate a computationally tractable mathematical model that captures the joint evolution of smallpox and vaccine hesitancy aka cooperativity (Section 2). Using this model we have shown that the evolution of cooperativity has a strong impact on fatality count. While it is intuitive that there would be some correlation between the two, the magnitude of this correlation and the nature of its dependence on the myriad of parameters that influence the evolution of the disease and opinions, can not be ascertained without a quantitative formulation. This is what this paper accomplishes. We show that for a large range of combination of parameters, namely different geographical topology of target region (number of clusters, organization of clusters), number of initially infected individuals, stage of disease of the initially infected individuals, mobility rates, preemption rates, overall fatality count sharply decreases (increases, respectively) with increase in rate of spread of opinion, $\alpha$ between cooperatives and noncooperatives when the former (latter, respectively) converts the latter (former, respectively). With increase in $\alpha$, the fatality count often changes by more than 50%. Thus, while the exact value of the fatality count depends on the choice of the values of the large number of parameters that arise in practice such as the above, we find that the pattern of the variation of fatality count with respect to $\alpha$ is similar for different choices of other parameters, e.g. number and organization of clusters, number and disease stage of the initially infected, origin cluster of the infection, mobility rate (Section 3.1), preemption rate, (Section 3.2), etc. This *stability* is not apriori evident and constitutes an useful artifact since the exact combination of the parameter values that arise in practice varies from one ambience to another. Given this stability, one can conclude based on quantitative findings that influencing exchange of opinion towards enhancing receptivity to vaccine incurs substantial public health benefits. Such influence may be attained through health education seminars, workshops, vigorous dissemination of health information on social, digital, and conventional media, and through direct engagement with influencers on these platforms.

## From results of investigation on mobility patterns and distribution of initial infection

We now demonstrate that our model can assess the impact of various different key attributes on the system above and beyond that of rate of exchange of opinions between individuals of different opinion. This helps us anticipate as to how initial infection is likely to be seeded by deliberate malevolent actors such as bioterrorists before they strike; such choices can not be apriori inferred based on intuition. We have shown that other things being equal, fatalities are (often, significantly) higher, when 1) the initial infections are uniformly distributed across various clusters than when all the initial infections only occur in one cluster; 2) the outbreak originates from the central cluster(s) than when it originates from the peripheral cluster(s) (Section 3.4). This rank ordering holds for a large number of choices of other parameters, namely, number and organization of clusters, mobility rates, preemption rates, opinion exchange rates, distribution of initial cooperatives etc (Section 3.4). The pattern is therefore stable. Even higher rate of exchange of opinions between cooperatives and noncooperatives that convert the latter to the former is least effective in reducing the fatalities when initially infected individuals are distributed uniformly and most effective when the initially infected individuals are all in the central cluster (Section 3.2). Also, when the initial infections are uniformly distributed, fatalities in each cluster are the same irrespective of the mobility rates between the clusters (Section 3.2). On the other hand, when the initial infections occur in one cluster only, total fatalities would escalate with an increase in mobility rates (Section 3.2). Thus, isolating regions from

one another by reducing the mobility rates between them provides no public health benefits when the initial infections are uniformly distributed. Considering all the above, including the stability of the observed patterns, bioterrorist attacks will likely seed the initially infected uniformly across a region.

## From results of investigations on impact of administering drugs and various drug administration policies

Our model can assess the combination of various countermeasures and application policies. We use it to investigate the impact of administering drugs and choose between various conceivable policies for administering drugs. We choose drugs because: 1) vaccine hesitancy and immunocompromise are widely prevalent and drugs are the only recourse in the former and often more effective recourse in the latter; 2) models that consider the impact of administering drugs to prevent and cure smallpox are rare (the only one we could find, namely [1], makes several restrictive assumptions e.g., supply of drugs is unlimited, vaccine hesitancy does not evolve with time, etc). Considering 3 drug administering policies, we find that the fatalities under different administering policies are substantially different. Unless the drug supply is very large, policy 1 has the least fatality count and health care load among these three policies. Policy 3 has the least fatality count and number of health care load only when the supply is very large. Thus, policy 1 (administering drugs to everyone with fever or rash) attains the best value of both public health metrics in a wide range of operating conditions (Sections 3.3, 3.5). These findings are not apriori evident.

We therefore study policy 1 in depth. Considering different numbers and organization of clusters and also different distributions of the initially infected individuals, we find that the fatality counts and health care load of policy 1 decrease approximately linearly with increase in supply of drugs until the supply is enough to administer to everyone who develops symptoms (fever and rash), and beyond that point these counts do not change with increase in the supply (Sections 3.3, 3.5). While it may be guessed apriori that these counts would decrease with increase in the supply of drugs, the linear nature of the decrease is not apriori evident. The significance of linearity is as follows. Any linear plot can be fully constructed from the knowledge of two points on it. Since the plots flatten after a linear phase, each such plot can be entirely constructed from two points during the linear decrease phase (which provide the slope and the starting point) and a third in the flat phase (which provides the ending value of the linear phase). Thus, three points, which can be chosen anywhere respectively in the linear and flat phases, provide the exact fatality count and health care load for a given amount of drug supply. From the characterization of the linear expressions connecting the supply and the above counts, the minimum supply needed to limit the above counts below acceptable limits can be easily determined. For example, if fatality count is $B - Ax$ where $x$ is the supply, and $A$, $B$ can be determined as above from the characterization above, and $C$ is the upper bound on fatality that can be tolerated, the requisite supply is obtained by solving for $x$ when $B - Ax = C$; it is $(B - C)/A$. This simple characterization of the requisite supply has practical utility in public health and can not be apriori guessed without the computations.

Next, considering policy 1, our numerical computations show that the most effective countermeasure consists of a combination of antiviral drugs and vaccines, but if only one countermeasure can be administered it ought to be the former (Section 3.3). Again, the fatality counts under different countermeasures are significantly different, in particular, administering drugs substantially reduces the fatality count. The nature of these findings can be explained after the fact, but magnitudes of the differences can only be obtained through the computations. Finally, we have shown that the fatalities substantially decrease if the delay in administering vaccines

and delivering drugs to individuals decreases. This is consistent with commonplace intuition, but the quantification of the amount of decrease allows for more founded public health policy choices (since the delay is a function of availability of vaccines and drugs and health workers at healthcare facilities).

### From results of investigation on impact of topology

Considering various values of opinion exchange rates, mobility rates, preemption rates, disease spread rates, initial cooperativities, we find that for each combination, other things being equal, the star topology has 1) considerably higher fatalities per cluster than the linear topology when the infection originates from one cluster; and 2) about same fatality counts per cluster as the linear topology when the initially infected individuals are uniformly distributed (Section 3.4). In the two cases, the differences respectively are 1) 30–40% 2) below 2%. Thus yet again there is a stable pattern (one that is consistent across the wide range of parameters mentioned above). Neither the specific pattern nor its stability is apriori evident. Nonetheless, after the fact, one can see that the underlying reason for the higher fatality is a higher spread in star because it has a central cluster in the star topology which is adjacent to all other clusters, while the central cluster in the linear topology is much farther off from the peripheral clusters. This has important implications on pandemic sensitive urban design—topologies that have central regions that are close to all other regions will be more vulnerable to an outbreak and hence better avoided. If such topologies are inevitable because of legacy issues or because of fundamental constraints such as geography, then there is a case to monitor those more closely and stock a greater amount of drugs for the residents, and incentivize formation of consensus to accept vaccines as and when available.

## 5 Conclusion and generalizations

In summary, we have formulated the first mathematical model for the joint spatio-temporal spread of an infectious disease and an opinion that affects behavioral dynamics pertinent to the spread of the disease. This model is flexible enough to incorporate the impact of various attributes that determine the nature of the spread of the disease and pertinent opinion, namely, spatial topology, different combinations of countermeasures, opinion spread rate, disease spread rate, mobility rate, preemption rate, initial cooperativity, distributions of initially infected individuals and initially cooperative individuals. It is also computationally tractable. As such, it helps provide a quantitative basis to public discourse on various elements at the intersection of spread of infectious diseases and information warfare that have hitherto been conducted in only qualitative sphere. The numerical computations using the model confirm various intuitions, quantifies public health metrics such as fatality, health care load, and reveal patterns of their variations with respect to specific parameters, the patterns that are stable to large scale variation of all other parameters. Such patterns can not be intuited apriori, and their stability is invaluable because many of these parameters assume widely differing values in different environments (that is, standard values can not be assumed). As such, stable patterns of variations help us anticipate strategies that deliberate malevolent actors like bioterrorists may adopt before the attacks are launched, and facilitates the designs of public health policies that may thwart and defend against pandemics including but not limited to deliberate attacks.

We have chosen smallpox as an example of an infectious disease in designing the above model because (1) it is highly infectious; (2) it has a high death rate; and (3) its disease progression parameters are known with reasonable certainty owing to years of research. We have chosen vaccine hesitancy as an example of an opinion that influences the spread of an infectious disease. But the model can be generalized to other infectious diseases which spread through

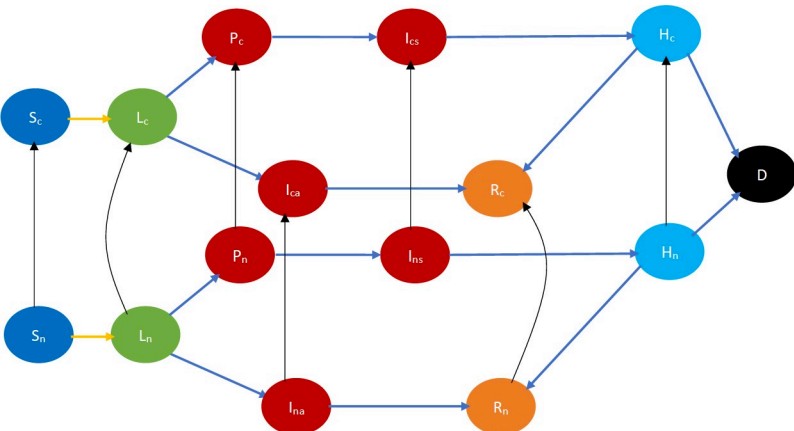

**Fig 20. COVID-19 state diagram.** The symbol *S* denotes susceptibles, *L* denotes individuals in the latent stage, *P* stands for pre-symptomatic, $I_s$ denotes symptomatic stage, $I_a$ denotes asymptomatic stage, *H* stands for hospitalized, *R* denotes recovered, and *D* stands for dead. The suffixes *c* and *n* respectively stand for cooperatives and non-cooperatives e.g., $S_c$ and $S_n$ denote cooperative and non-cooperative susceptible individuals respectively. Similarly, $I_{ca}$ and $I_{na}$ stand for cooperative asymptomatic and non-cooperative asymptomatic individuals respectively.

proximity between the individuals, and other opinions that influence their spread. We now describe the generalization to an arbitrary infectious disease. As for smallpox, first, consider the case that all individuals are in the same neighborhood, that is, they interact with each other at the same rate (homogeneous mixing). Each individual is either immunocompetent or immunodeficient. Now, considering the case in which there are no countermeasures, individuals in either category may be in a list of states, a list that is determined by the stages of progression of the disease in question. Susceptible, recovered and dead constitute stages for almost all diseases. For smallpox, the additional stages were early incubation, late incubation, prodrome, early rash, late rash. For another disease, COVID-19, for example, the additional stages are latent, pre-symptomatic, symptomatic, asymptomatic (see Fig 20). In the latent stage, an individual has been infected, but is not infectious; this stage is analogous to the incubation stages. The individual is infectious in the pre-symptomatic, symptomatic, asymptomatic stages, and has symptoms only in the symptomatic stage. One can also conceive of stages like hospitalized. For COVID-19, the immunodeficient characterization may be substituted by high age or comorbidity which renders an individual vulnerable to severe forms of the disease. Fig 20 depicts the interactional and the non-interactional state transitions pictorially.

The model for the cases in which drugs, vaccines can prevent and treat smallpox have been developed building upon the model for the scenario in which there is no countermeasure. The generalizations have been accomplished through addition of states such as preempted and including descriptors such as cooperatives, non-cooperatives to denote their willingness to receive vaccines. This methodology extends to other infectious diseases as well. Let us consider the case of COVID-19 as an example. There is till date no therapeutic that prevents the disease, though several have been developed for potentially reducing the severity once one develops the disease [39]. Thus, as of today, in the "drug only" scenario, there would not be any preempted state, but the probability of transition from hospitalized to dead or recovered will depend on whether therapeutics are administered. In the "vaccine only" and "both drug and vaccine" scenario we would have the preempted state, similar to Figs 3 and 4. The differences between COVID-19 and smallpox are: 1) the stages of the disease would be drawn from Fig 20 instead of Figs 1 and 2) the transition to the preempted state will only be due to vaccination and can therefore happen only from the susceptible or latent stages; 3) immunocompetent and

immunodeficient would get the same vaccine (the immunodeficient are more likely to develop severe forms of disease and have higher fatality rates). Thus, the "vaccine only" and "both drug and vaccine" scenarios differ only in probabilities of transition from hospitalized to dead and recovered. For COVID-19, the opinions of individuals regarding their willingness to receive the vaccine will be represented by classifying them as cooperative and non-cooperative as for smallpox. The interactional transitions due to 1) infection (from susceptible to latent stages); or 2) exchange of opinions on whether to receive vaccine are similar to the "vaccine only" scenario for smallpox (Fig 3). The latter constitutes transition from non-cooperative to cooperative. Note that those with specific allergies should not receive COVID-19 vaccines, or at least both doses of the vaccine [40]. The fraction of such individuals is small. One can either consider a separate category for such individuals, or equivalently consider these individuals as those for which vaccines do not provide immunity, and they do not transition to the pre-empted state even after receiving the vaccines. In the latter case, the probability that the vaccine provides immunity can be adjusted to account for the presence of these individuals. We adopt the second approach in our pictorial depiction.

Like for the spread of smallpox, the impact of spatial heterogeneity can be captured for other infectious diseases by dividing the target geographical area into *clusters*, and considering the cluster an individual inhabits as part of his state description in addition to his cooperativity, stage of the disease, preemption, immunocompetency or immunodeficiency. Mobility rates, infection rates and opinion exchange rates among clusters may be represented by matrices as for smallpox.

We now outline the design of the CEDE representing the spatio-temporal evolution of the system. Like for smallpox, each variable in the CEDE represents the fraction of the population who are in a particular system state, each state representing the combination of the cluster inhabited, the stage of the disease, immunocompetency, and cooperativity. Each differential equation captures the evolution of a particular variable. The terms in the differential equations are either quadratic or linear. The quadratic ones represent the interactional transitions (refer to the terms in green color in CEDE equation (1)—(4), (16)—(19), and the terms in orange color in (16)—(29)), and the linear ones represent the non-interactional transitions (refer to the terms in red in (3)—(14), (18)—(29), and (31), blue terms in (1)—(13), (16)—(30), as well as all the black terms in (15), (30), (32), and (33)). We omit the differential equations for brevity. But, as for smallpox, their solution provides the fraction of individuals in different states at given times, that is, the spatio-temporal distribution of the disease and opinion spread.

As the first work in this field, while developing a mathematical model for the joint evolution of disease and opinions we have resorted to some simplifications so as to focus on the essences. One such simplification has been to assume that the willingness to be vaccinated is a binary indicator; but in practice, beliefs have different strengths. We outline some generalizations that can render the model more realistic. One possibility is to consider different classes of non-cooperative individuals, with the connotation that the classes have different degrees of entrenchments in their beliefs. The classes that are more entrenched have lower probability of transition to cooperative after each interaction with cooperative individuals. If the probability of conversion is low in each interaction, a larger number of interactions would be required for a conversion on an average. Thus more entrenched individuals become cooperative only after several interactions with cooperatives. In this case, the state of an individual will denote which class of noncooperation he belongs to if he is noncooperative, and the probability of conversion upon interacting with a cooperative will depend on the class indicator. Another possibility is to consider that noncooperatives transition across various classes, or stages, before becoming a cooperative. Thus there are stages of conviction like stages of disease. That is, the noncooperatives in the heavily entrenched stage transition to a less entrenched one upon interacting

with a cooperative, and upon a subsequent interaction transition to an even less entrenched stage. This process continues till a noncooperative becomes a cooperative. Thus, every interaction with a cooperative increases the willingness of a noncooperative to receive the vaccine, and eventually the willingness reaches the level at which an individual agrees to receive a vaccine. In this case, the state of an individual will again denote which stage of noncooperation he belongs to and the interactional transitions will include transitions between these stages. Finally, another limitation of our model has been to assume only one kind of transition for opinion, that is either cooperatives convert noncooperatives or vice versa. In practice, both transitions may proceed simultaneously, that is, cooperatives convert noncooperatives and noncooperatives convert cooperatives. Such bidirectional transitions will include individuals changing their opinions back and forth. The CEDE can be easily generalized to accommodate this provision, we outline the last generalization in Appendix A.4 in S1 Appendix.

## Supporting information

**S1 Text. Proof of convergence of the stochastic state distribution to CEDE solution.** The proof proceeds in the following sequence. (1) We provide a Continuous Time Markov Chain (CTMC) formulation for the stochastic version of our system. (2) We present a classical result of probability theory that guarantees convergence of a CTMC to a system of differential equations provided the CTMC satisfies some regularity conditions. (3) and (4) We prove that the CTMC for our system satisfies the regularity conditions. (5) We show that the system of differential equations in the classical result is the CEDE for our system.
(PDF)

**S1 Appendix.**
(PDF)

## Author Contributions

**Conceptualization:** Rex N. Ali, Harvey Rubin, Saswati Sarkar.

**Data curation:** Rex N. Ali, Saswati Sarkar.

**Formal analysis:** Rex N. Ali, Harvey Rubin, Saswati Sarkar.

**Investigation:** Rex N. Ali, Harvey Rubin, Saswati Sarkar.

**Methodology:** Rex N. Ali, Harvey Rubin, Saswati Sarkar.

**Project administration:** Rex N. Ali, Harvey Rubin, Saswati Sarkar.

**Software:** Rex N. Ali.

**Supervision:** Harvey Rubin, Saswati Sarkar.

**Validation:** Harvey Rubin, Saswati Sarkar.

**Visualization:** Rex N. Ali, Saswati Sarkar.

**Writing – original draft:** Rex N. Ali, Saswati Sarkar.

**Writing – review & editing:** Rex N. Ali, Harvey Rubin, Saswati Sarkar.

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
