## [Decision Letter · Decision Letter 0]

18 Feb 2021

PONE-D-20-32001

Countering the potential re-emergence of a deadly infectious disease - information warfare, identifying strategic threats, launching countermeasures

PLOS ONE

Dear Dr. Ali,

Thank you for submitting your manuscript to PLOS ONE. After careful consideration, we feel that it has merit but does not fully meet PLOS ONE’s publication criteria as it currently stands. Therefore, we invite you to submit a revised version of the manuscript that addresses the points raised during the review process.

A particular attention should be addressed to highlight the central results of the research and their originality.  Also the modeling choices need to be better motivated.

We look forward to receiving your revised manuscript.

Kind regards,

Floriana Gargiulo

Academic Editor

PLOS ONE

Journal Requirements:

2. We note that the authors explore outcomes of theoretical policies using their epidemiological model, and then provide policy recommendations based on these results. PLOS ONE generally requires a higher bar of evidence for making actual policy recommendations, and we therefore ask you to remove your policy recommendations or references to policy recommendations in the abstract, keywords, last paragraph of the introduction, and throughout the discussion. We feel that you have provided evidence that may support your results and discussion relating to what your model shows in regards to the different theoretical policies tested, however we ask that you remove the specific policy recommendations that you provide to public health authorities.

3. In your financial disclosure in your online submission form, please describe the type of gift that was provided by SIGA Technologies, Inc. (monetary, analysis support, equiptment, etc.).

Reviewers' comments:

Reviewer's Responses to Questions

**Comments to the Author**

1. Is the manuscript technically sound, and do the data support the conclusions?

Reviewer #1: Yes

Reviewer #2: Partly

Reviewer #3: Yes

2. Has the statistical analysis been performed appropriately and rigorously? 

Reviewer #1: Yes

Reviewer #2: No

Reviewer #3: Yes

3. Have the authors made all data underlying the findings in their manuscript fully available?

Reviewer #1: Yes

Reviewer #2: Yes

Reviewer #3: No

4. Is the manuscript presented in an intelligible fashion and written in standard English?

Reviewer #1: Yes

Reviewer #2: Yes

Reviewer #3: No

5. Review Comments to the Author

Reviewer #1: Important note: This review pertains only to ‘statistical aspects’ of the study and so ‘clinical aspects’ [like medical importance, relevance of the study, ‘clinical significance and implication(s)’ of the whole study, etc.] are to be evaluated [should be assessed] separately/independently. Further please note that any ‘statistical review’ is generally done under the assumption that (such) study specific methodological [as well as execution] issues are perfectly taken care of by the investigator(s). This review is not an exception to that and so does not cover clinical aspects {however, seldom comments are made only if those issues are intimately / scientifically related & intermingle with ‘statistical aspects’ of the study}. Agreed that ‘statistical methods’ are used as just tools here, however, they are vital part of methodology [and so should be given due importance].

COMMENTS: Your ABSTRACT is well drafted but assay type. Please note that it is preferable [though this article is not of ‘usual’ type] to divide the ABSTRACT with small sections like ‘Objective(s)’, ‘Methods’, ‘Results’, ‘Conclusions’, etc. [refer to item 1b of CONSORT checklist 2010: Structured summary of trial design, methods, results, and conclusions] which is an accepted practice of most good/standard journals [including PLOS-ONE]. It will definitely be more informative then, I guess, whatever the article type may be {even if section headings may be different considering nature of this article}.

Surprisingly, in ‘Abstract’ the complete account [though very nice] is focused on ‘smallpox’ and no mention of ‘COVID-19’ {agreed, the article is on ‘Countering the potential re-emergence of a deadly infectious disease’ some mention of current pandemic / COVID-19 outbreak was/is expected to emphasize present context / implications / applications (mainly because at the end of ‘Introduction’ section, it is said that “we expect that our model would easily port to other infectious diseases such as COVID-19 through the consideration of a different set of disease states and parameters.”}. Many readers (as you are aware) decide to read full text on the basis of ‘abstract’.

Overall, the article is nicely presented, perfect by all the means, 100% correct/accurate (scientifically). However, there are few small questions: Why the table headings are below the tables? (usual practice is to place them above). Was it necessary to cover all possible scenarios [(1) No countermeasure - neither drugs nor vaccines administered, (2) Vaccines only, (3) Drugs only, (4) Both drugs and vaccines.] in such details? Could not avoid/reduce number of figures? (there are 17 which is too much, I guess). Is it essential to include such a lengthy ‘appendix’ {A. Equations of Smallpox disease dynamics, B. Parameter estimation i.e. pages 33 to 49}?

Finally, I have a fundamental/basic question (though the entire article/study is faultless) is this a right place for such an article? I humbly request the ‘Editor/Chief editor’ to kindly think over.

Reviewer #2: This paper is concerned with how to counter re-emergence of infectious disease using various strategies and provide insight into potential policy recommendations based on the attained results. The general problem is, of course, of importance, but I have a number of comments:

General comments:

- the title of the manuscript is quite broad, but the work is much more focused and this misrepresentation should be clarified

- the paper seems as it is trying to do too much, needing specific assumptions and considering only parts of the problem in order to attain the results

- due to so much attempting to be tackled in the paper, the writing is quite scattered and difficult to follow in many places

Specific:

- it is not clear why much of the work is focused on smallpox, and using only a single disease spread model: this is over simplified given the lofty title of the paper

- the models are primarily based on homogeneous mixing assumptions, even though spatial heterogeneity is considered, but such models do not account for many disease factors that impact vaccine targeting strategies, and hence this again very much restricts the potential policy interventions and also actual population behavior in response to the disease

- vaccine efficacy seems not to be considered in the model

- probability modeling for belief of fake news is much too basic, and doesn't consider that the population is not equally susceptible to such information; moreover, there is a dynamic to the belief itself that isn't discussed either. A much more thorough review of this topic and discussion of the basic assumptions is needed, and to also highlight how this limits the results.

- experimental results are not very thorough - for instance, they assume 10000 initially infected individuals but it is unclear why this makes any sense, how they are properly distributed among the population, how all such people ended up at the same disease state simultaneously, etc. Similar for mobility rates, R0, etc. This is critical since the eventual goal is to provide potential strategies to mitigate the scenario, but only very restricted scenarios are considered, without providing a lot of evidence as to their plausibility or risk in reality, and then the "policies" are implemented on the simplified models...hence, it is not clear how useful the results will actually be to those in public policy

- as for the results, it isn't clear what is new to the literature -- especially since the models are quite distant from reality, such conclusions like "fatality and opinion spread are strongly related" is intuitive and already generally accepted. Similarly for "distribution of infection and mobility of individuals have significant impact"..."administering drugs reduces fatality"...and so on.

Reviewer #3: In the paper “Countering the potential re-emergence of a deadly infectious disease - information warfare, identifying strategic threats, launching countermeasures”, the authors use a Clustered Epidemiological Differential Equation (CEDE) model to analyze the spatio-temporal evolution of smallpox and opinion dynamics linked to the vaccination for smallpox. The subject is interesting, the study is pertinent to the state of the art of viruses and opinion dynamics around vaccination. The paper is very well-motivated, however, it is unnecessary long. Some structural changes should be also done. There are strong issues with the English, the article should be proof-read before any further submission. The paper also lacks formal scientific language.

1) In the following, I list several general suggestions:

- The authors should be more concise on the introduction. An overall flavour of the main result could be mentioned in the introduction. However, the place to summarize their finding is at the end of the paper in the section Conclusions.

- The following statement: “Finally, we expect that our model would easily port to other infectious diseases such as COVID-19 through the consideration of a different set of disease states and parameters.” should be placed in the section Conclusion instead of the Introduction, and it is meaningless if they do not explain how to adapt the model for the case of COVID. At least some clues.

- Several claims in the introduction need to be referenced for the strongest receptivity of the paper from readers, as an example: Connection between individuals in geographically close neighborhoods is expected to be more frequent than those between the disparate ones leading to heterogeneous interaction rates.

- The claim: “Note that a Monte Carlo simulation relying on an agent based stochastic simulation will not scale to that magnitude.” it is not necessary.

- The section methodology should be divided to explain the several states considered, the topology, and the dynamics, separately. For example, the section Vaccine should only have the explanation of the stages and not about the interactions (four different outcomes for interaction). That explanation should be transfer to another place.

- If I understand well, what is written inside the section Both drugs and vaccines, is that the individuals there receive either the drug or the vaccine. In that case, this new section should no exist, as it is part of the two previous ones.

- Figures 3 and 4 are really difficult to understand.

- Maybe it is better if the authors explain in a very simple way why it is necessary the addition of quadratic terms representing the interactional transitions due to the spread of opinions. Maybe the authors could explain the general equation for the transitions between general states, in a simple way.

- Some conclusions are almost by definition, for example: “In addition, when the initial infections are uniformly distributed, fatalities in each cluster are the same irrespective of the mobility rates between the clusters. On the other hand, when the initial infections occur in one cluster only, total fatalities would escalate with an increase in mobility rates. Therefore, policymakers should opt for cordoning off areas where infection level is high before the infection spreads widely; once the latter happens reducing mobility rates across areas is unlikely to contain the disease.”

And

“We have shown that evolution of opinion with respect to receptivity of vaccine has a strong impact on fatality count. Therefore, health authorities and other policymakers should seek to influence exchange of opinion towards enhancing receptivity to vaccine, possibly through health education seminars, workshops, vigorous dissemination of health information on social, digital and conventional media and through direct engagement with influencers on these platforms.

The authors could give more singular conclusions, showing new and interesting conclusions to the field obtained from their research.

- A final paragraph summarizing the most important results is lacking.

2) I have the following question to the authors:

When they say: “We, therefore, introduce a state called Preempted, which we denote by Q, into which individuals receiving the drug transition to with the specified probability. Once an individual transition to this state, he remains in it (since he is either cured or does not develop the disease) - thus, this is an absorbing state.” → Why? individuals receiving the drug transition can be in S again later on, can’t them?

3) Further small points

- The statement “In other words, intra-cluster interactions are higher compare to inter-cluster interactions” could be explained as grouped into communities.

- The statement “Thus, the computation time gracefully scales with the increase of the size of the system” should be reformulated in more scientific terminology.

- The following part is meaningless, either the authors show the proof or cite the references:

“The CEDE is however a deterministic model, while many of the state transitions are stochastic. Nevertheless, through an application of the strong law of large numbers, under some commonly made regularity assumptions on the stochastic evolutions, one can show that as the

number of individuals increases, the fractions of individuals in different system states in the stochastic system converge to the solutions of the CEDE, and the convergence becomes exact in the limit that the number of individuals is infinity. For the mathematical proof we refer to

a recent work by one of the co-authors involving the application of the CEDE in a different domain [23]. Thus the CEDE approximates the stochastic process better as the number of individuals increase. The commonly made regulatory assumptions under which the convergence

guarantees hold are that the stochastic evolutions are Markov, that is, the amount of time an individual spends in each system state is exponentially distributed, which is what we assume to estimate the parameters of the system."

- The following part is not related to the research performed in the paper:

The generality of the model is another important strength - it can accommodate information dynamics, arbitrary topologies, mobility patterns, countermeasure combinations, countermeasure application strategies and constraints (e.g., finite or infinite supply). Despite this generality,

our model is computationally tractable and provides analytical convergence guarantees. Finally, the model can be easily modified to accommodate diseases such as COVID-19 that spreads through contact through appropriate selection of states and parameters.

6. PLOS authors have the option to publish the peer review history of their article (what does this mean?). If published, this will include your full peer review and any attached files.

Reviewer #1: **Yes: **Dr. Sanjeev Sarmukaddam

Reviewer #2: No

Reviewer #3: No

---

## [Author Response · Author response to Decision Letter 0]

22 May 2021

RESPONSE TO REVIEWS

We would like to thank the Academic Editor and the Reviewers for sharing their reviews. We have revised our paper to address the comments therein, we believe this has substantially improved our paper. The length of our paper has increased on account of the details we have provided to address the comments of the Academic Editor and Reviewers. We have also additionally proofread our paper. In the revised paper, all the changes, including the typographic corrections, are in red. We have added several new references, [32] to [43], which have been placed at the end of the existing references (pp. 49, 50). The new references have been colored in red. As instructed, we submit another version of our paper in which we use one color throughout. Below we describe how we have addressed each comment, dedicating separate sections for the responses to the Academic Editor and each of the three reviewers. 

RESPONSE TO THE COMMENTS OF THE ACADEMIC EDITOR

A particular attention should be addressed to highlight the central results of the research and their originality. Also the modeling choices need to be better motivated.

To highlight the central results of the research and their originality, we have now 1) rewritten the last two paragraphs of the Introduction (pp. 5 - 7) 2) rewritten the Discussion section in its entirety (pp. 37 - 41) 3) summarized our important findings in the first paragraph of a newly created Conclusion and Generalization section (pp. 41, 42). In writing these, we have drawn both from the figures and tables that were already present in the previous submission and from the new figures and tables added to bolster our case (Figures 7, 8b, 8d, 10b, 10d, 11, and 13, part of Figure 9, Tables 1 and 2). Refer to the discussions on these additions in the Result Section - the text in red on pp. 20, 21 (Figure 7), p. 21 (Figures 8b and 8d), pp. 22, 23 (part of Figure 9), pp. 23, 24 (Figures 10b, 10d and 11), pp. 26, 27 (Figure 13), pp. 29, 30 (Table 1), and pp. 31, 32 (Table 2). In addition, we have revised the discussion for some existing figures: pp. 34 – 37 (Figures 17 – 19). In the last two paragraphs of the Introduction, we have better positioned our findings in context of the existing literature (pp. 5 - 7).

Many of our modeling choices were motivated in the Appendix, namely Appendix B, in the earlier submission. We have retained those, but have now additionally motivated the choice of the parameters at the start of the Result section (pp. 17 - 19). In addition, we have added another Section in the Appendix to better motivate the choice of the initial values (Appendix B.5, pp. 69 - 71). We have also included additional numerical computations in which we assess the impact of values of parameters that we had not considered earlier - Figure 7 (text in red on pp. 20 - 21), Figures 9a and 9b (text in red on pp. 22, 23), and Figure 13 (text in red on pp. 26, 27). We have added content and reworded text throughout our paper to address specific questions on modeling choices (text in red on pp. 2, 3, 4, 8, 9, 10, 12, and 13), and have included pointers to all those in our response to reviewers. 

We confirm that we have adhered to PLOS ONE’s style requirements as contained in the PLOS ONE’s style templates. 

2. We note that the authors explore outcomes of theoretical policies using their epidemiological model, and then provide policy recommendations based on these results. PLOS ONE generally requires a higher bar of evidence for making actual policy recommendations, and we therefore ask you to remove your policy recommendations or references to policy recommendations in the abstract, keywords, last paragraph of the introduction, and throughout the discussion. We feel that you have provided evidence that may support your results and discussion relating to what your model shows in regards to the different theoretical policies tested, however we ask that you remove the specific policy recommendations that you provide to public health authorities.

 As per the Academic Editor’s suggestion, we have removed all references to policy recommendations from our paper.

3. In your financial disclosure in your online submission form, please describe the type of gift that was provided by SIGA Technologies, Inc. (monetary, analysis support, equipment, etc.).

SIGA Technologies, Inc. has provided our laboratory a monetary gift. We have now mentioned this in the financial disclosure in our online submission form.

 

RESPONSE TO REVIEWER 1

Your ABSTRACT is well drafted but assay type. Please note that it is preferable [though this article is not of ‘usual’ type] to divide the ABSTRACT with small sections like ‘Objective(s)’, ‘Methods’, ‘Results’, ‘Conclusions’, etc. [refer to item 1b of CONSORT checklist 2010: Structured summary of trial design, methods, results, and conclusions] which is an accepted practice of most good/standard journals [including PLOS-ONE]. It will definitely be more informative then, I guess, whatever the article type may be {even if section headings may be different considering nature of this article}.

We have restructured the Abstract as guided above. 

Surprisingly, in ‘Abstract’ the complete account [though very nice] is focused on ‘smallpox’ and no mention of ‘COVID-19’ {agreed, the article is on ‘Countering the potential re-emergence of a deadly infectious disease’ some mention of current pandemic / COVID-19 outbreak was/is expected to emphasize present context / implications / applications (mainly because at the end of ‘Introduction’ section, it is said that “we expect that our model would easily port to other infectious diseases such as COVID-19 through the consideration of a different set of disease states and parameters.”}. Many readers (as you are aware) decide to read full text on the basis of ‘abstract’.

We appreciate the reviewer’s suggestion towards enhancing the visibility of our article. Earlier we could not include the relevance to COVID-19 in our abstract because of the strict word limits. But, based on the reviewer’s suggestion we have now rewritten the Abstract (pp. 1- 2) to include a reference to COVID-19, while complying with the word limits. We have abbreviated other parts of the Abstract to comply with the specified limit on the number of words. 

Overall, the article is nicely presented, perfect by all the means, 100% correct/accurate (scientifically). However, there are few small questions: Why the table headings are below the tables? (usual practice is to place them above).

We have now placed the table headings above. 

Was it necessary to cover all possible scenarios [(1) No countermeasure - neither drugs nor vaccines administered, (2) Vaccines only, (3) Drugs only, (4) Both drugs and vaccines.] in such details?

We wanted to compare the efficacy 1) of different combinations of countermeasures, and 2) of using only vaccine with that for using only drugs. Hence, we covered all possible scenarios. Besides, we develop the models progressively, that is, we build on (1) to obtain (2) and (3) and obtain (4) by combining (2) and (3). In addition, scenarios (3) and (4) are quite complex owing to the presence of a large number of states and transitions between them. And the progressive organization improves clarity of presentation for the complex scenarios. 

Could not avoid/reduce number of figures? (there are 17 which is too much, I guess).

The number of figures is large because there are a large number of parameters involved and each figure illustrates the variation of the public health metrics with respect to one parameter. We apologize for the volume. 

Is it essential to include such a lengthy ‘appendix’ {A. Equations of Smallpox disease dynamics, B. Parameter estimation i.e., pages 33 to 49}?

Section A of the Appendix provides the equations that represent the mathematical model for the joint spread of an infectious disease and opinion that influences the spread. Our work is the first to present a mathematical model for this joint spread. Thus far the relation between disease and opinion spread processes have been extensively debated in public discourse, but the debates have largely remained qualitative. Our model and the associated numerical computations seek to provide quantitative foundations to this issue of immense public interest. We therefore present the model in sufficient detail so that it can be verified and reconstructed by readers for utilization in follow-up works. We have omitted repetitive (or almost repetitive) equations, however, and have only outlined how those directly follow from what has been presented. Section B of the Appendix describes how the various parameters of the model can be estimated from available data reported in literature on smallpox. The length in this part owes to the presence of very many parameters that arise naturally for the various scenarios we consider. We have presented these Sections in the Appendix because of the length of these parts. We hope that the placement of these parts in the Appendix enables the readers who are not interested in the details to obtain the essence of the paper by reading its main body. Nonetheless we apologize for the length and thank the reviewer for his patience in perusing this lengthy paper.

 

RESPONSE TO REVIEWER 2

It is not clear why much of the work is focused on smallpox, and using only a single disease spread model: this is over simplified given the lofty title of the paper

We have chosen smallpox as a specific example of an infectious disease, since (1) it is highly infectious (2) it has a high death rate and (3) its disease progression parameters are known with reasonable certainty owing to years of research. But, our framework ports to any other infectious disease (e.g., COVID-19) that spreads between individuals in proximity, through the consideration of a different set of disease states and parameters. We now illustrate this generalization considering COVID-19 as an example. We have now mentioned these in the last paragraph of Introduction (p. 7) and also in the Conclusion and Generalizations section (p. 42) and have provided the generalization in the Conclusion and Generalizations section (pp. 41 - 44) and Figure 20. 

The models are primarily based on homogeneous mixing assumptions, even though spatial heterogeneity is considered, but such models do not account for many disease factors that impact vaccine targeting strategies, and hence this again very much restricts the potential policy interventions and also actual population behavior in response to the disease

We have considered spatial heterogeneity throughout our paper. In the previous submission we had considered spatial homogeneity both in the model (pp. 10, 11, CEDE formulation in Appendix A (pp. 33 - 44), equations (1) – (33), and in the numerical computations (Figures 7, 8b, 8c, 9b, 9c, 10, 11, 13, 14b, 14c, 15 - 17, and Tables 1 – 5). In the current version of the manuscript, we have also considered spatial heterogeneity throughout: in the model (pp. 13, 14), result section (third paragraph of p. 17), in the CEDE formulation in Appendix A (pp. 50 - 69), equations (1) – (33), and in the numerical computations: Figures 8, 9c - f, 10 – 13, 15, 16b, 16c, 17 - 19, Tables 1b, 1c, 2, 3. Throughout the paper we have also considered both immunocompetent and immunodeficient individuals, in the model (pp. 8, 10), in CEDE formulation (pp. 50 – 69), in equations (1) – (33), in Figures 1 – 4, and in all numerical computations. We have also mentioned immunocompetency and immunodeficiency in the introduction (pp. 3, 5). The immunocompetent and immunodeficient individuals receive different kinds of vaccines and have different death rates (pp. 5, 8, 66, 68). Individuals become immunodeficient because of underlying medical conditions (p. 3). Thus, we had considered disease factors that impact vaccine targeting strategies. Note that the initial values of the state variables of the CEDE model that we formulated are obtained from the percentage of immunodeficient individuals in US, this percentage is available in literature on smallpox. We have now added a Section in Appendix to show how these initial values are calculated from the percentage of immunodeficient individuals in US (Section B.5, pp. 69 - 71). 

Vaccine efficacy seems not to be considered in the model

We had considered that vaccine provides immunity (that is prevents the disease) with a certain probability and had used 0.8 as the value of this probability following the literature on smallpox ([14] in the earlier version and [13] in the current version). Refer to pp. 16, 17, 39, 47 in the earlier version for the specific sentences in question. Recognizing that these mentions may not have been prominent, we have now stated at the start of the “vaccine only” section that upon receiving a vaccine the immunity level rises in a recipient to the point that the disease is prevented (even upon reception of the virus), only with a certain probability (red text on p. 10). We have also motivated the choice of all parameters at the start of the Result section, and explicitly stated that vaccine provides immunity with a certain probability (red text on p. 19), above and beyond the details in Section B of the Appendix (red text on p. 68). We have also reworded the relevant parts to improve clarity (refer to the red text on p. 58). 

Probability modeling for belief of fake news is much too basic, and doesn't consider that the population is not equally susceptible to such information; moreover, there is a dynamic to the belief itself that isn't discussed either. A much more thorough review of this topic and discussion of the basic assumptions is needed, and to also highlight how this limits the results.

As the first work in the field, while developing a mathematical model for the joint evolution of disease and opinions we had indeed resorted to some simplifications so as to focus on the essences. One such simplification has been to assume that the willingness to be vaccinated is a binary indicator. But, in practice, beliefs have different strengths. Thus, different individuals may be entrenched to different extents in their conviction against vaccines (equivalently, drawing from the reviewer’s characterization, different sections of the populace have different susceptibilities to information they receive). Also, individuals progress through different stages of the beliefs over time, rather than undergoing direct transition between the binaries of cooperative and non-cooperative (we guess this is what the “dynamic to the belief” phrase used by the reviewer means). In the revised version we have acknowledged these simplifications and have outlined some generalizations that render the model more realistic. Refer to the last paragraph of the Conclusion and Generalizations section (pp. 44, 45). 

Experimental results are not very thorough - for instance, they assume 10000 initially infected individuals but it is unclear why this makes any sense, how they are properly distributed among the population, how all such people ended up at the same disease state simultaneously, etc. Similar for mobility rates, R0, etc. This is critical since the eventual goal is to provide potential strategies to mitigate the scenario, but only very restricted scenarios are considered, without providing a lot of evidence as to their plausibility or risk in reality, and then the "policies" are implemented on the simplified models...hence, it is not clear how useful the results will actually be to those in public policy

We have revised the Result Section to motivate the choice of various parameters at the start (pp. 17 - 19). In the second paragraph of the Result Section, we now state the following: “We now describe how we choose the parameters for the numerical computation and their default values. In the numerical computation, unless otherwise stated, we use the default value for each parameter. But we also vary the value of each parameter in a wide range because in practice these parameters assume different values in different environments and there is no one standard value …”.

Note that mobility rate is an example of a parameter which assumes different values in different environments and there is no one standard value for it. Thus, we use a default value for it in many instances, but also vary it in a wide range in many other instances. We describe the default choice at the start of the Result Section (p. 17). Refer to Section 2 of the Result Section titled “Distribution of initial infections, initial cooperatives, and mobility of individuals have a significant impact”, for a study on how the fatality count changes when mobility rates are varied (Figures 12, 13 and the discussions on pp. 25 - 27). We also choose different values for the mobility rate when we study the variation of the fatality count with respect to the opinion spread rate (Figure 8 and the discussions on p. 21). Note that the pattern of the variation of the fatality count with respect to the opinion spread rate does not change as the mobility rate is varied, though the values of the fatality counts are different for different values of the mobility rates (discussion on pp. 21, 22). This statement holds for variation of the fatality count with respect to other parameters as well, though we did not produce figures to illustrate the same to not further lengthen the paper. 

We used the value of R0 that has been reported for smallpox in literature, namely of R0 = 6.9 as reported in [10, 13] ([11, 14] in the previous version). We obtain the default value of the disease spread rate ϕ from the above R0 following the methodology in [19, 30]. In the previous version of the paper, we had mentioned these in the Appendix (Section B.1, p. 45). Now, we have noted these at the start of the Result section (p. 18). We had also studied the variation of the fatality counts when ϕ (equivalently R0) is varied in a range, for different combinations of countermeasure (Figures 12a and 13a in the original submission). We retain these figures in the revised version (Figures 14a and 15a). 

We would first outline the contents of the originally submitted manuscript relevant to the comment on the number of initially infected individuals, and subsequently describe how we augmented those in the revised version to address the comments. In the originally submitted manuscript, we had extensively investigated the impact of the distribution across clusters (i.e., spatial distribution) of the initially infected individuals. Refer to the abstract, introduction (p. 4, last paragraph), Results section (p. 14) and Figures 7, 8b, 8c, 9b, 9c, 10, 11, 13, 14b, 14c, 15 – 17, and the discussions on them (pp. 18 – 33), Tables 1 – 5, and Discussion section (pp. 34 - 35). Distribution of the initially infected individuals across clusters is their distribution among the population. We had noticed that the patterns of variations of the fatality counts with respect to different parameters, remain same for different numbers of the initially infected individuals, though values of this metric increases with increase in the number of initially infected individuals. Thus, we had considered 10000 as the default value of the initially infected individuals and presented one figure considering two different numbers of initially infected individuals, namely 10000 and 1000 (Figures 6, 8a, 9a) to demonstrate that the pattern of variations remains same for both. For clarity of exposition, we had presented the results only for 10000 initially infected individuals in the rest of the figures. We had also presented the results only for the case that the initially infected individuals are all in the prodrome stage. 

Given the comment by the reviewer, in the revised version of the manuscript, we have justified the choice of our default value of the number and disease stage of the initially infected individuals at the start of the Result section (p. 18). We have included the results for 10000, 1000, 1 initially infected individuals in 3 figures (Figures 7, 9a, 9b). We have additionally considered cases where the initially infected individuals are distributed in prodrome and early rash stages (Figures 7 and 9a, 9b). We again notice that the patterns of the variation of fatality counts with respect to the rate of opinion exchange remain same for these different choices for the number and stages of disease of the initially infected individuals, though the values of the fatality counts are different for different choices. Therefore, for clarity of presentation, in the rest of the figures, we present the results only for 10000 initially infected individuals all of whom are in the prodrome stage. We have retained our findings with respect to the impact of the distribution of the initially infected individuals across clusters (i.e., spatial distribution). Refer to Figures 8 – 13, among these we have newly added: Figures 8b, 8d, part of 9, 10b, 10d, 11 and 13.

As for the results, it isn't clear what is new to the literature -- especially since the models are quite distant from reality, such conclusions like "fatality and opinion spread are strongly related" is intuitive and already generally accepted. Similarly for "distribution of infection and mobility of individuals have significant impact"..."administering drugs reduces fatality"...and so on.

To better position our findings in context of the existing literature, we have now rewritten 1) the last two paragraphs of the Introduction (pp. 5 - 7); and 2) the Discussion section in its entirety (pp. 37 - 41). For the same purpose, we have added a new section titled, “Conclusion and Generalization”, of specific importance in this section is the first paragraph (pp. 41, 42). In writing these, we have drawn both from the figures and tables that were already present in the previous submission and from the new figures and tables added to bolster our case (Figures 7, 8b, 8d, 10b, 10d, 11, and 13, part of Figure 9, Tables 1 and 2). Refer to the discussions on these additions in the Result Section - the text in red on pp. 20, 21 (Figure 7), p. 21 (Figures 8b and 8d), pp. 22, 23 (part of Figure 9), pp. 23, 24 (Figures 10b, 10d and 11), pp. 26, 27 (Figure 13), pp. 29, 30 (Table 1), and pp. 31, 32 (Table 2). In addition, we have revised the discussion for some existing figures: pp. 34 – 37 (Figures 17 – 19). 

RESPONSE TO REVIEWER 3

The authors should be more concise on the introduction. An overall flavour of the main result could be mentioned in the introduction. However, the place to summarize their finding is at the end of the paper in the section Conclusions.

We have now provided only a general flavor of the results in the Introduction section (refer to the last paragraph of the Introduction, pp. 6, 7). We have also edited the penultimate paragraph of the Introduction so that the broad results can be seen in context of the existing literature (pp. 5, 6). Following the reviewer’s guidance, we have created a Conclusion and Generalization section and summarized the findings in the first paragraph of this section (pp. 41, 42). These findings have been summarized from the Discussion section, which have also been rewritten (pp. 37 - 41). In writing these, we have drawn both from the figures and tables that were already present in the previous submission and from the new figures and tables added to bolster our case (Figures 7, 8b, 8d, 10b, 10d, 11, and 13, part of Figure 9, Tables 1 and 2). Refer to the discussions on these additions in the Result Section - the text in red on pp. 20, 21 (Figure 7), p. 21 (Figures 8b and 8d), pp. 22, 23 (part of Figure 9), pp. 23, 24 (Figures 10b, 10d and 11), pp. 26, 27 (Figure 13), pp. 29, 30 (Table 1), and pp. 31, 32 (Table 2). In addition, we have revised the discussion for some existing figures: pp. 34 – 37 (Figures 17 – 19). 

The following statement: “Finally, we expect that our model would easily port to other infectious diseases such as COVID-19 through the consideration of a different set of disease states and parameters.” should be placed in the section Conclusion instead of the Introduction, and it is meaningless if they do not explain how to adapt the model for the case of COVID. At least some clues.

One of the strengths of our model is that it readily ports to any other infectious disease (e.g., COVID-19) that spreads between individuals in proximity, through the consideration of a different set of disease states and parameters. We have therefore made a stronger case for this claim by outlining this generalization in the newly created “Conclusion and Generalization” section, considering COVID-19 as an example (pp. 41 – 44, Figure 20). Given that we have made this case, we have retained a sentence similar to the one that the Reviewer has flagged in the Introduction. The concluding part of the Introduction now reads as follows: “Finally, we have chosen smallpox as a specific example of an infectious disease, since (1) it is highly infectious (2) it has a high death rate and (3) its disease progression parameters are known with reasonable certainty owing to years of research. But, our framework ports to any other infectious disease (e.g., COVID-19) that spreads between individuals in proximity, through the consideration of a different set of disease states and parameters. We illustrate this generalization considering COVID-19 as an example.” (last paragraph of Introduction section, p. 7). Another reason we have retained this sentence is that Reviewer 1 has advised us to include this claim in the Abstract, which we have complied with. But if the Abstract has this claim, then we believe we should also mention it in the Introduction. We hope this will be acceptable. 

 Several claims in the introduction need to be referenced for the strongest receptivity of the paper from readers, as an example: Connection between individuals in geographically close neighborhoods is expected to be more frequent than those between the disparate ones leading to heterogeneous interaction rates.

We have now referenced this claim citing two additional papers (refer to the text in red on p. 4, [35, 41]). We have referenced several other claims in the introduction as well, as suggested by the reviewer. Refer to the texts in red on pp. 2, 3. 

The claim: “Note that a Monte Carlo simulation relying on an agent based stochastic simulation will not scale to that magnitude.” it is not necessary.

 Per the reviewer’s comment, we have removed the sentence from the revised version. 

The section methodology should be divided to explain the several states considered, the topology, and the dynamics, separately. For example, the section Vaccine should only have the explanation of the stages and not about the interactions (four different outcomes for interaction). That explanation should be transfer to another place.

We would like to explain the organization we have adopted. The four scenarios we consider: 1) no countermeasure 2) drug only 3) vaccine only 4) both drug and vaccine, have different (but overlapping) sets of states and transitions (dynamics). In an attempt to maximize accessibility while conserving space to the extent possible, we first explain the set of states and transitions that are common to all four (p. 7) and subsequently introduce the additional states and transitions needed in each scenario in the parts devoted to each (p. 8 for “no countermeasure”, pp. 9, 10 for the “drug only”, pp. 10, 11 for the “vaccine only”, and pp. 12, 13 for “both drug and vaccine”). Thus, each scenario has both additional states and transitions. We have included the four different outcomes in the vaccine section to illustrate the transitions introduced for that case; note that this illustration may enhance the quality of exposition as several new states (partition of each state to cooperative and noncooperative) and new transitions (corresponding to conversion of opinion) arise for this scenario. The last scenario, “both drug and vaccine”, builds on the “drug only” and “vaccine only” scenarios. We have dedicated a separate section for capturing the impact of spatial heterogeneity (pp. 13, 14); new states (partition of each state into various clusters) and new transitions (corresponding to mobility) arise when spatial heterogeneity (equivalently, topology) is considered. 

If I understand well, what is written inside the section Both drugs and vaccines, is that the individuals there receive either the drug or the vaccine. In that case, this new section should no exist, as it is part of the two previous ones.

An individual in the “both drug and vaccine” may enter the preempted state by receiving either the drug or the vaccine. But he may receive both drug and vaccine in the course of the outbreak. For example, a vaccine is effective in an individual (that is, his immunity does not increase to the level that future infection is prevented) only with a certain probability. If a vaccine is not effective in a recipient, then he does not transition to the preempted state after receiving the vaccine. Then even after receiving the vaccine he may develop the disease on receiving the virus from an infectious individual. Once he develops symptoms he may be treated with the drug, which may cure him and then he would transition to the preempted state. Thus, in this case the individual receives both vaccine and drug, in that sequence. Similarly, an individual who has not been infected may be treated with the drug and during the treatment he transitions to the preempted state. Subsequently, instead of continuing his drug treatment until the end of the outbreak, he may be vaccinated, and the drug treatment terminated after the vaccine develops the immunity in him which ensures that he remains in the preempted state. Thus, in this case the individual receives both drug and vaccine, in that sequence. We have now clearly stated these cases at the start of the section on “both drug and vaccine” scenario (text in red on p. 12).

Thus, the “both drug and vaccine” scenario is a generalization of the “drug only” and “vaccine only” scenario, rather than being part of these two. Thus, this section needs to exist. By generalization we mean that all results for the “drug only” and “vaccine only” scenarios can be reconstructed from the generalization (that is, the “both drug and vaccine” scenario) by appropriately choosing the parameters of the generalization. The reverse is not true. That is, one can not obtain the results for the generalization by choosing values of parameters of the “drug only” and “vaccine only” scenarios.

Figures 3 and 4 are really difficult to understand.

Figure 3 represents the state transitions for the “vaccine only” scenario which does have many states because it involves both the spread of disease and opinion. Accordingly Figure 3 indeed involves many states and many transitions. We had therefore color coded the states and transitions, explained those in the caption and also explained how the states in this case build upon those in Figure 1, since Figure 1 is easy to understand. We have now added some more explanation in the caption of Figure 3 (refer to the text in red) and also explained connections to Figure 2, which is easier to understand. We hope this makes it easier to understand. We have also explained the additional states and transitions in the text of the corresponding section (“vaccine only” section, pp. 10 - 12). Figure 4 is similar to Figure 3 (though not the same). If Figure 3 is understood, Figure 4 will become easy to understand as well. So, we have not changed the caption of Figure 4. 

Maybe it is better if the authors explain in a very simple way why it is necessary the addition of quadratic terms representing the interactional transitions due to the spread of opinions. Maybe the authors could explain the general equation for the transitions between general states, in a simple way.

We had explained the model in different levels of detail. We had first explained the nature of the terms (both quadratic and linear) by connecting our model to the metapopulation epidemic models that have been extensively used for modeling the spread of infectious diseases (pp. 11, 12 of the first submission). The connection involved mentions of both the similarities and differences of our work with the metapopulation epidemic models. We had explained in broad strokes as to why all interactional transitions, both in our model and existing metapopulation models, correspond to quadratic terms. Thus, since spread of opinion happens due to interaction, the associated transitions are modeled by quadratic terms in our work. Our work considers both spread of disease and opinions, metapopulation models available in literature consider only one kind of interactional transition, that through the spread of disease; therefore, our model has additional quadratic terms. Subsequently after stating the equations, we had explained each term, both quadratic and linear (pp. 12, 36, 39, 41 of the first submission). We have now enriched all these explanations by rewording and by providing additional content. Refer to the texts in red on pp. 15, 16, 54, 60, 61 of the current submission. Specifically, we now explain in greater detail as to why interactional transitions due to the spread of opinions are represented by quadratic terms (p. 60, 61). We have also proved that if the system is modeled as a Markov process, as the number of individuals increase, at every time instant the fraction of individuals in different states converge to the solution of the CEDE (Supporting information). The proof will rigorously establish the nature of each term (both quadratic and linear). 

Some conclusions are almost by definition, for example: “In addition, when the initial infections are uniformly distributed, fatalities in each cluster are the same irrespective of the mobility rates between the clusters. On the other hand, when the initial infections occur in one cluster only, total fatalities would escalate with an increase in mobility rates. Therefore, policymakers should opt for cordoning off areas where infection level is high before the

infection spreads widely; once the latter happens reducing mobility rates across areas is unlikely to contain the disease.”

And

“We have shown that evolution of opinion with respect to receptivity of vaccine has a strong impact on fatality count. Therefore, health authorities and other policymakers should seek to influence exchange of opinion towards enhancing receptivity to vaccine, possibly through health education seminars, workshops, vigorous dissemination of health information on social, digital and conventional media and through direct engagement with influencers on these platforms.

The authors could give more singular conclusions, showing new and interesting conclusions to the field obtained from their research.

We have now rewritten the Discussion section in its entirety to better present our conclusions (pp. 37 - 41). In writing these, we have drawn both from the figures and tables that were already present in the previous submission and from the new figures and tables added to bolster our case (Figures 7, 8b, 8d, 10b, 10d, 11, and 13, part of Figure 9, Tables 1 and 2). Refer to the discussions on these additions in the Result Section - the text in red on pp. 20, 21 (Figure 7), p. 21 (Figures 8b and 8d), pp. 22, 23 (part of Figure 9), pp. 23, 24 (Figures 10b, 10d and 11), pp. 26, 27 (Figure 13), pp. 29, 30 (Table 1), and pp. 31, 32 (Table 2). In addition, we have revised the discussion for some existing figures: pp. 34 – 37 (Figures 17 – 19). We have also provided a general flavor of the results in the Introduction section (refer to the last paragraph of the Introduction, p. 6, 7). We have edited the penultimate paragraph of the Introduction so that the broad results can be seen in context of the existing literature (p. 5, 6). 

A final paragraph summarizing the most important results is lacking.

Following the reviewer’s guidance, we have created a Conclusion and Generalization section and summarized the most important findings in the first paragraph of this section (pp. 41, 42). 

 I have the following question to the authors: When they say: “We, therefore, introduce a state called Preempted, which we denote by Q, into which individuals receiving the drug transition to with the specified probability. Once an individual transition to this state, he remains in it (since he is either cured or does not develop the disease) - thus, this is an absorbing state.” → Why? individuals receiving the drug transition can be in S again later on, can’t them?

If an individual has symptoms of smallpox when he is administered the drug, he is cured with a certain probability and if he is cured, he does not develop the disease again even if he receives the virus in future. If an individual who has not received the virus yet is treated with an antiviral drug, he does not develop the disease (with a high probability) during the period he is receiving the drug, e.g., TPOXX prevents smallpox (with a high probability) during the period it is taken orally twice daily. We had assumed that an individual who had no symptom when he is administered the drug, will continue to receive the drug until the outbreak is over or until he receives a vaccine. Thus, once an individual starts to receive the drug, he can no longer have smallpox (with the specified probability) until the containment of the outbreak (that is, until the end of the duration we consider). Thus, for the purpose of our model, without loss of generality, individuals enter the preempted state with the specified probability if they are administered drugs, and the preemption state is an absorbing state, i.e., individuals can only enter this state, not leave it. The probability is 0.99 before the onset of rash, 0.8 during early rash, and 0 during late rash ([13] in the paper). If the drug is not effective, we consider that the recipient does not enter the preempted state at all (rather than entering and leaving the state to return to S). Considering the probability that the drug is ineffective, even upon administering the drug, the recipient will not enter the preempted state with probability 0.01 (i.e., 1 - 0.99) if he starts receiving the drug before the onset of rash, with probability 0.2 (i.e., 1 - 0.8) if he starts receiving the drug during early rash, and with probability 1 (i.e., 1- 0) if he starts receiving the drug during late rash. Thus, in the duration we consider, that is, until the containment of the current outbreak, an individual who receives the drug can not be in S. 

In the first version of the paper, we had mentioned this briefly in the following sentence of p. 6: “We assume that once individuals without symptoms are administered this drug, to prevent the onset of smallpox even after possible exposure, they continue to receive the drug until the disease is completely contained.” Given the question by the reviewer, we have now elaborated further on this issue in context of the drug administering policies we consider. We have also cited additional references in this elaboration (refer to the text in red on pp. 9, 10, 12, the additional references cited are [4, 13, 37], reference [37] have been newly added to the paper).

Further small points

- The statement “In other words, intra-cluster interactions are higher compare to inter-cluster interactions” could be explained as grouped into communities.

We have edited this part of the paper as per the reviewer’s suggestion (p. 13). 

- The statement “Thus, the computation time gracefully scales with the increase of the size of the system” should be reformulated in more scientific terminology.

We have edited this part of the paper as per the reviewer’s suggestion (p. 16).

- The following part is meaningless, either the authors show the proof or cite the references: “The CEDE is however a deterministic model, while many of the state transitions are stochastic. Nevertheless, through an application of the strong law of large numbers, under some commonly made regularity assumptions on the stochastic evolutions, one can show that as the number of individuals increases, the fractions of individuals in different system states in the stochastic system converge to the solutions of the CEDE, and the convergence becomes exact in the limit that the number of individuals is infinity. For the mathematical proof we refer to a recent work by one of the co-authors involving the application of the CEDE in a different domain [23]. Thus the CEDE approximates the stochastic process better as the number of individuals increase. The commonly made regulatory assumptions under which the convergence guarantees hold are that the stochastic evolutions are Markov, that is, the amount of time an individual spends in each system state is exponentially distributed, which is what we assume to estimate the parameters of the system."

The proof is long but follows from a direct application of a result well-known in applied probability. We resorted to a similar application in [23] ([22] in the revised version) albeit in a different domain. Nonetheless, given the reviewer’s comment, we have provided the proof as a supporting information. 

- The following part is not related to the research performed in the paper:

The generality of the model is another important strength - it can accommodate information dynamics, arbitrary topologies, mobility patterns, countermeasure combinations, countermeasure application strategies and constraints (e.g., finite or infinite supply). Despite this generality, our model is computationally tractable and provides analytical convergence guarantees. Finally, the model can be easily modified to accommodate diseases such as COVID-19 that spreads through contact through appropriate selection of states and parameters.

The above describes what the model we propose can accommodate. In the subsequent Result section, we assess various public health metrics for different opinion dynamics, topologies, mobility patterns, countermeasure combinations, countermeasure application strategies and constraints (e.g., finite or infinite supply). As noted before, considering COVID-19 as an example, we have now shown that CEDE model readily ports to any other infectious disease (pp. 41 – 44, Figure 20). So, this paragraph is related to the research performed in the paper. We have now reworded this paragraph to enhance the exposition (p. 16).

---

## [Decision Letter · Decision Letter 1]

1 Jul 2021

PONE-D-20-32001R1

Countering the potential re-emergence of a deadly infectious disease - information warfare, identifying strategic threats, launching countermeasures

PLOS ONE

Dear Dr. Ali,

Thank you for submitting your manuscript to PLOS ONE. After careful consideration, we feel that it has merit but does not fully meet PLOS ONE’s publication criteria as it currently stands. Therefore, we invite you to submit a revised version of the manuscript that addresses the points raised during the review process.

As you can see the reviewers positively evaluated your manuscript. However, both of them remarked a difficulty for the readability of the manuscript due to its length. I suggest to slightly modify the results section, adding a small summary of the main finding at the end of each subsection. Moreover, it would be better to number the section and the subsections.  

We look forward to receiving your revised manuscript.

Kind regards,

Floriana Gargiulo

Academic Editor

PLOS ONE

Journal Requirements:

Reviewers' comments:

Reviewer's Responses to Questions

**Comments to the Author**

1. If the authors have adequately addressed your comments raised in a previous round of review and you feel that this manuscript is now acceptable for publication, you may indicate that here to bypass the “Comments to the Author” section, enter your conflict of interest statement in the “Confidential to Editor” section, and submit your "Accept" recommendation.

Reviewer #2: All comments have been addressed

Reviewer #3: All comments have been addressed

2. Is the manuscript technically sound, and do the data support the conclusions?

Reviewer #2: Yes

Reviewer #3: Yes

3. Has the statistical analysis been performed appropriately and rigorously? 

Reviewer #2: Yes

Reviewer #3: I Don't Know

4. Have the authors made all data underlying the findings in their manuscript fully available?

Reviewer #2: Yes

Reviewer #3: Yes

5. Is the manuscript presented in an intelligible fashion and written in standard English?

Reviewer #2: Yes

Reviewer #3: No

6. Review Comments to the Author

Reviewer #2: I would like to that the authors for the substantial revision. While the manuscript is still quite lengthy, and could use some tweaking in that regard, if the authors choose, I nonetheless recommend to accept as is.

Reviewer #3: Most of my comments were either corrected or answered. My main comment now is that the article is exaggeratedly long. Although the results must be correct, the analysis is difficult to follow. For this reason, the authors must, at least, draw substantial conclusions as a result of their entire mathematical apparatus to be added to the abstract, as well as, to the last section.

The abstract has been restructured. However, what they call Objectives in the current abstract, are not the objectives, they are more related to Motivation.

Again, the authors should include specific results at the end of the abstract, instead of light phrases like: “Our findings provide a quantitative foundation for various important elements of public health discourse…”

I do not know how else to help the authors better restructuring their article. So, those would be my last comments. I have not extra issues for Publication in PlosOne.

7. PLOS authors have the option to publish the peer review history of their article (what does this mean?). If published, this will include your full peer review and any attached files.

Reviewer #2: No

Reviewer #3: No

---

## [Author Response · Author response to Decision Letter 1]

20 Jul 2021

The Academic Editor has made two concrete suggestions based on the reviews of the revision we had submitted in the previous round. We have complied with the suggestions and provide the details pertaining to our compliance next. 

RESPONSE TO THE COMMENTS OF THE ACADEMIC EDITOR

As you can see the reviewers positively evaluated your manuscript. However, both of them remarked a difficulty for the readability of the manuscript due to its length. I suggest to slightly modify the results section, adding a small summary of the main finding at the end of each subsection. Moreover, it would be better to number the section and the subsections. 

We have complied with both the suggestions of the Academic Editor. Specifically, we have added a paragraph at the end of each subsection of the Result Section in which we have summarized the main finding in the subsection. Please refer to the text in red on p. 21 (Section 3.1), p. 25 (Section 3.2), pp. 30, 31 (Section 3.3), p.36 (Section 3.5). There was already such a paragraph at the end of Section 3.4 in the previous submission (p. 31 in the previous submission) which we have retained in the version we are submitting (pp. 31, 32 in the current version). We have also numbered the sections and subsections as advised by the Academic Editor. Now that we have numbered the sections and subsections, we have utilized the numberings to refer to the content as necessary in the rest of the paper (text in red on pp. 7, 10, 15, 30, 36 - 39, 52, 58). We hope that both these modifications enhance the readability of our work and we thank the Academic Editor for the thoughtful suggestion. 

Journal Requirements:

We confirm that we have reviewed our reference list and that it is complete and correct. We did not cite any paper that has been retracted.

---

## [Editor Report · Decision Letter 2]

29 Jul 2021

Countering the potential re-emergence of a deadly infectious disease - information warfare, identifying strategic threats, launching countermeasures

PONE-D-20-32001R2

Dear Dr. Ali,

We’re pleased to inform you that your manuscript has been judged scientifically suitable for publication and will be formally accepted for publication once it meets all outstanding technical requirements.

Kind regards,

Floriana Gargiulo

Academic Editor

PLOS ONE
---

## [Editor Report · Acceptance letter]

9 Aug 2021

PONE-D-20-32001R2 

Countering the potential re-emergence of a deadly infectious disease - information warfare, identifying strategic threats, launching countermeasures 

Dear Dr. Ali:

I'm pleased to inform you that your manuscript has been deemed suitable for publication in PLOS ONE. Congratulations! Your manuscript is now with our production department. 

Kind regards, 

on behalf of

Dr. Floriana Gargiulo 

Academic Editor

PLOS ONE